# One Model, Any Conjunctive Query: Graph Neural Networks for Answering Queries over Incomplete Knowledge Graphs

## Abstract

Motivated by the incompleteness of modern knowledge graphs, a new setup for query answering has emerged, where the goal is to predict answers that do not necessarily appear in the knowledge graph, but are present in its *completion*. In this paper, we formally introduce and study two query answering problems, namely, *query answer classification* and *query answer retrieval*. To solve these problems, we propose AnyCQ, a model that can classify answers to *any* conjunctive query on *any* knowledge graph. At the core of our framework lies a graph neural network trained using a reinforcement learning objective to answer Boolean queries. Trained only on simple, small instances, AnyCQ generalizes to *large queries* of *arbitrary* structure, reliably classifying and retrieving answers to queries that existing approaches fail to handle. This is empirically validated through our newly proposed, challenging benchmarks. Finally, we empirically show that AnyCQ can effectively transfer to *completely novel* knowledge graphs when equipped with an appropriate link prediction model, highlighting its potential for querying incomplete data.

## 1 Introduction

Knowledge graphs (KGs) are an integral component of modern information management systems for *storing*, *processing*, and *managing* data. Informally, a KG is a finite collection of facts representing different relations between pairs of nodes, which is typically highly incomplete (Toutanova and Chen, 2015; Vrandečić and Krötzsch, 2014). Motivated by the incompleteness of modern KGs, a new setup for classical query answering has emerged (Ren et al., 2020; Ren and Leskovec, 2020; Zhu et al., 2022; Bai et al., 2023; Yin et al., 2024; Wang et al., 2022; Xu et al., 2023), where the goal is to predict answers that do not necessarily appear in the KG, but are potentially present in its *completion*. This task is commonly referred to as complex query answering (CQA), and poses a significant challenge, going beyond the capabilities of classical query answering engines, which typically assume every fact missing from the *observable* KG is incorrect, following a form of *closed-world assumption* (Libkin and Sirangelo, 2009).

In its current form, CQA is formulated as a *ranking* problem: given an input query $Q(x)$ over a KG $G$, the objective is to rank all possible answers based on their likelihood of being a correct answer. Unfortunately, this setup suffers from various limitations. Firstly, this evaluation becomes intractable for cases where multiple free variables are allowed[1]. Moreover, to avoid explicitly enumerating solutions, existing methods need to resort to various heuristics and most of them can only handle tree-like queries (Arakelyan et al., 2020; Bai et al., 2023; Zhu et al., 2022) or incur an exponential overhead in more general cases (Yin et al., 2024). Consequently, the structural oversimplification of queries is also reflected in the existing benchmarks. We argue for an alternative problem formulation, more aligned with classical setup, to alleviate these problems.

**Problem setup.** In this work, we deviate from the existing ranking-based setup, and instead propose and study two query answering problems based on classification. Our first task of interest, *query answer classification*, involves classifying solutions to queries over knowledge graphs, as true or false. The second objective, *query answer retrieval*, requires predicting a correct answer to the query or deciding none exists.

---

[1]As a result, almost all existing proposals focus on queries with only *one* free variable.

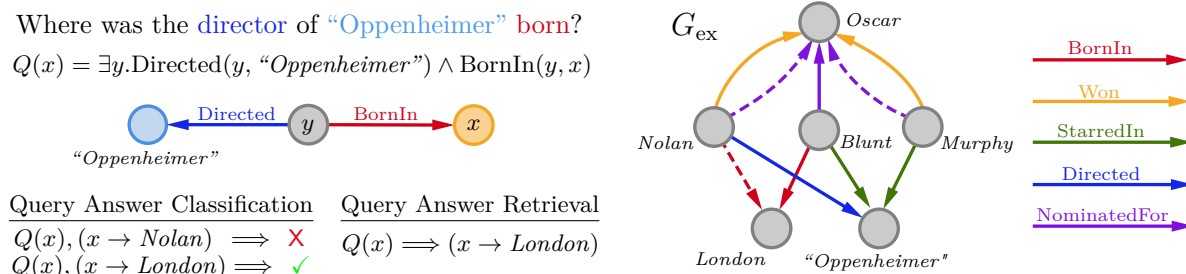

Figure 1: An example of a query $Q(x)$ over an incomplete knowledge graph, its query graph representation, and relevant query answer classification and query answer retrieval instances.

Let us illustrate these tasks on a knowledge graph $G_{\text{ex}}$ (Figure 1), representing relationships between actors, movies, and locations. The dashed edges denote the missing facts from $G_{\text{ex}}$ and we write $\tilde{G}_{\text{ex}}$ to denote the complete version of $G_{\text{ex}}$ which additionally includes all missing facts. Consider the following first-order query:

$$Q(x) = \exists y.\text{Directed}(y, \text{``Oppenheimer''}) \land \text{BornIn}(y, x),$$

which asks about the birthplace of the director of "Oppenheimer".

- **Query answer classification.** An instance of query answer classification is to classify a *given* answer, such as $x \to London$, as true or false based on the observed graph $G_{\text{ex}}$. In this case, the answer $x \to London$ should be classified as true, since this is a correct answer to $Q(x)$ in the complete graph $\tilde{G}_{\text{ex}}$, whereas any other assignment should be classified as false.

- **Query answer retrieval.** An instance of query answer retrieval is to predict a correct answer to $Q(x)$ based on the observed graph $G_{\text{ex}}$. In this case, the only correct answer is $x \to London$, which should be retrieved as an answer to the query $Q(x)$. If no correct answer exists, then None should be returned as an answer.

**Approach and contributions.** To solve these tasks, we introduce $\textsc{AnyCQ}^2$, a graph neural network that can predict the satisfiability of a Boolean query over *any* (incomplete) KG, provided with a function assessing the truth of unobserved links. $\textsc{AnyCQ}$ acts as a search engine exploring the space of assignments to the free and existentially quantified variables in the query, eventually identifying a satisfying assignment to the query. $\textsc{AnyCQ}$ can handle *any* existentially quantified first-order query in conjunctive or disjunctive normal form. Our contributions can be summarized as follows:

1. We extend the classical query answering problems to the domain of incomplete knowledge graphs and formally define the studied tasks of query answer classification and retrieval, introducing challenging benchmarks consisting of formulas with demanding structural complexity.

2. We propose $\textsc{AnyCQ}$, a neuro-symbolic framework based on graph neural networks for answering Boolean conjunctive queries over incomplete KGs, which is able to solve existentially quantified queries of *arbitrary* structure.

3. We demonstrate the strength of $\textsc{AnyCQ}$ on the studied objectives through various experiments, illustrating its strength on both benchmarks.

4. Specifically, we highlight its surprising generalization properties, including transferability between different datasets and ability to extrapolate to very large queries, *far beyond* the processing capabilities of existing query answering approaches.

We think that our work will inspire further research focusing on classifying query answers, a problem setup that has not been broadly considered so far.

---

[2]The code and data can be found in this GitHub repository.

## 2 Related work

**Link prediction.** Earlier models for link prediction (LP) on knowledge graphs, such as TransE (Bordes et al., 2013), RotatE (Sun et al., 2019), ComplEx (Trouillon et al., 2016) and BoxE (Abboud et al., 2020), focused on learning fixed embeddings for seen entities and relations, thus confining themselves to *transductive* setting. Later, graph neural networks (GNNs) emerged as powerful architectures, with prominent examples including RGCNs (Schlichtkrull et al., 2018) and CompGCNs (Vashishth et al., 2020). These models adapt the message-passing paradigm to multi-relational graphs, thus enabling *inductive* link prediction on unseen entities. Building on this, Zhu et al. (2021) designed NBFNets which exhibited strong empirical performance via conditional message passing due to its enhanced expressivity (Huang et al., 2023). Recently, ULTRA (Galkin et al., 2024a) became one of the first foundation models on LP over both unseen entities and relations.

**Complex query answering.** Complex query answering (CQA) (Ren et al., 2020; Ren and Leskovec, 2020) extends the task of link prediction to a broader scope of first-order formulas with one free variable, considering queries with conjunctions ($\wedge$), disjunctions ($\vee$) and negations ($\neg$). *Neuro-symbolic* models decompose the CQA task into a series of link prediction problems and employ fuzzy logic to aggregate these individual results. CQD (Arakelyan et al., 2020) was among the first to use fuzzy logic for CQA, relying on pre-trained embeddings and beam search for approximate inference. QTO (Bai et al., 2023) improved on this by exploiting the sparsity of neural score matrices to compute exact solutions without approximation. GNN-QE (Zhu et al., 2022) further advanced the field by training directly over queries, without relying on pre-trained embeddings. FIT (Yin et al., 2024) extended the methodology introduced in QTO to queries containing cycles, at the cost of high complexity. More recently, ULTRAQUERY (Galkin et al., 2024b) has been proposed, leveraging the CQA framework of GNN-QE with ULTRA as the link predictor, achieving the first foundation model for CQA, capable of zero-shot inference on unseen KGs. *Neural* methods generally rely on neural networks to deduce relations and execute logical connectives simultaneously. CQD-CO (Arakelyan et al., 2020) formulates query answering as a continuous optimization problem: it uses a pre-trained link predictor and assigns continuous embeddings to variables, optimizing the fuzzy logic score via gradient descent. LMPNN (Wang et al., 2022) employs a novel logical message-passing scheme, leveraging existing KG embeddings to conduct one-hop inferences on atomic formulas. CLMPT (Zhang et al., 2024) extends this methodology by using an attention-based mechanism to aggregate messages incoming from neighbor nodes. Q2T (Xu et al., 2023) utilized the adjacency matrix of the query graph as an attention mask in Transformers (Vaswani et al., 2017) model. While flexible, these approaches lack explicit variable grounding and symbolic interpretability, and tend to underperform as the size of the query graph increases.

**Combinatorial reasoning.** GNNs have emerged as a powerful tool for solving combinatorial optimization problems (Cappart et al., 2021). Their power to leverage the inherent structural information encoded in graph representations of instances has been successfully utilized for solving various combinatorial tasks (Joshi et al., 2019; Lemos et al., 2019; Bosnić and Šikić, 2023). As a method of our particular interest, ANYCSP Tönshoff et al. (2023), introduced a novel computational graph representation for arbitrary constraint satisfaction problems (CSP), demonstrating state-of-the-art performance on MAX-CUT, MAX-$k$-SAT and $k$-COL.

In this work, we identify answering conjunctive queries as a CSP, tailoring the ANYCSP framework to suit the task of deciding the satisfiability of Boolean formulas over incomplete KGs. Particularly, we integrate link predictors into our architecture to account for the necessity of inferring relations missing in the observable data. We also devise new guidance mechanisms to navigate the search during the early stages, targeting the large domain size. The augmented framework, named ANYCQ, inherits the extrapolation and generalization strength of ANYCSP, resulting in an efficient and effective model for query answer classification and retrieval.

## 3 Preliminaries

**Knowledge graphs.** A *knowledge graph* (KG) is a set of facts over a relational vocabulary $\sigma$, which is typically represented as a graph $G = (V(G), E(G), R(G))$, where $V(G)$ is the set of nodes (or vertices), $R(G)$ is the set of relation types, and $E(G) \subseteq R(G) \times V(G) \times V(G)$ is the set of relational edges (i.e., facts), denoted as $r(u, v) \in E(G)$ with $r \in R(G)$ and $u, v \in V(G)$. We write $G \models r(a, b)$ to mean $r(a, b) \in E(G)$. We consider each given KG $G = (V(G), E(G), R(G))$ as an *observable part* of a complete graph $\tilde{G} = (V(G), E(\tilde{G}), R(G))$

that consists of all true facts between entities in $V(G)$. Under this assumption, reasoning over the known facts $E(G)$ is insufficient, requiring deducing the missing edges $E(\tilde{G}) \backslash E(G)$. Note that this formulation follows the transductive scenario, in which $\tilde{G}$ covers the same sets of entities and relation types as $G$.

**Link predictor.** We call a *link predictor* for a KG $G$ a function $\pi : R(G) \times V(G) \times V(G) \to [0, 1]$, where $\pi(r, a, b)$ represents the probability of the atom $r(a, b)$ being a fact in $E(\tilde{G})$. The *perfect link predictor* $\tilde{\pi}$ for $\tilde{G}$ is defined as $\tilde{\pi}(r, a, b) = 1$ if $r(a, b) \in E(\tilde{G})$, and 0 otherwise.

**First-order logic.** A *term* is either a constant or a variable. A (binary) *atom* is an expression of the form $r(t_1, t_2)$, where $r$ is a binary relation, and $t_1, t_2$ are terms. A *fact*, or a *ground atom*, has only constants as terms. A *literal* is an atom or its negation. A variable in a formula is *quantified* (or *bound*) if it is in the scope of a quantifier; otherwise, it is *free*. A *Boolean formula* is a formula without any free variables. A *quantifier-free formula* is a formula that does not use quantifiers. For notational convenience, we write $\vec{x} = x_1, ..., x_k$ and $\vec{y} = y_1, ..., y_l$ to represent sequences of variables and $\Phi(\vec{x}, \vec{y})$ to represent a quantifier-free formula $\Phi$ using variables from $\{\vec{x}, \vec{y}\}$. Similarly, we write $\vec{a}$ to represent tuples of constants of the form $\vec{a} = a_1, ..., a_k$. For a first-order logic formula $\Phi(\vec{x})$ with $k$ free variables, we use the notation $\Phi(\vec{a}/\vec{x})$ to represent the Boolean formula obtained by substitution of each free occurrence of $x_i$ for $a_i$, for all $i$.

**Query answering.** The focus of this work is on conjunctive queries, i.e., existentially quantified first-order formulas. A *conjunctive query (CQ)* is a first-order formula of the form $Q(\vec{x}) = \exists \vec{y} \Phi(\vec{x}, \vec{y})$, where $\Phi(\vec{x}, \vec{y})$ is a conjunction of literals using variables from $\{\vec{x}, \vec{y}\}$. We reserve $\{\vec{y}\}$ for existentially quantified variables and $\{\vec{x}\}$ for free variables. If the query is Boolean, we write $Q = \exists \vec{y} \Phi(\vec{y})$.

Given a KG $G$ and a query $Q(\vec{x}) = \exists \vec{y} \Phi(\vec{x}, \vec{y})$, the assignments $\nu : \{\vec{x}\} \to V(G)$, $\mu : \{\vec{y}\} \to V(G)$ respectively map the *free* and *quantified* variables to constants. For notational convenience, we denote with $\vec{x} \to \vec{a}$ the assignment $x_1 \to a_1, ..., x_k \to a_k$. We represent by $\Phi(\vec{a}/\vec{x}, \vec{e}/\vec{y})$ the formula obtained by substituting the variables with constants according to the assignments $\vec{x} \to \vec{a}$ and $\vec{y} \to \vec{e}$. We write $\nu_{x \to a}$ for an assignment such that $\nu_{x \to a}(x) = a$ and $\nu_{x \to a}(z) = \nu(z)$ whenever $z \neq x$.

A Boolean query $Q = \exists \vec{y} \Phi(\vec{y})$ evaluates to true on $G$, denoted $G \models Q$, if there exists an assignment $\vec{y} \to \vec{e}$ such that all positive facts that appear in $\Phi(\vec{e}/\vec{y})$, appear in the set $E(G)$ and none of the negated facts that appear in $\Phi(\vec{e}/\vec{y})$ are present in $E(G)$. In this case, the assignment $\vec{y} \to \vec{e}$ is called a *match*. For a query $Q(\vec{x}) = \exists \vec{y} \Phi(\vec{x}, \vec{y})$, an assignment $\vec{x} \to \vec{a}$ is called an *answer* if $G \models Q(\vec{a}/\vec{x})$.

In our study, we make a distinction between *easy* and *hard* answers to queries, depending on whether the answers can already be obtained from the observed KG $G$ or only from its completion $\tilde{G}$. Formally, given an observed KG $G$ and its completion $\tilde{G}$, we say that an answer $\vec{a}$ is *easy* (or *trivial*) if $G \models Q(\vec{a}/\vec{x})$. If, however, $\tilde{G} \models Q(\vec{a}/\vec{x})$ while $G \nvDash Q(\vec{a}/\vec{x})$ then the answer is *hard* (or *non-trivial*).

**Query graphs.** Given a conjunctive query $Q(\vec{x})$, its *query graph* has the terms of $Q(\vec{x})$ as vertices, and the atoms of $Q(\vec{x})$ as relational edges. If the underlying undirected version of the resulting query graph is a tree, we call the query *tree-like*, otherwise, we say it is *cyclic*.

**Fuzzy logic.** Fuzzy logic extends Boolean Logic by introducing continuous truth values. A formula $Q$ is assigned a truth value in range $[0, 1]$, evaluated recursively on the structure of $Q$ using $t$-norms and $t$-conorms. In particular, Gödel $t$-norm is defined as $\top_G(a, b) = \min(a, b)$ with the corresponding $t$-conorm $\bot_G(a, b) = \max(a, b)$. For any existential Boolean formulas $Q$ and $Q'$, the respective *Boolean formula score* $S_{\pi, G}$, w.r.t. a link predictor $\pi$ over a KG $G$ is then evaluated recursively as:

$$S_{\pi, G}(r(a, b)) = \pi(r, a, b)$$
$$S_{\pi, G}(\neg Q) = 1 - S_{\pi, G}(Q)$$
$$S_{\pi, G}(Q \wedge Q') = \min(S_{\pi, G}(Q), S_{\pi, G}(Q'))$$
$$S_{\pi, G}(Q \vee Q') = \max(S_{\pi, G}(Q), S_{\pi, G}(Q'))$$
$$S_{\pi, G}(\exists x. Q'(x)) = \max_{a \in V(G)} S_{\pi, G}(Q'(a/x))$$

Note that the Boolean formula score with a classification threshold of 0.5 is equivalent to applying propositional logic over link predictions binarized at the same threshold. We hence adopt this setup to mitigate score degradation in large queries and ensure more stable evaluation (see Appendix A.6).

# 4 Query answering on incomplete KGs

## 4.1 Limitations of existing problem formulations

**Intractability of high-arity query evaluation.** The objective of complex query answering is to rank all possible answers to a given logical formula. Already for queries $Q(x_1, x_2)$ with two free variables, this entails scoring $|V(G)|^2$ pairs of entities $(a_1, a_2) \in V(G)^2$, which is computationally infeasible for modern knowledge graphs (Toutanova et al., 2015; Carlson et al., 2010) containing thousands of nodes. As a result, most of the existing approaches are not designed to handle higher arity queries, either resolving to inefficient enumeration strategies or approximating answers by marginal predictions. This scalability bottleneck has already been observed by Yin et al. (2023), who suggested more tractable evaluation methodologies, yet again being only marginal approximations of the true performance. Therefore, we argue that the ranking-based formulation has significantly limited the progress in query answering over formulas with multiple free variables.

**Limited structural complexity in existing benchmarks.** A related limitation lies in the structural simplicity of existing benchmarks (Ren and Leskovec, 2020). Standard CQA literature predominantly focuses on tree-like queries, which aligns with the capabilities of most current models. More recently, Yin et al. (2023) introduced a dataset containing cyclic queries and queries with up to two free variables; however, the overall structures remained constrained - featuring at most four variables and a single cycle. We argue that addressing structurally richer queries is essential for advancing automated reasoning systems. In real-world applications to autonomous systems, such as an AI trip planner that simultaneously books flights, accommodations, and activities while satisfying budget and availability constraints, the underlying reasoning involves large, complex queries with multiple variables. As AI agents become more capable, the complexity of the queries they must resolve is only expected to increase, hence requiring more expressive answering engines.

**Lack of probabilistic calibration in ranking-based methods.** Practical applications often demand binary decisions - answering questions like "Is X true?" or "What is the correct answer to Y?", requiring models to classify candidate solutions as either true or false (van Bakel et al., 2021). However, many ranking-based CQA methods do not natively support this decision-making paradigm, as they focus on ordering candidates without enforcing a meaningful threshold to distinguish valid answers from incorrect ones. While many of these models are trained using classification losses, such training does not guarantee the output scores correspond to calibrated satisfiability probabilities. In fact, several approaches rely on Noisy Contrastive Estimation (Wang et al., 2022; Zhang et al., 2024) or apply ad hoc score-to-probability transformations (Bai et al., 2023; Yin et al., 2024), further weakening the reliability of predicted scores in downstream tasks.

## 4.2 Query Answer Classification & Query Answer Retrieval

To address these limitations, we propose two new query answering tasks designed to provide more targeted responses while ensuring scalability for more complex logical queries.

*Query answer classification* reflects real-world scenarios where users seek to verify the correctness of a specific answer rather than navigating through a ranked list of possibilities. It better captures the nature of many real-world queries, aligning the model's output with the user's intent:

QUERY ANSWER CLASSIFICATION (QAC)

**Input**: A query $Q(\vec{x})$, tuple $\vec{a}$ and an observed graph $G$.
**Output**: Does $\tilde{G} \models Q(\vec{a}/\vec{x})$ hold?

*Query answer retrieval* assesses the correctness of the top-ranked result. By requiring models to either deliver a correct assignment to the free variables of the input query or confidently assert the absence of one, QAR aligns more closely with practical decision-making processes, ensuring the output is both relevant and reliable:

QUERY ANSWER RETRIEVAL (QAR)

**Input**: A query $Q(\vec{x})$ and an observed graph $G$.
**Output**: $\vec{x} \rightarrow \vec{a}$ where $\tilde{G} \models Q(\vec{a}/\vec{x})$ or None

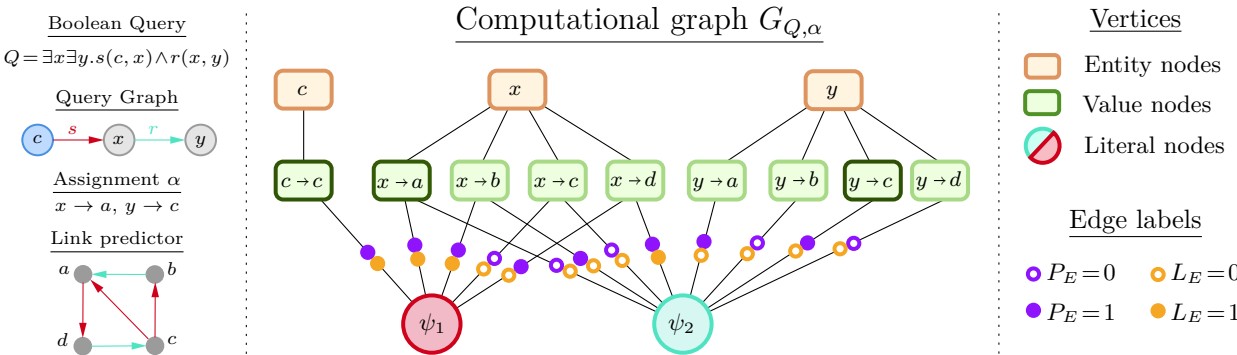

Figure 2: The ANYCQ computational graph $G_{Q,\alpha}$ for the query $Q = \exists x \exists y.s(c,x) \wedge r(x,y)$ and assignment $\alpha : x \to a, y \to c$, over a KG with 4 entities and 2 relation types. Literals $\psi_1$ and $\psi_2$ correspond to $s(c,x)$ and $r(x,y)$, respectively. Predictions of the equipped link predictor exceeding the probability threshold of 0.5 are displayed in a graph form in the bottom left. Value nodes corresponding to the assignment $\alpha$ are highlighted.

**Reduction to conjunctive query answering.** Following earlier works (Wang et al., 2022; Yin et al., 2024; Zhang et al., 2024), we note that each existential first-order logic (EFO) query $Q(\vec{x})$ can be represented as an equivalent $Q'(\vec{x}) \equiv Q(\vec{x})$ in disjunctive normal form: $Q'(\vec{x}) = \exists \vec{y}.(D_1(\vec{x}, \vec{y}) \vee ... \vee D_n(\vec{x}, \vec{y}))$, where each $D_i$ is a conjunction of literals. Under this decomposition, $Q'(\vec{x}) \equiv Q_1(\vec{x}) \vee ... \vee Q_n(\vec{x})$, where $Q_i(\vec{x}) = \exists \vec{y}.D_i(\vec{x}, \vec{y})$. Hence $Q(\vec{a}/\vec{x})$ is satisfiable if and only if one of $Q_i(\vec{a}/\vec{x})$ is satisfiable, reducing the problem to *conjunctive queries.* Contrary to the ranking setting, which requires combining the results for components $Q_i(\vec{a}/\vec{x})$ to obtain a score for $Q(\vec{a}/\vec{x})$, in the classification-based formulation, this aggregation considers only binary 'true' and 'false' labels. Consequently, we concentrate on conjunctive queries, since answering $Q(\vec{x})$ is equivalent to solving each $Q_i(\vec{x})$ individually, and the need for merging the outcomes does not introduce further complexity.

## 5 AnyCQ: framework for query answering

To address the introduced tasks of query answer classification and retrieval, we propose a neuro-symbolic framework for scoring arbitrary existential Boolean formulas, called ANYCQ. Let $\pi$ be a link predictor for an observable knowledge graph $G$. An ANYCQ model $\Theta$ equipped with $\pi$ can be viewed as a function $\Theta(G, \pi) : \mathsf{CQ}^0(G) \to [0,1]$ where $\mathsf{CQ}^0(G)$ is the class of conjunctive Boolean queries over the same vocabulary as $G$. For input $Q = \exists \vec{y}.\Phi(\vec{y})$, $\Theta$ searches over the space of assignments to $\vec{y}$ for:

$$\alpha_{\max} = \underset{\alpha : \vec{y} \to V(G)}{\arg \max} \, S_{\pi, G}(\Phi(\alpha(\vec{y})/\vec{y})).$$

and returns an approximation $\Theta(Q|G, \pi)$ of $S_{\pi, G}(Q)$:

$$\Theta(Q|G, \pi) = \underset{\text{visited } \alpha}{\max} \, S_{\pi, G}(\Phi(\alpha(\vec{y})/\vec{y}))$$

Note that unfolding the Boolean formula score gives:

$$S_{\pi, G}(Q) = S_{\pi, G}(\Phi(\alpha_{\max}(\vec{y})/\vec{y})) \approx \Theta(Q|G, \pi).$$

Hence, leveraging the strength of GNNs to find optimal solutions to combinatorial optimization problems, we can recover promising potential candidates for $\alpha_{\max}$, allowing to accurately estimate $S_{\pi, G}(Q)$.

**Overview.** During the search, our method encodes the query $Q$ and its relation to the current assignment $\alpha$ into a computational graph $G_{Q,\alpha}$ (Section 5.1). This graph is then processed with a simple GNN $\theta$ (whose architecture is described in details in Appendix A.1), which updates its hidden embeddings and generates distributions $\mu$ from which the next assignment $\alpha'$ is sampled (Section 5.2).

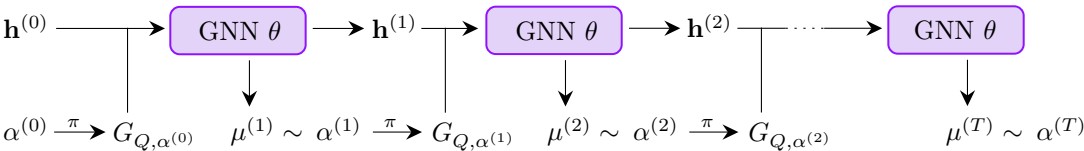

Figure 3: Overview of the ANYCQ framework. At search step $t$, GNN $\theta$ generates hidden embeddings $\mathbf{h}^{(t)}$ and a set of distributions $\mu^{(t)}$. The next assignment $\alpha^{(t)}$ is sampled from $\mu^{(t)}$ and used to update the edge labels of $G_{Q,\alpha^{(t)}}$ with respect to the equipped link predictor $\pi$.

## 5.1 Query representation

We transform the input queries into an undirected computational graph (Figure 2), whose structure is adopted from ANYCSP (Tönshoff et al., 2023). Consider a conjunctive Boolean query $Q = \exists \vec{y}.\Phi(\vec{y})$ over a knowledge graph $G$, with $\Phi$ quantifier-free, and let $\pi$ be a link predictor for $G$. Let $c_1, ..., c_n$ be constant symbols mentioned in $\Phi$, and $\psi_1, ..., \psi_l$ be the literals in $\Phi$. We define the domain $\mathcal{D}(e)$ of the term $e$ as $\mathcal{D}(y) = V(G)$ for each existentially quantified variable $y$ and $\mathcal{D}(c_i) = \{c_i\}$ for each constant $c_i$. Given an assignment $\alpha : \vec{y} \to \vec{a}$, the computational graph $G_{Q,\alpha}$ is constructed as follows:

**Vertices.** The vertices of $G_{Q,\alpha}$ are divided into three groups. Firstly, the *term* nodes, $v_{y_1}, ..., v_{y_k}$ and $v_{c_1}, ..., v_{c_n}$, represent variables and constants mentioned in $\Phi$. Secondly, *value vertices* correspond to feasible term-value assignments. Formally, for each term $e$ mentioned in $\Phi$ and any value $a \in \mathcal{D}(e)$, there exists a value vertex $v_{e \to a}$. Finally, *literal nodes* $v_{\psi_1}, ..., v_{\psi_l}$ represent literals $\psi_1, ..., \psi_l$ of $\Phi$.

**Edges.** We distinguish two types of edges in $G_{Q,\alpha}$. The *term-value* edges connect term with value nodes: for any term vertex $v_e$ representing $e$ and any $a \in \mathcal{D}(e)$, there exists an undirected edge $\{v_e, v_{e \to a}\}$. Additionally, *value-literal* edges are introduced to propagate information within literals. If a literal $\psi_i$ mentions a term $e$, then for all $a \in \mathcal{D}(e)$ there is an edge between $v_{\psi_i}$ and $v_{e \to a}$.

**Edge labels.** Edge labels embed the predictions of the link predictor $\pi$ into the computational graph $G_{Q,\alpha}$ to support guided search. Each value-literal edge connecting a literal vertex $v_{\psi_i}$ with a value node $v_{e \to a}$ is annotated with the *potential edge* (PE) and the *light edge* (LE) labels. The PE label $P_E(v_{\psi_i}, v_{e \to a})$ is meant to answer the question: "Can $\psi_i$ be satisfied under the substitution $e \to a$?". For example, when $\psi_2 = s(x, y)$, as in Figure 2, $P_E(v_{\psi_2}, v_{x,a})$ denotes whether $\exists y.s(a, y)$ is satisfiable, according to $\pi$. We pre-compute the PE labels using $\pi$, binarizing the Boolean formula scores of the form $S_{\pi,G}(\exists y.s(a, y))$ with the threshold 0.5.

In contrast to PE labels, which are independent of the assignment $\alpha$, light edge (LE) labels reflect how local changes to $\alpha$ affect satisfiability of the literals. Formally, we set $L_E(v_{\psi_i}, v_{e \to a}; \alpha) = 1$ if $\psi_i$ is satisfied under the assignment $\alpha_{z \to a}$, and 0 otherwise. In other words, LE labels answer the question: "If we change $\alpha$ so that $z$ is assigned to $a$, will $\psi_i$ be satisfied?". Satisfiability is again determined by binarizing the prediction score returned by the link predictor $\pi$. Hence, through these edge labels, $\pi$ effectively guides the search toward promising updates in the assignment space. Further explanation of edge labels is provided in Appendix A.7.

## 5.2 AnyCQ search process

The outline of the search process conducted by ANYCQ is presented in Figure 3. Before the search commences, the hidden embeddings $\mathbf{h}^{(0)}$ of all *value nodes* are set to a pre-trained vector $\mathbf{h} \in \mathbb{R}^d$ and an initial assignment $\alpha^{(0)}$ is drafted, independently sampling the value for each variable $y \in \{\vec{y}\}$ uniformly at random from $\mathcal{D}(y)$. The variable and literal nodes are not assigned any hidden embeddings, serving as intermediate steps for value node embedding updates. At the beginning of search step $t$, $G_{Q,\alpha^{(t-1)}}$ is processed with a GNN $\theta$, which generates new value node embeddings $\mathbf{h}^{(t)}$, and for each variable $y \in \vec{y}$ returns a distribution $\mu_y^{(t)}$ over $\mathcal{D}(y)$. Finally, the next assignment $\alpha^{(t)}$ is sampled by drawing the value $\alpha^{(t)}(y)$ from $\mu_y^{(t)}$, independently for each $y \in \{\vec{y}\}$. A precise description of the architecture of $\theta$ is provided in Appendix A.1. The search terminates after $T$ steps, and the generated assignments $\alpha^{(0)}, \alpha^{(1)}, ..., \alpha^{(T)}$ are used to approximate $S_{\pi,G}(Q)$:

$$\Theta(Q|G, \pi) = \max_{0 \leq t \leq T} S_{\pi,G}\left(\Phi\left(\alpha^{(t)}(\vec{y})/\vec{y}\right)\right)$$

### 5.3 Training

During training on each dataset, we equip AnyCQ with a predictor $\pi_{\text{train}}$, representing the training graph $G_{\text{train}}$. Thus, the only trainable component of $\Theta$ remains the GNN $\theta$. We utilize the training splits from the existing CQA datasets Ren and Leskovec (2020), hence limiting the scope of queries viewed during training to formulas mentioning at most three variables. Moreover, we restrict the number of search steps $T$ to at most 15, encouraging the network to quickly learn to apply logical principles locally. Inspired by prior work on combinatorial optimization (Shi and Zhang, 2022; Tönshoff et al., 2023; Abe et al., 2019), we train $\theta$ in a reinforcement learning setting via REINFORCE Williams (2004), treating $\theta$ as a search policy network with the objective of maximizing $\Theta(Q|G, \pi)$. This setup enables AnyCQ to generalize across different query types, scaling to formulas of size several times larger than observed during training, as demonstrated in Section 6. The complete methodology is presented in Appendix A.2.

### 5.4 Theoretical and conceptual properties

AnyCQ is *complete* given sufficiently many search steps – *any* AnyCQ model equipped with *any* link predictor will eventually return the corresponding Boolean formula score:

**Theorem 5.1.** *(Completeness) Let $Q = \exists \vec{y}.\Phi(\vec{y})$ be a conjunctive Boolean query and let $\Theta$ be any AnyCQ model equipped with a predictor $\pi$. For any execution of $\Theta$ on $Q$, running for $T$ steps:*

$$\mathbb{P}\left(\Theta(Q|G, \pi) = S_{\pi,G}(Q)\right) \to 1 \qquad as \qquad T \to \infty$$

When AnyCQ is equipped with the *perfect link predictor* for $\tilde{G}$, we can guarantee the *soundness* of predictions, i.e., all positive answers will be correct (proofs in Appendix B):

**Theorem 5.2.** *(Soundness) Let $Q = \exists \vec{y}.\Phi(\vec{y})$ be a conjunctive Boolean query over a knowledge graph $G$ and let $\Theta$ be any AnyCQ model equipped with a perfect link predictor $\tilde{\pi}$ for $\tilde{G}$. If $\Theta(Q|G, \tilde{\pi}) > 0.5$, then $\tilde{G} \models Q$.*

**Transferability.** By construction, our trained GNN $\theta$ is independent of the equipped link predictor $\pi$ and the knowledge graph $G$. Therefore, once trained, an AnyCQ model can be applied to answer queries over any dataset, with the only necessary augmentation being the change of the equipped link predictor. We validate this transferability in Section 6.4, comparing the performance of models trained on FB15k-237 and NELL datasets on the QAR task.

**Generality.** Although this work focuses on knowledge graphs with binary relations, by construction, AnyCQ can handle predicates with arbitrary arity requiring no changes in its structure. Assuming the availability of a relevant link predictor, AnyCQ can hence be applied to hyper-relational data. Similarly, by equipping a fully inductive link predictor (Galkin et al., 2024a), AnyCQ can serve as a general query answering engine without the need for interchanging the equipped predictor between input samples. Experiments with equipping ULTRA (Appendix D.5) already point out the potential of AnyCQ in the inductive setting. Importantly, our method is not limited to scoring only conjunctive Boolean queries and can process all formulas in conjunctive or disjunctive normal form (see Appendix A.4).

## 6 Experimental evaluation

We empirically evaluate AnyCQ on the tasks of QAC (Section 6.2) and QAR (Section 6.3). We also conduct two ablation studies (Section 6.4): first, we assess AnyCQ with a perfect link predictor to isolate and measure the quality of the search engine independent of the predictor's imperfections; second, we examine the generalizability of AnyCQ by applying it to out-of-distribution KGs.

### 6.1 Experimental setup

**Benchmarks and datasets.** Existing benchmarks (Ren and Leskovec, 2020; Yin et al., 2023) comprise formulas with simple structures, thereby impeding the comprehensive evaluation and advancement of novel methodologies. We address this gap by creating new datasets on top of well-established benchmarks, consisting

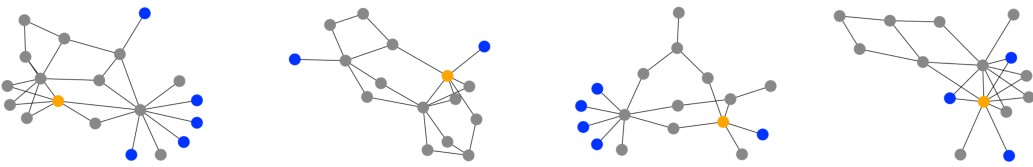

Figure 4: Examples of query graphs of formulas from our FB15k-237-QAR benchmark. Blue nodes represent constants, grey nodes - distinct existentially quantifies variables, and orange nodes - free variables.

of queries with demanding structural complexity. Specifically, we generate formulas mentioning up to 20 distinct terms, allowing the presence of multiple cycles, non-anchored leaves, long-distance reasoning steps, and multi-way conjunctions (see Figure 4). The generation process is described in Appendix C. Building on the existing CQA datasets, we propose two novel benchmarks for QAC: FB15k-237-QAC and NELL-QAC, each divided into 9 splits, consisting of small and large formulas. For the task of QAR, we observe that many instances of the simpler query structures, inherited from existing CQA benchmarks, admit easy answers, i.e. have at least one satisfying assignment supported entirely by observed facts. Combined with their limited structural complexity, this makes them trivial under the QAR objective, which only requires recovering a single correct answer. To evaluate reasoning under incompleteness and structural difficulty, we introduce new benchmarks: FB15k-237-QAR and NELL-QAR, consisting of large formulas with up to three free variables. In QAC, we focus exclusively on single-variable instances, as multi-variable cases reduce trivially to the single-variable setting. For example, $\langle Q(x_1, x_2), (a_1, a_2), G\rangle$ is equivalent to a single-variable instance $\langle Q(x_1, a_2/x_2), a_1, G\rangle$, as they both ask if $\tilde{G} \models Q(a_1/x_1, a_2/x_2)$.

**Baselines.** As the baselines for the small-query split on our QAC task, we choose the state-of-the-art solutions from CQA capable of handling the classification objective: QTO (Bai et al., 2023), FIT (Yin et al., 2024), GNN-QE (Zhu et al., 2022) and ULTRAQUERY (Galkin et al., 2024b). Considering the large-query splits, we were surprised to notice that no existing approaches can be applied in this setting, as none of them can simultaneously: 1) efficiently handle **cyclic** queries and 2) produce **calibrated** probability estimates, without the knowledge of the trivial answers.

Indeed, standard methods like BetaE (Ren and Leskovec, 2020), CQD (Arakelyan et al., 2020), ConE (Zhang et al., 2021), GNN-QE (Zhu et al., 2022) or QTO (Bai et al., 2023) are limited to tree-like queries. Neural approaches, such as LMPNN (Wang et al., 2022) or CLMPT (Zhang et al., 2024), are trained using Noisy Contrastive Estimation; hence their predictions do not meaningfully translate to desired probabilities. Finally, FIT (Yin et al., 2024) and Q2T (Xu et al., 2023) require transforming scores predicted by their ComplEx-based link predictors, while all known schemes (see Appendix D.1) assume the set of easy answers is known, or otherwise, trivial to recover. Hence, we furthermore use an SQL engine, implemented by DuckDB (Raasveldt and Mühleisen, 2019), reasoning over the observable graph. For the same reasons, extended by the need of reasoning over queries with multiple variables, we consider *only* the SQL engine as the baseline for QAR experiments. In both cases, we limit the processing time to 60 seconds, ensuring termination in a reasonable time. Additional evaluations ablating the impact of this timeout, using 30, 60, and 120 seconds thresholds, are included in Appendix E. Training details for the considered baselines are provided in Appendix D.

**Methodology.** Given a Boolean query $Q$ over an observable KG $G$, an ANYCQ model $\Theta$ equipped with a link predictor $\pi$ for $G$ can decide if $\tilde{G} \models Q$, by returning whether $\Theta(Q|G, \pi) > 0.5$. We use this functionality to solve QAC instances by applying our ANYCQ models directly to $Q(\vec{a}/\vec{x})$. For the QAR task, given a query $Q(\vec{x})$ over an observable KG $G$, we run our ANYCQ framework on the Boolean formula $\exists \vec{x}.Q(\vec{x})$, returning None if the returned $\Theta(\exists \vec{x}.Q(\vec{x})|G, \pi)$ was less than 0.5. Otherwise, we return $\alpha(\vec{x})$ where $\alpha$ is the visited assignment maximizing the Boolean formula score. In both scenarios, we perform 200 search steps on each input instance in the large query splits, and just 20 steps for small QAC queries. We equip NBFNet as the link predictor for both QAC and QAR evaluations (details in Appendix D.5).

**Metrics.** Given the classification nature of both our objectives, we use the F1-score as the metric for query answer classification and retrieval (see Appendix C.4 for details). In QAR, we mark a positive solution as

Table 1: Average F1-scores of considered methods on the query answer classification task.

| Dataset | Model | 2p | 3p | pi | ip | inp | pin | 3-hub | 4-hub | 5-hub |
|---|---|---|---|---|---|---|---|---|---|---|
| **FB15k-237-QAC** | SQL | 66.0 | 61.7 | 70.0 | 67.0 | 78.1 | 74.8 | 37.0 | 32.2 | 35.3 |
| | QTO | 67.1 | 64.4 | 70.8 | 67.7 | 78.5 | 75.9 | – | – | – |
| | FIT | 68.0 | 65.1 | 71.4 | 67.8 | 78.6 | 76.7 | – | – | – |
| | GNN-QE | **77.1** | **73.5** | 80.1 | **81.2** | **79.0** | 77.0 | – | – | – |
| | UltraQuery | 75.2 | 68.9 | 79.8 | 76.8 | 75.9 | **78.6** | – | – | – |
| | AnyCQ | 75.8 | 71.3 | **82.1** | 78.8 | 76.7 | 75.7 | **52.4** | **49.9** | **51.9** |
| **NELL-QAC** | SQL | 60.9 | 58.8 | 63.3 | 59.6 | **76.7** | 74.9 | 33.9 | 31.4 | 27.0 |
| | QTO | 63.9 | 64.1 | 68.2 | 61.7 | 74.5 | 75.3 | – | – | – |
| | FIT | 63.9 | 64.6 | 68.4 | 61.7 | 73.6 | **75.7** | – | – | – |
| | GNN-QE | 70.4 | 69.7 | 71.2 | 72.1 | 72.2 | 74.9 | – | – | – |
| | UltraQuery | 66.3 | 65.6 | 73.2 | 71.1 | 73.2 | 73.4 | – | – | – |
| | AnyCQ | **76.2** | **72.3** | **79.0** | **75.4** | **76.7** | 75.3 | **57.2** | **52.6** | **58.2** |

Table 2: F1-scores on the QAR datasets. $k$ is the number of free variables in the input query.

| Dataset | Model | 3-hub | | | | 4-hub | | | | 5-hub | | | |
|---|---|---|---|---|---|---|---|---|---|---|---|---|---|
| | | $k=1$ | $k=2$ | $k=3$ | **total** | $k=1$ | $k=2$ | $k=3$ | **total** | $k=1$ | $k=2$ | $k=3$ | **total** |
| **FB15k-237-QAR** | SQL | 65.8 | 46.2 | 17.8 | 45.7 | 59.9 | 50.2 | 33.7 | 48.7 | 60.6 | 49.3 | 42.5 | 51.2 |
| | AnyCQ | **67.8** | **62.3** | **50.2** | **60.5** | **60.4** | **54.0** | **48.2** | **54.5** | **63.0** | **56.9** | **43.1** | **54.8** |
| **NELL-QAR** | SQL | 63.5 | 41.3 | 24.0 | 46.7 | 60.6 | 42.1 | 32.9 | 47.7 | 52.7 | 42.5 | 27.6 | 42.8 |
| | AnyCQ | **66.7** | **55.1** | **39.1** | **55.8** | **65.1** | **57.1** | **46.5** | **57.6** | **58.7** | **51.1** | **39.6** | **51.1** |

correct only if the returned assignment is an answer to the input query. In contrast to the CQA evaluation, we do not distinguish between easy and hard answers, since the task of efficiently answering queries with advanced structural complexity, even admitting answers in the observable knowledge graph, is not trivial.

## 6.2 Query answer classification experiments

The results of evaluation on the introduced QAC benchmarks are presented in Table 1. As expected, GNN-QE and UltraQuery outperform ComplEx-based FIT and QTO, with GNN-QE displaying the best scores out of the considered baselines. Equipped with the same NBFNet predictors, AnyCQ matches its performance, achieving only marginally (within 3% relative) lower F1-scores on FB15k-237-QAC, and leading by far on NELL-QAC evaluations. Importantly, AnyCQ successfully extrapolates to formulas beyond the processing power of the existing CQA approaches. On all proposed large query splits AnyCQ consistently outperforms the SQL baseline: SQL classifies *only* easy answers accurately, mapping all the hard answers to false, and as a result falls behind AnyCQ.

## 6.3 Query answer retrieval experiments

We present the QAR evaluation results across all splits of the two proposed datasets consisting of large formulas with multiple free variables in Table 2. Compared with the SQL engine which can only extract easy answers, AnyCQ can reliably achieve similar performance on easy answers while also extracting hard answers over these large formulas, thus outperforming the standard SQL engine in almost all metrics. Importantly, as the number of free variables $k$ increases, the performance gap between AnyCQ and SQL becomes more pronounced. This improvement is due to AnyCQ's computation graphs that effectively handle the rising query complexity without sacrificing performance, substantially gaining over traditional solvers.

**Easy and hard recall on QAR.** To further interpret the results on our query answer retrieval benchmarks, we analyze the recall metric, distinguishing between easy and hard samples (Table 3). As anticipated, the performance of the SQL engine degrades as the number of free variables in the input query increases. While

Table 3: Recall on the easy and hard samples from the QAR datasets. $k$ is the number of free variables.

| Dataset | Model | 3-hub | | | | 4-hub | | | | 5-hub | | | |
|---|---|---|---|---|---|---|---|---|---|---|---|---|---|
| | | $k=1$ | $k=2$ | $k=3$ | total | $k=1$ | $k=2$ | $k=3$ | total | $k=1$ | $k=2$ | $k=3$ | total |
| | | | | | | *Easy instances* | | | | | | | |
| **FB15k-237-QAR** | SQL | **88.7** | 61.2 | 26.4 | 62.8 | **87.2** | **71.7** | 52.6 | 71.9 | **85.3** | 63.6 | **61.4** | **70.5** |
| | AnyCQ | 83.3 | **84.2** | **71.6** | **80.5** | 79.1 | 71.1 | **72.1** | **74.3** | 74.0 | **67.0** | 52.3 | 65.0 |
| **NELL-QAR** | SQL | **94.5** | 69.0 | 55.4 | 79.6 | **90.1** | 69.0 | 63.4 | 77.9 | 88.8 | 78.6 | 64.0 | 80.2 |
| | AnyCQ | 93.0 | **89.4** | **86.5** | **90.7** | 89.6 | **88.8** | **79.6** | **87.1** | **89.4** | **84.5** | **81.3** | **86.1** |
| | | | | | | *Hard instances* | | | | | | | |
| **FB15k-237-QAR** | SQL | 0 | 0 | 0 | 0 | 0 | 0 | 0 | 0 | 0 | 0 | 0 | 0 |
| | AnyCQ | **11.7** | **7.8** | **11.1** | **10.2** | **8.8** | **7.0** | **6.5** | **7.4** | **16.8** | **10.8** | **8.0** | **11.7** |
| **NELL-QAR** | SQL | 0 | 0 | 0 | 0 | 0 | 0 | 0 | 0 | 0 | 0 | 0 | 0 |
| | AnyCQ | **7.0** | **7.0** | **4.0** | **5.9** | **9.7** | **9.2** | **8.2** | **9.0** | **9.2** | **8.1** | **5.8** | **7.7** |

Table 4: F1-scores of AnyCQ models applied outside the training knowledge graph domain.

| AnyCQ **specification** | | FB15k-237-QAR | | | NELL-QAR | | |
|---|---|---|---|---|---|---|---|
| **Predictor type** | **Training dataset** | **3-hub** | **4-hub** | **5-hub** | **3-hub** | **4-hub** | **5-hub** |
| NBFNet-based, | FB15k-237 | 60.5 | 54.5 | 54.8 | 55.9 | 58.7 | 49.4 |
| pre-trained on $G$ | NELL | 58.8 | 53.5 | 52.6 | 55.8 | 57.6 | 51.1 |
| perfect $\tilde{\pi}$ for $\tilde{G}$ | FB15k-237 | 94.4 | 93.4 | 93.0 | 95.5 | 96.4 | 96.2 |
| | NELL | 92.2 | 90.4 | 90.2 | 94.5 | 95.7 | 94.9 |

a similar behavior can be witnessed for AnyCQ models, it progresses at a much slower rate, and as a consequence, AnyCQ consistently outperforms SQL on nearly all splits involving more than one free variable. Additionally, for unary queries, AnyCQ's recall remains within 15% relative to SQL, retrieving answers for over 70% of trivial queries across most splits.

As shown in Table 3, AnyCQ is capable of finding non-trivial answers, even to complicated queries. In contrast, the SQL engine, as a classical approach, cannot retrieve answers to queries that lack a correct solution in the observable knowledge graph. AnyCQ, leveraging a link predictor, demonstrates the ability to retrieve unobserved yet correct answers, even for large, structurally complex queries with multiple free variables.

### 6.4 Ablation studies

**How does AnyCQ perform outside the training domain?** As mentioned in Section 5.4, we expect the search engine to exhibit similar behavior on processed instances, regardless of the underlying knowledge graph. We validate this claim by applying AnyCQ models trained on FB15k-237 or on NELL to both datasets, equipping a relevant link predictor. The results on our QAR benchmarks are presented in Table 4. Notably, the differences between models' accuracies in QAR are marginal, confirming that the resulting search engine is versatile and not dataset-dependent. In fact, the model trained on FB15k-237 exhibits better performance on *both* datasets, further aligning with our assumption on the transferability and generalizability of AnyCQ.

**How does AnyCQ perform with a prefect link predictor?** The AnyCQ framework's performance heavily depends on the underlying link prediction model, responsible for guiding the search and determining the satisfiability of generated assignments. Hence, to assess purely the quality of our search engines, we equipped them with perfect link predictors for the test KGs, eliminating the impact of predictors' imperfections. The results of experiments on our QAR and QAC benchmarks are available in Table 4 and Table 5 , respectively. Remarkably, the simple query types in QAC pose no challenge for AnyCQ, which achieves 100% F1-score on all of them. Furthermore, on large formula splits, the F1-score remains over 90%, displaying the accuracy of

Table 5: F1-scores of ANYCQ model equipped with a perfect link predictor on the QAC task.

| Dataset | 2p | 3p | pi | ip | inp | pin | 3-hub | 4-hub | 5-hub |
|---------|-----|------|-----|-----|-----|-----|-------|-------|-------|
| **FB15k-237-QAC** | 100 | 99.9 | 100 | 100 | 100 | 100 | 92.4 | 91.4 | 93.8 |
| **NELL-QAC** | 100 | 100 | 100 | 100 | 100 | 100 | 93.0 | 89.4 | 91.3 |

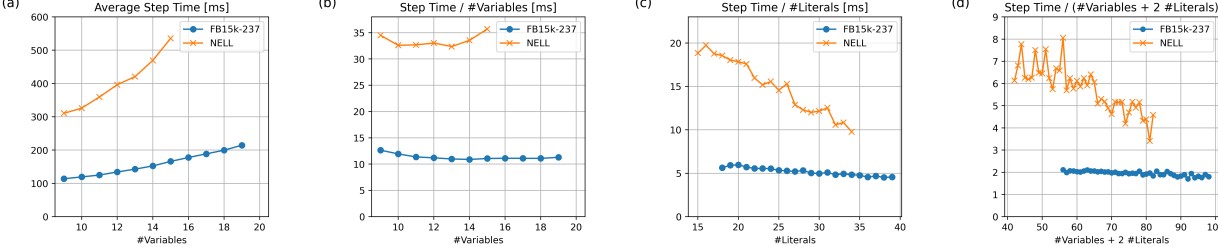

Figure 5: ANYCQ search step time analysis for queries of different complexities: a) average step time (AST) per the number of variables $|\vec{y}|$, b) AST divided by the number of variables $|\vec{y}|$, c) AST divided by the number of literals $|Q|$, d) AST divided by $|\vec{y}| + 2|Q|$, the complexity factor indicated by the theoretical analysis.

our search framework. Similarly, for the task of QAR, ANYCQ with a perfect link predictor achieved over 90% F1-score, establishing the engine's excellent ability to retrieve answers to structurally complex questions.

**How does AnyCQ's scale when the query size increases?** Let $Q = \exists \vec{y}.\Phi(\vec{y})$ be a conjunctive Boolean query over a knowledge graph $G$. Denote by $|Q|$ the number of literals in $Q$, let $h$ be the maximum arity of a literal in $Q$, and let $\vec{c} = (c_0, ..., c_s)$ be the mentioned constants. Then, the corresponding computational graph contains $(|V(G)| + 1) \cdot |\vec{y}| + 2 \cdot |\vec{c}| + |Q|$ vertices and at most $|V(G)| \cdot (|\vec{y}| + h \cdot |Q|) + h \cdot |\vec{c}| \cdot |Q|$ edges. Since ANYCQ processes this graph in linear time, the computational complexity of a single search step is:

$$O\left(|V(G)| \cdot (|\vec{y}| + h \cdot |Q|) + h \cdot |\vec{c}| \cdot |Q|\right)$$

Importantly, this complexity does not depend on the structure of the query graph of $Q$ and scales only linearly with the sizes of the input formula and the KG $G$. We validate this linearity, evaluating the average ANYCQ search step time for queries of different sizes from the '3hub' splits. The results, presented in Figure 5, indicate that the empirical performance matches the theoretical analysis. In particular, Figure 5 b) and c) show that the processing time, divided by the number of variables $|\vec{y}|$ or the number of literals $|Q|$ in the input query, respectively, does not grow as the size of the query increases. We even notice a slight decreasing trend, which we attribute to efficient GPU accelerations. The difference between step times on FB15k-237-QAR and NELL-QAR remains consistent with the relative sizes of the underlying knowledge graphs. A further analysis of processing time, including a comparison with the SQL engine, is provided in Appendix E.

## 7 Summary, limitations, and outlook

In this work, we devise and study two new tasks from the query answering domain: query answer classification and query answer retrieval. Our formulations target the challenge of classifying and generating answers to structurally complex formulas with an arbitrary number of free variables. Moreover, we introduce datasets consisting of instances beyond the processing capabilities of existing approaches, creating strong benchmarks for years to come. To address this demanding setting, we introduce ANYCQ, a framework applicable for scoring and generating answers for large conjunctive formulas with arbitrary arity over incomplete knowledge graphs. We demonstrate the effectiveness over our QAC and QAR benchmarks, showing that on simple samples, ANYCQ matches the performance of state-of-the-art CQA models, while setting challenging baselines for the large instance splits. One potential limitation is considering by default the input query in disjunctive normal form, converting to which may require exponentially many operations. We hope our work will motivate the field of query answering to recognize the classification nature of the induced tasks and expand the scope of CQA to previously intractable cases.

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

# A   AnyCQ details

## A.1   Architecture

AnyCQ's architecture is based on the original ANYCSP Tönshoff et al. (2023) framework. The trainable components of the AnyCQ GNN model $\theta$ are:

- a GRU Chung et al. (2014) cell $\mathbf{G} : \mathbb{R}^d \times \mathbb{R}^d \to \mathbb{R}^d$ with a trainable initial state $\mathbf{h} \in \mathbb{R}^d$

- a Multi Layer Perceptron (MLP) value encoder $\mathbf{E} : \mathbb{R}^{d+1} \to \mathbb{R}^d$

- two MLPs $\mathbf{M}_V, \mathbf{M}_R : \mathbb{R}^d \to \mathbb{R}^{4d}$ sending information between value and literal vertices

- three MLPs $\mathbf{U}_V, \mathbf{U}_R, \mathbf{U}_X : \mathbb{R}^d \to \mathbb{R}^d$ aggregating value, literal and variable messages

- an MLP $\mathbf{O} : \mathbb{R}^d \to \mathbb{R}$ that generates logit scores for all variable nodes.

We denote the set of neighbors of term and literal nodes by $\mathcal{N}(\cdot)$. In the case of value nodes, we distinguish between the corresponding term node and the set of connected literal vertices, which we represent by $\mathcal{N}_R(\cdot)$.

The model starts by sampling an initial assignment $\alpha^{(0)}$, where the value of each variable is chosen uniformly at random from $V(G)$, and proceeds for $T$ search steps. In step $t$:

- If $t = 1$, initialize the hidden state of each value node to be $\mathbf{h}^{(0)}(v_{z \to a}) = \mathbf{h}$.

- Generate light edge labels under the assignment $\alpha^{(t-1)}$ for all value-literal edges. Precisely, let $v_{\psi_i}$ be a literal node corresponding to an atomic formula $\psi$ and $v_{z \to a}$ be a connected value node. The light edge label $L_E^{(t-1)} (v_{\psi_i}, v_{z \to a}; \alpha)$ is a binary answer to the question: "Is $\psi$ satisfied under $\left[ \alpha^{(t-1)} \right]_{z \to a}$?" with respect to the equipped predictor.

- For each value node $v_{z \to a}$, generate its new latent state

$$\mathbf{x}^{(t)}(v_{z \to a}) = \mathbf{E} \left( \left[ \mathbf{h}^{(t-1)}(v_{z \to a}), \delta_{\alpha(x)=v} \right] \right)$$

where $[\cdot, \cdot]$ denotes concatenation and $\delta_C = 1$ if the condition $C$ holds, and 0 otherwise.

- Derive the messages to be sent to the constraint nodes:

$$\mathbf{m}^{(t)}(v_{z \to a}, 0), ..., \mathbf{m}^{(t)}(v_{z \to a}, 3) = \mathbf{M}_V \left( \mathbf{x}^{(t)}(v_{z \to a}) \right)$$

- For each literal node $v_\psi$, gather the messages from its value neighbors, considering the light and potential labels:

$$\mathbf{y}^{(t)}(v_\psi) = \bigoplus_{v_{z \to a} \in \mathcal{N}(v_\psi)} \mathbf{m}^{(t)} \left( v_{z \to a}, 2 \cdot P_E(v_\psi, v_{z \to a}) + L_E^{(t-1)}(v_\psi, v_{z \to a}; \alpha) \right)$$

where $\bigoplus$ denotes element-wise max.

- The messages to be sent to the value nodes are then evaluated as:

$$\mathbf{m}^{(t)}(v_\psi, 0), ..., \mathbf{m}^{(t)}(v_\psi, 3) = \mathbf{M}_R \left( \mathbf{y}^{(t)}(v_\psi) \right)$$

- Aggregate the messages in each value node $v_{z \to a}$:

$$\mathbf{y}^{(t)}(v_{z \to a}) = \bigoplus_{v_\psi \in \mathcal{N}_{\mathcal{R}}(v_{z \to a})} \mathbf{m}^{(t)} \left( v_{z \to a}, 2 \cdot P_E(v_\psi, v_{z \to a}) + L_E^{(t-1)}(v_\psi, v_{z \to a}; \alpha) \right)$$

and integrate them with current hidden state:

$$\mathbf{z}^{(t)}(v_{z \to a}) = \mathbf{U}_V \left( \mathbf{x}^{(t)}(v_{z \to a}) + \mathbf{y}^{(t)}(v_{z \to a}) \right) + \mathbf{x}^{(t)}(v_{z \to a})$$

- For each term node $v_z$, aggregate the states of the corresponding value nodes:

$$\mathbf{z}^{(t)}(v_z) = \mathbf{U}_X \left( \bigoplus_{v_{z \to a} \in \mathcal{N}(v_z)} \mathbf{z}^{(t)}(v_{z \to a}) \right)$$

- For each value node $v_{z \to a}$, update its hidden state as:

$$\mathbf{h}^{(t)}(v_{z \to a}) = \mathbf{G} \left( \mathbf{h}^{(t-1)}(v_{z \to a}), \mathbf{z}^{(t)}(v_{z \to a}) + \mathbf{z}^t(v_z) \right)$$

- Generate logits and apply softmax within each domain:

$$\mathbf{o}^{(t)}_{z \to a} = \text{clip} \left( \mathbf{O} \left( \mathbf{h}^{(t)}(v_{z \to a}) \right) - \max_{a \in \mathcal{D}(z)} \mathbf{O} \left( \mathbf{h}^{(t)}(v_{z \to a}) \right), [-100, 0] \right)$$

$$\mu^{(t)}(v_{z \to a}) = \frac{\exp \mathbf{o}^{(t)}_{z \to a}}{\sum_{a' \in \mathcal{D}(z)} \exp \mathbf{o}^{(t)}_{z \to a'}}$$

- Sample the next assignment $\alpha^{(t)}$, selecting the next value independently for each variable $x$, with probabilities $\mathbb{P}\left(\alpha^{(t)}(x) = a\right) = \mu^{(t)}(v_{x \to a})$ for all $a \in \mathcal{D}(x)$.

Note that the suggested methodology for evaluating probabilities $\mathbb{P}\left(\alpha^{(t)}(x) = a\right)$ is approximately equivalent to applying softmax directly on $\mathbf{O}\left(\mathbf{h}^{(t)}(v_{x \to a})\right)$. However, applying this augmentation, we are guaranteed that for any variable $x$ and a relevant value $a \in \mathcal{D}(x)$:

$$\mathbb{P}\left(\alpha^{(t)}(x) = a\right) = \frac{\exp \mathbf{o}^{(t)}_{x \to a}}{\sum_{a' \in \mathcal{D}(x)} \exp \mathbf{o}^{(t)}_{x \to a'}} \geq \frac{e^{-100}}{|\mathcal{D}(x)|} \geq \frac{1}{e^{100}|V(G)|}.$$

## A.2 Training methodology

Suppose we are given a training query $Q(x) = \exists \vec{y}.\Phi(x, \vec{y})$. We run $\Theta$ on $\exists x.Q(x)$ for $T_{\text{train}}$ search steps, recovering the assignments $\alpha^{(0)}, ..., \alpha^{(T_{\text{train}})}$ and the intermediate value probability distributions:

$$\mu^{(1)} = \left\{ \mu^{(1)}_z | z \in \{\vec{x}, \vec{y}\} \right\}, \ \dots \ , \mu^{(T_{\text{train}})} = \left\{ \mu^{(T_{\text{train}})}_z | z \in \{\vec{x}, \vec{y}\} \right\}$$

The reward $R^{(t)}$ for step $1 \leq t \leq T$ is calculated as the difference between the score for assignment $\alpha^{(t)}$ and the best assignment visited so far:

$$R^{(t)} = \max \left( 0, S^{(t)} - \max_{t' < t} S^{(t')} \right)$$

where $S^{(t)} = S_{\pi_{\text{train}}}\left(\Phi(\alpha^{(t)}(x)/x, \alpha^{(t)}(\vec{y})/\vec{y})\right)$. Additionally, the transition probability

$$P^{(t)} = \mathbb{P}\left(\alpha^{(t)}|\mu^{(t)}\right) = \prod_{z \in \{\vec{x}, \vec{y}\}} \mu^{(t)}_z \left(\alpha^{(t)}(z)\right)$$

represents the chance of drawing assignment $\alpha^{(t)}$ at step $t$, given distributions $\left\{\mu^{(t)}_z | z \in \{\vec{x}, \vec{y}\}\right\}$. The corresponding REINFORCE's training loss is evaluated as a weighted sum of rewards generated during $T_{\text{train}}$ search steps and the model weights are then updated using the gradient descend equation:

$$\theta \leftarrow \theta - \alpha \cdot \nabla_\theta \left( - \sum_{i=0}^{T_{\text{train}}-1} \gamma^i \left( \left(\log P^{(t)}\right) \cdot \sum_{t=i+1}^{T_{\text{train}}} \left(\gamma^{t-i-1} R^{(t)}\right) \right) \right)$$

where $\gamma \in (0, 1]$ is a discount factor and $\alpha \in \mathbb{R}$ is the learning rate.

For the training data, we use the training splits of the existing FB15k-237 and NELL CQA datasets Ren and Leskovec (2020), consisting of queries of types: '1p', '2p', '3p', '2i', '3i', '2in', '3in', 'pin', 'inp' (see Table 6 for the corresponding first-order logic formulas). Hence, during training, AnyCQ witnesses queries with projections, intersections and negations, learning principles of this logical structures. However, all of these queries mention at most 3 free variables, remaining limited in size.

Table 6: Simple query types

| Split | Formula |
|-------|---------|
| 1p | $Q(x_1) = r_1(x, c_1)$ |
| 2p | $Q(x_1) = \exists y_1. r_1(x_1, y_1) \wedge r_2(y_1, c_1)$ |
| 3p | $Q(x_1) = \exists y_1, y_2. r_1(x_1, y_1) \wedge r_2(y_1, y_2) \wedge r_3(y_2, c_1)$ |
| 2i | $Q(x_1) = r_1(x, c_1) \wedge r_2(x, c_2)$ |
| 3i | $Q(x_1) = r_1(x, c_1) \wedge r_2(x, c_2) \wedge r_3(x, c_3)$ |
| pi | $Q(x_1) = \exists y_1. r_1(x_1, y_1) \wedge r_2(y_1, c_1) \wedge r_3(x_1, c_2)$ |
| ip | $Q(x_1) = \exists y_1. r_1(x_1, y_1) \wedge r_2(y_1, a_1) \wedge r_3(y_1, a_2)$ |
| 2i | $Q(x_1) = r_1(x, c_1) \wedge \neg r_2(x, c_2)$ |
| 3i | $Q(x_1) = r_1(x, c_1) \wedge r_2(x, c_2) \wedge \neg r_3(x, c_3)$ |
| inp | $Q(x_1) = \exists y_1. r_1(x_1, y_1) \wedge r_2(y_1, c_1) \wedge \neg r_3(y_1, c_2)$ |
| pin | $Q(x_1) = \exists y_1. r_1(x_1, y_1) \wedge r_2(y_1, c_1) \wedge \neg r_3(x_1, c_2)$ |

## A.3 Hyperparameters and implementation

**Architecture.** We choose the hidden embedding size $d = 128$ in the ANYCQ architecture for all experiments. All MLPs used in our model consist of two fully connected layers with ReLU Agarap (2018) activation function. The intermediate dimension of the hidden layer is chosen to be 128.

**Training.** The REINFORCE Williams (2004) discount factor $\lambda$ is set to 0.75 for both datasets, following the best configurations in ANYCSP experiments. During training, we run our models for $T_{\text{train}} = 15$ steps. The batch size is set to 4 for FB15k-237 and 1 for NELL, due to the GPU memory constraints. All models are trained with an Adam Kingma and Ba (2015) optimizer with learning rate $5 \cdot 10^{-6}$ on a single NVIDIA Tesla V100 SXM2 with 32GB VRAM. We let the training run for 4 days, which translates to 500,000 batches on FB15k-237 and 200,000 batches for NELL, and choose the final model for testing.

**Inference.** To run all experiments, we use an Intel Xenon Gold 6326 processor with 128GB RAM, and an NVIDIA A10 graphics card with 24GB VRAM.

## A.4 Scope of formulas

Importantly, our method is not limited to conjunctive formulas. Suppose we are given a Boolean formula $\varphi = \exists \vec{y}. \Psi(\vec{y})$ where $\Psi(\vec{y})$ is quantifier-free and in disjunctive normal form (DNF), so that $\Psi(\vec{y}) = C_1 \vee ... \vee C_n$ where each $C_i$ is a conjunction of literals. Then:

$$\varphi \equiv (\exists \vec{y}. C_1) \vee ... \vee (\exists \vec{y}. C_n)$$

which can be processed by ANYCQ by independently solving each $(\exists \vec{y}. C_i)$ and aggregating the results. Moreover, the ability of our model to handle higher arity relations enables efficient satisfiability evaluation for existential formulas in the conjunctive normal form. Let $\psi = \exists \vec{y}. (D_1 \wedge ... \wedge D_n)$ where each $D_i$ is a disjunction of literals. Consider $D_i = l_{i,1} \vee ... \vee l_{i,m}$ and let $\vec{z}_i = \text{Var}(D_i)$. We can view the disjunctive clause $D_i$ as a single relation $D_i(\vec{z}_i)$ evaluating to

$$S_{\pi,G}(D_i(\alpha(\vec{z}_i)/\vec{z}_i)) = \max_j S_{\pi,G}(l_{i,j}(\alpha(\text{Var}(l_{i,j}))/\text{Var}(l_{i,j})))$$

Under this transformation, $\psi = \exists \vec{y}. (D_1(\vec{z}_1) \wedge ... \wedge D_n(\vec{z}_n))$ becomes a conjunctive query, hence processable by ANYCQ. Up to our knowledge, we present the first query answering approach efficiently scoring arbitrary CNF Boolean queries over incomplete knowledge graphs.

### A.5 Expressivity

Standard graph neural networks are known to have limited expressive power Xu et al. (2019), e.g. MPNNs cannot produce different outputs for graphs not distinguishable by the Weisfeiler-Lehman algorithm Leman and Weisfeiler (1968). We argue that ANYCQ does not suffer from this limitation. It has been noticed that including randomness in GNN models increases their expressiveness Abboud et al. (2021). In our case, for any Boolean conjunctive query $Q = \exists \vec{y} \Phi(\vec{y})$ over a knowledge graph $G$ and a relevant link predictor $\pi$, for any assignment $\alpha : \{\vec{y}\} \to V(G)$, there is a non-zero probability of $\alpha$ being selected at some point of the search (see Appendix A.1). Hence, any ANYCQ model has a chance of correctly predicting $S_{\pi,G}(Q)$, making it fully expressive for the tasks of QAC and QAR.

### A.6 Fuzzy logic

Fuzzy logic has been widely adopted in the CQA literature as a way to evaluate the satisfiability of logical formulas in a continuous, differentiable manner. It underpins several prominent methods, including CQD (Arakelyan et al., 2020), GNN-QE (Zhu et al., 2022), and QTO (Bai et al., 2023), due to its modularity and interpretability. However, especially when applied to large and structurally complex queries, fuzzy logic introduces several limitations that should be taken into account.

**Score vanishing.** Consider a conjunction of 10 literals, each scored at 0.9 by the link predictor. When using the product $t$-norm, the formula score becomes $0.9^{10} \approx 0.35$, despite all individual facts being highly probable. This effect becomes more pronounced in long formulas, leading to overly conservative judgments. To mitigate this, we adopt the Gödel $t$-norm (min operator), which in the same scenario would return a more stable score of 0.9. Additionally, using the Gödel t-norm with a 0.5 threshold is equivalent to applying propositional logic over binarized literal scores, making it well-suited for our classification-based objectives.

**Gradient instability in supervised learning.** As discussed by Van Krieken et al. (2022), another issue with fuzzy logic arises in differentiable learning settings, where gradients must propagate through the query structure and the fuzzy connectives. This can lead to vanishing or unstable gradients, especially for large or cyclic queries. In our case, however, this problem is largely avoided: ANYCQ is trained using reinforcement learning, where the fuzzy logic score is used as a scalar reward signal and not differentiated through. During training, we apply REINFORCE, which treats the Boolean score as an external reward, and during inference, fuzzy logic is only used to rank complete assignments. As such, our framework sidesteps the gradient-related challenges described in Van Krieken et al. (2022), while retaining the benefits of fuzzy logic for scoring.

### A.7 Edge labels

To effectively navigate the space of variable assignments, our framework augments the computational graph $G_{Q,\alpha}$ with edge labels that encode information from the link predictor $\pi$. These edge labels play a critical role in guiding the search process by helping the model answer two fundamental questions:

- Which assignments to variables are worth considering? (exploration)

- How should the current assignment be changed to satisfy more literals? (exploitation).

To this end, we define two types of edge labels on the graph edges connecting literal vertices $v_{\psi_i}$ with value vertices $v_{e \to a}$: potential edge (PE) labels and light edge (LE) labels. PE labels are used to identify whether a particular substitution could lead to a satisfying assignment and are independent of the current state. They support *exploration* by indicating globally promising directions in the search space and can be seen as a way to constrain the search to regions of high potential. In contrast, LE labels are assignment-dependent and indicate whether a local change - modifying a single variable's value, would make a particular literal true. They enable exploitation by directing the model toward refinements of the current assignment that increase the satisfiability of the formula. We describe each type of label in detail in the following subsections.

### A.7.1 LE Labels: Guiding Local Improvements

Light edge (LE) labels were originally introduced in the ANYCSP (Tönshoff et al., 2023) framework as the primary mechanism for guiding discrete search. In the context of query answering, their purpose is to identify marginal changes to the current variable assignment that are likely to increase the number of satisfied literals in the query. That is, given an assignment $\alpha$, LE labels help determine which single-variable substitutions are most promising for improving the current solution. This makes them particularly useful during local exploitation, where the goal is to refine an existing candidate assignment rather than explore the full space.

**Formal definition.** Let $Q = \exists \vec{y}.\Phi(\vec{y})$ be an existential Boolean conjunctive query, and let $\psi_i$ be a literal in $Q$, mentioning terms $\vec{z}$. Let $\alpha$ be the current assignment to the variables of $Q$, and let $e \in \vec{z}$ be a term in $\psi_i$. For a candidate entity $a \in \mathcal{D}(e)$ (recall that $\mathcal{D}(e) = V(G)$ for variables and $\mathcal{D}(e) = \{e\}$ for constants), the LE label on the edge between the literal vertex $v_{\psi_i}$ and the value vertex $v_{e \to a}$ is defined as follows:

$$L_E(v_{\psi_i}, v_{z,a}; \alpha) = \begin{cases} 1 & \text{if } S_{\pi,G}(\psi_i(\alpha_{z \to a}(\vec{z})/\vec{z})) \geq 0.5 \\ 0 & \text{otherwise} \end{cases}$$

This definition reflects whether updating the current assignment $\alpha$ by changing only the value of $e$ to $a$ (keeping all other variable assignments fixed) is sufficient to make the literal $\psi_i$ true.

**Example.** Suppose the query is:

$$Q = \exists y_1, y_2. r(a, y_1) \wedge s(y_1, y_2)$$

with the current assignment $\alpha = \{y_1 \to a_1, y_2 \to a_2\}$, and we focus on the literal $\psi_2 = s(y_1, y_2)$. Let's consider a marginal update to the variable $y_2$, and let $a_2' \in \mathcal{D}(y_2)$. To determine the LE label $L_E(v_{\psi_2}, v_{y_2 \to a_2'}; \alpha)$, we check whether $s(a_1, a_2')$ holds in the (predicted) KG $\tilde{G}$. If it does, then this local update would satisfy $\psi_2$, and the label is set to 1. Otherwise, the label is 0. This allows the model to reason about whether changing $y_2$ to $a_2'$ would improve the current assignment in terms of satisfying more of the query structure.

### A.7.2 PE Labels: Prioritizing Promising Assignments

Potential edge (PE) labels are introduced in this work as an extension to the ANYCSP framework, specifically to address the challenges posed by the large domain sizes in modern knowledge graphs. While LE labels guide the refinement of a given assignment, PE labels serve a complementary purpose: they help identify which candidate variable assignments are worth considering in the first place. In other words, PE labels support exploration by informing the model which edges in the computational graph represent substitutions that are likely to participate in some satisfying assignment, independent of the current state.

**Formal definition.** Formally, let $Q = \exists \vec{y}.\Phi(\vec{y})$ be a conjunctive Boolean query, let $\psi_i \in \Phi$ be a literal mentioning terms $\vec{z}$, and let $e \in \vec{z}$. Then, for every $a \in \mathcal{D}(e)$, the PE label on the edge between $v_{\psi_i}$ and $v_{e \to a}$ is defined as follows:

$$P_E(v_{\psi_i}, v_{z,a}) = \begin{cases} 1 & \text{if } \exists \alpha. \, (\alpha(e) = a \wedge S_{\pi,G}(\psi_i(\alpha(\vec{z})/\vec{z})) \geq 0.5) \\ 0 & \text{otherwise} \end{cases}$$

Intuitively, the label is set to 1 if there exists any full assignment to the variables of $\psi_i$ such that $\psi_i$ becomes true when $e$ is set to $a$. Importantly, this is evaluated without reference to the current partial assignment $\alpha$, making PE labels suitable for filtering the search space early in the computation.

**Example.** Consider the same example query as before:

$$Q = \exists y_1, y_2. r(a, y_1) \wedge s(y_1, y_2)$$

and the literal $\psi_2 = s(y_1, y_2)$. Let $a_2 \in \mathcal{D}(y_2)$ be a viable assignment to $y_2$. To evaluate the PE label $P_E(v_{\psi_2}, v_{y_2 \to a_2})$, we check whether $\exists y_1.s(y_1, a_2)$ is satisfied, i.e. whether there exists some $a_1$ such that the literal $s(a_1, a_2)$ is true, according to the link predictor $\pi$. If such $a_1$ exists, we set the label to 1, and otherwise - to 0. This allows the GNN to prioritize reasoning about value assignments that could plausibly contribute to satisfying the query, rather than wasting capacity on highly unlikely candidates.

Table 7: F1-scores of AnyCQ models with and without PE labels.

| PE labels | FB15k-237-QAR | | | NELL-QAR | | |
|:---:|:---:|:---:|:---:|:---:|:---:|:---:|
| | 3-hub | 4-hub | 5-hub | 3-hub | 4-hub | 5-hub |
| ✓ | 56.3 | 52.7 | 54.1 | 51.4 | 53.0 | 48.4 |
| ✗ | 0.0 | 0.0 | 0.0 | 0.0 | 0.0 | 0.0 |

**Importance of PE labels.**  We empirically validate the significance of this modification on the proposed QAR benchmark. To this end, we train an AnyCQ model from scratch, disabling the signal from PE labels by setting all of them to 0 throughout the training and inference. The results, shown in Table 7, demonstrate that without access to PE labels, AnyCQ fails to generalize to queries of large size and is unable to produce a correct answer, even for a single sample.

**PE label generation.**  Given the critical role of this modification in our framework, it is essential to address the efficient generation of PE labels. In this work, we pre-compute PE labels for both datasets, aligning them precisely with the definitions, with respect to the selected test link predictors. However, this process can become computationally expensive, potentially requiring hours, and becoming highly inefficient, particularly in scenarios where the link predictor frequently changes, e.g. during validation. To mitigate this inefficiency, we propose alternative methodologies to approximate true PE labels, enabling faster cold-start inference.

Our main alternative bases on the closed world assumption (CWA) Libkin and Sirangelo (2009), which restricts the set of entities that should be considered for prediction of unobserved facts. Formally, let $G$ be an observable knowledge graph and let $\tilde{G}$ be its completion. Then, for any $r \in R(G)$ and any $a, b \in V(G)$:

$$\tilde{G} \models r(a, b) \implies \exists b' \in V(G) \,.\, G \models r(a, b')$$

$$\tilde{G} \models r(a, b) \implies \exists a' \in V(G) \,.\, G \models r(a', b)$$

With this assumption, the set of pairs for which $\tilde{G} \models r(a, b)$ holds becomes limited. Indeed, $a$ needs to be a head of an observable relation $r(a, b')$ and analogously, $b$ needs to be a tail of an observable $r(a', b)$. Therefore, the induced approximation of PE labels:

$$\hat{P}_E(v_{r(x,y)}, v_{x,a}) = \begin{cases} 1 & \text{if } \exists b' \in \mathcal{D}(y).G \models r(a, b') \\ 0 & \text{otherwise} \end{cases}$$

$$\hat{P}_E(v_{r(x,y)}, v_{y,b}) = \begin{cases} 1 & \text{if } \exists a' \in \mathcal{D}(x).G \models r(a', b) \\ 0 & \text{otherwise} \end{cases}$$

can be efficiently derived in time $O(|E(G)|)$. We use this modification during the validation process to avoid the necessity of computing the precise PE labels.

An alternative approach, not explored in this work, involves incorporating domain-specific information about the underlying knowledge graph. For instance, if the relation in a given query is *fatherOf*, both entities are likely to be humans. By labeling all entities in $V(G)$ with relevant tags, such information could be extracted, and objects classified as 'people' could be assigned a corresponding PE label of 1. While we prioritize generalizability and do not pursue this direction, we recognize its potential, particularly for sparse knowledge graphs where CWA-derived PE labels may be too restrictive.

**PE labels versus domain restriction.**  An alternative to relying on an additional set of labels to prevent the search from accessing unreasonable assignments could be restricting the domains $\mathcal{D}(y)$ of the considered variables. In the current formulation, each variable $y$ mentioned in the input Boolean query $Q$ is assigned a domain $\mathcal{D}(y) = V(G)$. Reducing the considered domains can significantly shrink the computational graph, leading to faster computation. Such a solution would be specifically beneficial when operating on large knowledge graphs, and even essential for applications to milion-scale KGs, such as Wikidata-5M (Wang et al.,

2021). While this approach improves inference efficiency, improper application can render correct answers unreachable due to excessively restrictive domain reductions. Consequently, we leave further exploration of this direction for future work.

## B    Proofs

**Theorem 5.1.** *Let $Q = \exists \vec{y}.\Phi(\vec{y})$ be a conjunctive Boolean query and let $\Theta$ be any ANYCQ model equipped with a predictor $\pi$. For any execution of $\Theta$ on $Q$, running for $T$ steps:*

$$\mathbb{P}\left(\Theta(Q|G, \pi) = S_{\pi,G}(Q)\right) \to 1 \qquad as \qquad T \to \infty$$

*Proof.* Let $\Theta$ be an ANYCQ model equipped with a predictor $\pi$ for a knowledge graph $G$. Let $Q = \exists \vec{y}.\Phi(\vec{y})$ be a conjunctive Boolean query with $\vec{y} = y_1, ..., y_k$. Let

$$\alpha_{\max} = \underset{\alpha:\vec{y} \to V(G)}{\arg\max} \ S_{\pi,G}(\Phi(\alpha(\vec{y})/\vec{y}))$$

so that

$$S_{\pi,G}(Q) = S_{\pi,G}(\Phi(\alpha_{\max}(\vec{y})/\vec{y}))$$

Consider an execution of $\Theta$, running for $T$ steps, and let $\alpha^{(0)}, ..., \alpha^{(T)}$ be the generated assignments. Then,

$$\mathbb{P}\left(\Theta(Q|G, \pi) \neq S_{\pi,G}(Q)\right) = \mathbb{P}\left(S_{\pi,G}(Q) \neq S_{\pi,G}\left(\Phi\left(\alpha^{(t)}(\vec{y})/\vec{y}\right)\right) \text{ for all } 0 \leq t \leq T\right)$$

$$\leq \mathbb{P}\left(S_{\pi,G}(Q) \neq S_{\pi,G}\left(\Phi\left(\alpha^{(t)}(\vec{y})/\vec{y}\right)\right) \text{ for all } 1 \leq t \leq T\right)$$

$$\leq \mathbb{P}\left(\alpha^{(t)} \neq \alpha_{\max} \text{ for all } 1 \leq t \leq T\right)$$

By the remark at the end on Appendix A.1:

$$\mathbb{P}\left(\alpha^{(t)}(y) = a\right) \geq \frac{1}{e^{100}|V(G)|} \qquad \forall 1 \leq t \leq T \ \forall a \in V(G) \ \forall y \in \vec{y}$$

In particular:

$$\mathbb{P}\left(\alpha^{(t)}(y) = \alpha_{\max}(y)\right) \geq \frac{1}{e^{100}|V(G)|} \qquad \forall 1 \leq t \leq T \ \forall y \in \vec{y}$$

so as the value for each variable in $\alpha^{(t)}$ is sampled independently:

$$\mathbb{P}\left(\alpha^{(t)} = \alpha_{\max}\right) \geq \left(\frac{1}{e^{100}|V(G)|}\right)^k$$

Therefore:

$$\mathbb{P}\left(\Theta(Q|G, \pi) \neq S_{\pi,G}(Q)\right) \leq \mathbb{P}\left(\alpha^{(t)} \neq \alpha_{\max} \text{ for all } 1 \leq t \leq T\right)$$

$$\leq \left(1 - \left(\frac{1}{e^{100}|V(G)|}\right)^k\right)^T$$

which tends to 0 as $T \to \infty$. $\qquad\qquad\qquad\qquad\qquad\qquad\qquad\qquad\qquad\qquad\qquad\qquad\qquad\quad$ $\square$

**Proposition B.1** (Scores of a Perfect Link Predictor)**.** *Let $Q$ be a quantifier-free Boolean formula over an observable knowledge graph $G$. Then, the score of $Q$ w.r.t. the perfect link predictor $\tilde{\pi}$ for the completion $\tilde{G}$ of $G$ satisfies:*

$$S_{\tilde{\pi},G}(Q) = \begin{cases} 0 & \text{if } \tilde{G} \not\models Q \\ 1 & \text{if } \tilde{G} \models Q \end{cases}$$

*Proof.* The claim follows from the structural induction on the formula $Q$. For the base case, suppose that $Q$ is an atomic formula $r(a,b)$. The result follows trivially from the definition of a perfect link predictor $\tilde{\pi}$. Assume the claim holds for boolean formulas $Q, Q'$. Then:

$$S_{\tilde{\pi},G}(\neg Q) = 1 - S_{\tilde{\pi},G}(Q) = \begin{cases} 1 & \text{if } \tilde{G} \not\models Q \\ 0 & \text{if } \tilde{G} \models Q \end{cases} = \begin{cases} 1 & \text{if } \tilde{G} \models \neg Q \\ 0 & \text{if } \tilde{G} \not\models \neg Q \end{cases}$$

For $(Q \wedge Q')$, note that $\tilde{G} \models (Q \wedge Q') \iff \left((\tilde{G} \models Q) \wedge (\tilde{G} \models Q')\right)$ and hence

$$S_{\tilde{\pi},G}(Q \wedge Q') = \min\left(S_{\tilde{\pi},G}(Q), S_{\tilde{\pi},G}(Q')\right) = \begin{cases} 1 & \text{if } S_{\tilde{\pi},G}(Q) = S_{\tilde{\pi},G}(Q') = 1 \\ 0 & \text{otherwise} \end{cases}$$

$$= \begin{cases} 1 & \text{if } \tilde{G} \models Q \wedge \tilde{G} \models Q' \\ 0 & \text{otherwise} \end{cases}$$

$$= \begin{cases} 1 & \text{if } \tilde{G} \models (Q \wedge Q') \\ 0 & \text{if } \tilde{G} \not\models (Q \wedge Q') \end{cases}$$

Similarly, for $(Q \vee Q')$, since $\tilde{G} \models (Q \vee Q') \iff \left((\tilde{G} \models Q) \vee (\tilde{G} \models Q')\right)$, we can deduce:

$$S_{\tilde{\pi},G}(Q \vee Q') = \max\left(S_{\tilde{\pi},G}(Q), S_{\tilde{\pi},G}(Q')\right) = \begin{cases} 0 & \text{if } S_{\tilde{\pi},G}(Q) = S_{\tilde{\pi},G}(Q') = 0 \\ 1 & \text{otherwise} \end{cases}$$

$$= \begin{cases} 0 & \text{if } \tilde{G} \not\models Q \wedge \tilde{G} \not\models Q' \\ 1 & \text{otherwise} \end{cases}$$

$$= \begin{cases} 0 & \text{if } \tilde{G} \not\models (Q \vee Q') \\ 1 & \text{if } \tilde{G} \models (Q \vee Q') \end{cases}$$

which completes the inductive step. □

**Theorem 5.2.** *Let $Q = \exists \vec{y}.Q(\vec{y})$ be a conjunctive Boolean query over an unobservable knowledge graph $\tilde{G}$ and let $\Theta$ be any ANYCQ model equipped with a perfect link predictor $\tilde{\pi}$ for $\tilde{G}$. If $\Theta(Q|G, \tilde{\pi}) > 0.5$, then $\tilde{G} \models Q$ .*

*Proof.* Consider the setup as in the theorem statement and suppose $\Theta(Q|G, \tilde{\pi}) > 0.5$. Then, there exists an assignment $\alpha : \vec{y} \to V(G)$ (found at some search step) such that

$$S_{\tilde{\pi},G}(\Phi(\alpha(\vec{y})/\vec{y})) = \Theta(Q|G, \tilde{\pi}) > 0.5$$

By Proposition B.1, this implies:

$$S_{\tilde{\pi},G}(\Phi(\alpha(\vec{y})/\vec{y})) = 1 \qquad \text{and} \qquad \tilde{G} \models \Phi(\alpha(\vec{y})/\vec{y})$$

Hence, $\tilde{G} \models \exists \vec{y}.\Phi(\vec{y}) = Q$. □

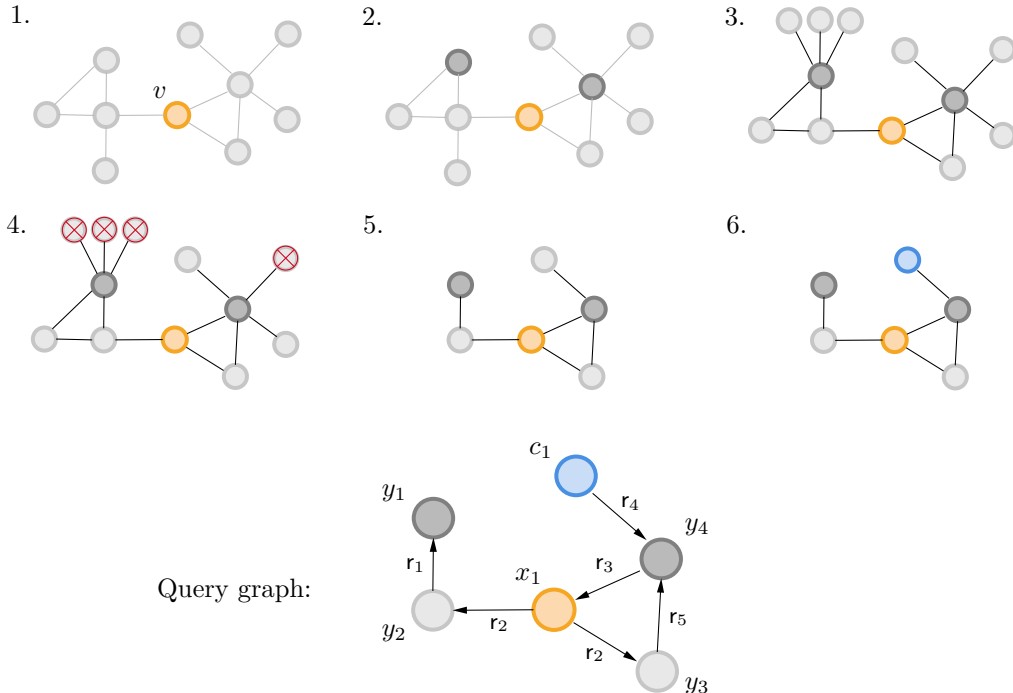

Figure 6: Visualisation of the process of generating large queries for our benchmark datasets, with $n_{hub}$ = 2, $p_{out}$ = 0.5 and $n_{min}$ = 5. The resulting sampled query is: $Q(x_1) = \exists y_1, y_2, y_3, y_4(\mathsf{r}_2(x_1, y_2) \wedge \mathsf{r}_1(y_2, y_1) \wedge \mathsf{r}_2(x_1, y_3) \wedge \mathsf{r}_5(y_3, y_4) \wedge \mathsf{r}_3(y_4, x_1) \wedge \mathsf{r}_4(c_1, y_4))$

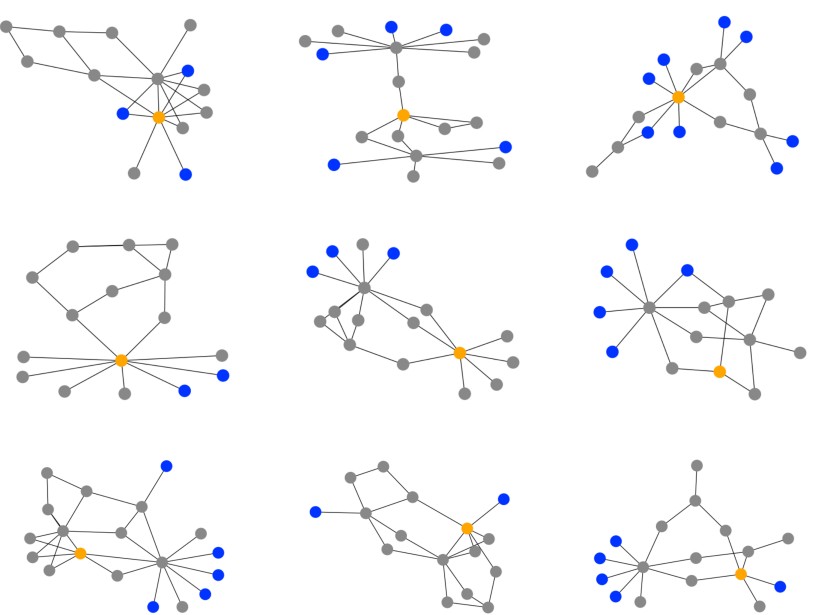

Figure 7: Examples of undirected query graphs of formulas from the FB15k-237-QAR '3-hub' split. Blue nodes represent constant terms, while grey - to the existentially quantified variables. The orange node corresponds to the free variable.

## C Dataset constructions

Benchmark datasets in the existing query answering literature, FB15k-237 Toutanova et al. (2015) and NELL Carlson et al. (2010), comprise formulas with simple structures, thereby impeding the comprehensive evaluation and advancement of methodologies and algorithms. We address this gap by creating new validation and test datasets on top of well-established benchmarks, consisting of queries with complexity exceeding the processing power of known approaches. In particular, we increase the number of variables mentioned in the considered formulas from 3 to between 12 and 20, while imposing structural difficulty by sampling query graphs with multiple cycles, long-distance reasoning steps and multi-way conjunctions.

### C.1 Base large query generation

Each of the considered datasets: FB15k-237 and NELL, provides three knowledge graphs $G_{\text{train}}, G_{\text{val}}, G_{\text{test}}$, for training, validation and testing, respectively, satisfying $E(G_{\text{train}}) \subset E(G_{\text{val}}) \subset E(G_{\text{test}})$. During validation, $G_{\text{train}}$ is treated as the observable graph $G$, while $G_{\text{val}}$ as its completion $\tilde{G}$. Similarly, for testing, $G = G_{\text{val}}$ and $\tilde{G} = G_{\text{test}}$.

We begin the dataset generation by sampling base formulas, to be later converted into instances for the QAC and QAR benchmarks. During sampling, we use four hyperparameters: $n_{\text{hub}}, n_{\text{min}}, p_{\text{const}}$ and $p_{\text{out}}$, whose different values contribute to creating different benchmark splits. The process is visualized in Figure 6. A single base query is sampled as follows:

1. A vertex $v \in V(G)$ is sampled uniformly at random from $V(G)$.

2. Let $\mathcal{N}_i(v)$ be the set of nodes whose distance from $v$ in $\tilde{G}$ is at most $i$. Without repetitions, sample $n_{\text{hub}}$ 'hub' vertices from $\mathcal{N}_2(v)$ and call their set $P$. If $|\mathcal{N}_2(v)| < n_{\text{hub}}$, return to step 1.

3. Consider the union of 1-hop neighborhoods of the 'hub' vertices: $D = \bigcup_{w \in P \cup \{v\}} \mathcal{N}_1(w)$.

4. If $w \in D$ is a leaf in the restriction $\tilde{G}_D$ of $\tilde{G}$ to D, remove it from $D$ with probability $p_{\text{out}}$.

5. Sample a set $D'$ of $n_{\text{min}}$ vertices from $D$, such that the restriction of $\tilde{G}$ to $D' \cup P \cup \{v\}$ is a connected subgraph. Let $P' = D' \cup P \cup \{v\}$. If the restriction $\tilde{G}_{P'}$ of $\tilde{G}$ to $P'$ is a subgraph of the observable graph $G$, return to step 1.

6. For each node $w$ in $D'$ independently, choose it to be portrayed by a constant term with probability $\frac{p_{\text{const}}}{d^2_{P'}(w)}$, where $d_{P'}(w)$ is the degree of $w$ in restriction of $\tilde{G}$ to $P'$.

7. The restriction $\tilde{G}_{P'}$ of $\tilde{G}$ to $P'$ is then converted into the corresponding conjunctive formula, by transforming each edge $r(w_1, w_2) \in E(\tilde{G}_{P'})$ into a literal $r(w_1, w_2)$. The vertex $v$ is then replaced by the single free variable $x_1$ and all nodes that were not chosen to be constant, are realized by distinct existentially quantified variables.

For formulas sampled from FB15k-237, we choose $n_{\text{min}} = 15$, while for NELL instances, $n_{\text{min}} = 12$, due to the sparsity of the knowledge graph. We consider three different choices of the parameters $n_{\text{hub}}, p_{\text{const}}$ and $p_{\text{out}}$, resulting in three distinct splits, namely "3-hub", "4-hub" and "5-hub", and sample 1000 formulas of each type. Using an SQL engine (Raasveldt and Mühleisen, 2019), we then solve these queries with respect to both observable and unobservable knowledge graphs, discarding those with no hard answers. The parameter values for each split are presented in Table 8.

### C.2 Query answer classification datasets

We propose two benchmarks for query answer classification: FB15k-237-QAC and NELL-QAC. Instances in each dataset are stored in a unified form:

$$(Q(x), C_Q, W_Q)$$

Table 8: Hyperparameters for the generated dataset splits.

| Split | $n_{\text{hub}}$ | $p_{\text{const}}$ | $p_{\text{out}}$ |
|-------|------|--------|------|
| 3-hub | 2 | 0.6 | 0.95 |
| 4-hub | 3 | 0.8 | 0.97 |
| 5-hub | 4 | 1.0 | 0.99 |

Table 9: Statistics of introduced QAC datasets.

| | **2p** | **3p** | **pi** | **ip** | **inp** | **pin** | **3-hub** | **4-hub** | **5-hub** |
|---|---|---|---|---|---|---|---|---|---|
| | | | | | **FB15k-237-QAC** | | | | |
| **#queries** | 500 | 500 | 500 | 500 | 500 | 500 | 300 | 300 | 300 |
| **#answers** | 9818 | 9828 | 9632 | 9358 | 9808 | 9898 | 2036 | 1988 | 2028 |
| **%easy** | 26.5% | 24.0% | 27.5% | 28.7% | 35.8% | 32.9% | 18.7% | 16.6% | 17.0% |
| **%hard** | 23.5% | 26.0% | 22.5% | 21.3% | 14.2% | 17.1% | 31.3% | 33.4% | 33.0% |
| **%neg** | 50.0% | 50.0% | 50.0% | 50.0% | 50.0% | 50.0% | 50.0% | 50.0% | 50.0% |
| | | | | | **NELL-QAC** | | | | |
| **#queries** | 500 | 500 | 500 | 500 | 500 | 500 | 300 | 300 | 300 |
| **#answers** | 9708 | 9702 | 9478 | 9694 | 9698 | 9888 | 2174 | 2186 | 1922 |
| **%easy** | 23.6% | 22.6% | 25.2% | 23.6% | 35.8% | 32.8% | 15.9% | 14.4% | 13.7% |
| **%hard** | 26.4% | 27.4% | 24.8% | 26.4% | 14.2% | 17.2% | 34.1% | 35.6% | 36.3% |
| **%neg** | 50.0% | 50.0% | 50.0% | 50.0% | 50.0% | 50.0% | 50.0% | 50.0% | 50.0% |

where $Q(x)$ is the input formula and $C_Q, W_Q$ are subsets of $V(G)$ with $|C_Q| = |W_Q|$ such that:

$$\tilde{G} \models Q(a/x) \quad \forall a \in C_Q \qquad \text{and} \qquad \tilde{G} \not\models Q(b/x) \quad \forall b \in W_Q$$

Each of our QAC benchmarks includes 9 splits, which can be broadly divided into two parts. Their statistics are more broadly described in Table 9. %easy, %hard, and %neg represent the proportions of easy answers, hard answers, and incorrect proposals in each split, respectively.

In the first part of our benchmarking, we utilize samples from existing CQA datasets, focusing exclusively on formulas that include projections. This choice is crucial, as grounding the free variables in non-projection queries (e.g., '2i', '3i', '2u', '2in', '3in') reduces the task to a set of independent link prediction problems, which do not meaningfully test reasoning capabilities beyond atomic fact retrieval. Similarly, disjunctive queries (e.g., 'up') can be decomposed into independent subqueries under the QAC setting, introducing little additional complexity and offering limited insight into a model's reasoning abilities.

We instead select a representative subset of six query types: '2p', '3p', 'ip', 'pi', 'inp', and 'pin', spanning key logical constructs such as projection, conjunction, and negation. This selection allows for both robust evaluation and continuity with prior work, enabling meaningful comparison with classical and neural CQA baselines under the classification-based objective. For each query type, we sample 500 queries to ensure a balanced and reliable evaluation.

For the main components of FB15k-237-QAC and NELL-QAC, we convert large base queries into QAC instances, reducing the size of each split to 300 queries. These samples are characterized by significant structural complexity, presenting a substantial challenge for both existing and future query answering methods.

In both cases, the size $|C_Q| = |W_Q|$ is chosen as $clip(|\{a \in V(G) : \tilde{G} \models Q(a/x)\}|, 5, 10)$. $W_Q$ is then sampled uniformly from the set of incorrect groundings for $Q(x)$, while $C_Q$ is drawn from the set of answers to $Q(x)$, assigning non-trivial answers twice higher probability than the easy ones.

Table 10: Statistics of introduced QAR datasets.

| | FB15k-237-QAR | | | NELL-QAR | | |
|---|---|---|---|---|---|---|
| | 3-hub | 4-hub | 5-hub | 3-hub | 4-hub | 5-hub |
| #queries | 1200 | 1200 | 1200 | 1000 | 1000 | 1000 |
| #trivial | 565 | 537 | 586 | 387 | 416 | 417 |
| #free=1 | 400 | 400 | 400 | 400 | 400 | 400 |
| #free=2 | 400 | 400 | 400 | 300 | 300 | 300 |
| #free=3 | 400 | 400 | 400 | 300 | 300 | 300 |

### C.3 Query answer retrieval datasets

Most samples in CQA benchmarks yield answers within the observable knowledge graph $G$. Due to their simplicity, these instances are trivial for query answer retrieval, as classical solvers can efficiently derive the correct answers. Consequently, we do not include such small queries in our FB15k-237-QAR and NELL-QAR datasets. Instead, we focus on addressing the limitations of current benchmarks by including more complex queries involving multiple free variables.

For the single free variable case, we select 400 base queries from each split. To generate formulas of arity 2, we randomly remove the quantification over one of the existentially quantified variables. The resulting query is then solved using an SQL engine, leveraging information from the initial answer set to optimize computation. An analogous methodology is applied to extend the arity 2 formulas to instances with 3 free variables. Statistics of the generated test splits are available in Table 10. **#trivial** is the number of samples admitting a trivial answer, and **#free=k** - arity $k$ formulas.

### C.4 Evaluation protocol

**Query answer classification.** We use the F1-score as the metric to measure the performance on the task of query answer classification. The reported F1-scores (Table 1) are an average of F1-scores for single instances $(Q(x), C_Q, W_Q)$ taken over the whole dataset. Formally, letting $\mathcal{D}$ be the considered dataset and denoting by $A(\theta, Q)$ the set of entities from $C_Q \cup W_Q$ marked by the model $\theta$ as correct answers to $Q(x)$, we report:

$$F1_{\text{QAC}}(\theta) = \frac{1}{|\mathcal{D}|} \sum_{(Q(x), C_Q, W_Q) \in \mathcal{D}} \frac{2|A(\theta, Q) \cap C_Q|}{2|A(\theta, Q) \cap C_Q| + |A(\theta, Q) \backslash C_Q| + |W_Q \cap A(\theta, Q)|}$$

**Query answer retrieval.** We adapt the F1-score metric to the task of QAR. In particular, we count a positive outcome (i.e. solution prediction) as correct if and only if it is a true answer to the query. Given a model $\theta$, let $\text{Rec}(\theta)$ be the proportion of *correctly answered* positive instances in the dataset, while $\text{Prec}(\theta)$ be the ratio of *correctly answered* positive instances among the queries for which $\theta$ predicted a solution. We then report:

$$F1_{\text{QAR}}(\theta) = \frac{2}{\frac{1}{\text{Prec}(\theta)} + \frac{1}{\text{Rec}(\theta)}}$$

## D Link predictors

As mentioned in Section 5.1, we incorporate a link predictor into our architecture, to address the problem of deducing facts not presented in the observable knowledge graph. We consider three different model types from the existing CQA literature: transductive knowledge graph embedding method ComplEx (Trouillon et al., 2016) used in QTO and FIT, inductive (on nodes) method NBFNet (Zhu et al., 2021) employed by GNN-QE, and inductive (on nodes and relations) knowledge graph foundation model ULTRA (Galkin et al., 2024a), lying at the heart of ULTRAQUERY.

### D.1 ComplEx

Recall that a ComplEx model $\chi$ assigns each entity $e \in V(G)$ and each relation $r \in R(G)$, a $d_\chi$-dimensional complex-valued vector $v_e, w_r \in \mathbb{C}^{d_\chi}$. We choose the hidden dimension of $d_\chi = 1000$ for all experiments. For each triple $(r, a, b) \in R(G) \times V(G) \times V(G)$, the score of the entities $a, b$ being in relation $r$ is derived as:

$$\chi(r, a, b) = \Re e\left(\langle v_a, w_r, \overline{v}_b \rangle\right) = \Re e\left(\sum_{i=1}^{d_\chi} (v_a)_i (w_r)_i \overline{(v_b)_i}\right)$$

**Training.** For training, we follow the relation prediction methodology, presented in Chen et al. (2021), evaluating the loss as a sum over all known facts $r(a, b) \in E(G)$ of three cross-entropy losses, marginalizing the head, the relation and the tail:

$$\mathcal{L}_r(\chi) = -\sum_{r(a,b) \in E(G)} \left(\log(p_{\chi,\tau}(a|r, b)) + \log(p_{\chi,\tau}(b|a, r)) + \lambda_{rel} \log(p_{\chi,\tau}(r|a, b))\right) + \mathcal{L}_{reg}$$

where $\mathcal{L}_{reg}$ is a nuclear 3-norm Lacroix et al. (2018) regularization term and the marginal probabilities are evaluated as:

$$\mathcal{L}_{reg} = \sum_{i=1}^{d_\chi} \left(2 \cdot \sum_{a \in V(G)} |(v_a)_i|^3 + \sum_{r \in R(G)} |(w_r)_i|^3\right)$$

$$p_{\chi,\tau}(a|r, b) = \frac{\exp(\tau \cdot \chi(r, a, b))}{\sum_{a' \in V(G)} \exp(\tau \cdot \chi(r, a', b))}$$

$$p_{\chi,\tau}(b|a, r) = \frac{\exp(\tau \cdot \chi(r, a, b))}{\sum_{b' \in V(G)} \exp(\tau \cdot \chi(r, a, b'))}$$

$$p_{\chi,\tau}(r|a, b) = \frac{\exp(\tau \cdot \chi(r, a, b))}{\sum_{r' \in R(G)} \exp(\tau \cdot \chi(r', a, b)))}$$

where $\tau$ is a factor controlling the temperature of the applied softmax function. During training, we set $\tau = 1$. For each dataset, the model is trained using the AdaGrad Duchi et al. (2011) optimizer with a learning rate 0.1 for 500 epochs, and the checkpoint maximizing validation accuracy is chosen for testing.

**Conversion to the probability domain.** To match the definition of a link predictor from Section 3, the uncalibrated scores $\chi(r, a, b)$ assigned by the ComplEx model $\chi$ need to be converted into probabilities $\rho_\mathbb{C}(r, a, b) = \mathbb{P}(r(a, b) \in E(\tilde{G})|\chi)$. We follow the ideas used in QTO (Bai et al., 2023) and FIT Yin et al. (2024), and set them as proportional to the marginal probabilities $p_{\chi,\tau}(b|a, r)$. By definition, $p_{\chi,\tau}(\cdot|a, r)$ defines a distribution over $V(G)$:

$$\sum_{b \in V(G)} p_{\chi,\tau}(b|a, r) = 1$$

Therefore, to match the objective:

$$\sum_{b \in V(G)} \mathbb{P}(r(a, b) \in E(\tilde{G})|\chi) = \left|\left\{b \in V(G) : r(a, b) \in E(\tilde{G})\right\}\right|$$

we multiply the marginal probabilities by a scaling factor $Q_{a,r}$, specific to the pair $(a, r)$:

$$\rho_\mathbb{C}(r, a, b) = \mathbb{P}(r(a, b) \in E(\tilde{G})|\chi) = Q_{a,r} \cdot p_{\chi,\tau}(b|a, r)$$

We consider two scaling schemes: $Q_{a,r}^{\text{QTO}}$ introduced in QTO, and $Q_{a,r}^{\text{FIT}}$ described by FIT. Both methods base on the cardinality of the set $E_{a,r} = \{b \in V(G) : r(a, b) \in E(G)\}$ of trivial answers to the query $Q(x) = r(a, x)$:

$$Q_{a,r}^{\text{QTO}} = |E_{a,r}|$$

$$Q_{a,r}^{\text{FIT}} = \frac{|E_{a,r}|}{\sum_{b \in E_{a,r}} p_{\chi,\tau}(b|a, r)}$$

During validation, we search for the best values for $\tau$ among $[0.5, 1, 2, 5, 10, 20]$ on each validation query type. We notice that $\tau = 20$ performs best in all experiments. The resulting link predictors $\rho_{\mathbb{C}}^{\text{FIT}}$ and $\rho_{\mathbb{C}}^{\text{QTO}}$, after augmenting them with links from the observable graphs as described below, are then plugged into the respective neuro-symbolic frameworks for QTO and FIT evaluations on small-query QAC splits. For experiments with AnyCQ equipped with ComplEx-based predictors, we use the FIT, as it proved more accurate during validation.

## D.2 NBFNet

As the second studied predictor, we consider Neural Bellman-Ford Network (Zhu et al., 2021), constituting the main processing unit in GNN-QE (Zhu et al., 2022). For the AnyCQ experiments, we reuse the NBFNet checkpoints obtained from training GNN-QE over the considered datasets. We follow the configurations from the original repository – models are trained for 10 epochs, processing 48,000 instances per epoch for the FB15k-237 training, and 24,000 samples per epoch for NELL, with Adam (Kingma and Ba, 2015) optimizer with learning rate 0.005. We validate 0.25 to be the optimal threshold for binarizing GNN-QE predictions, and apply it for the small-query QAC experiments. When testing AnyCQ with the underlying NBFNet models, we first binarize the output of the NBFNet $\nu$:

$$\rho_\nu(r, a, b) = \begin{cases} 1 & \text{if } \nu\big(r(a, b)\big) \geq t \\ 0 & \text{if } \nu\big(r(a, b)\big) < t \end{cases}$$

After validation, we set $t = 0.5$ for the small-query FB15k-237-QAC splits, $t = 0.4$ for the small-query NELL-QAC splits and $t = 0.6$ for all large-query evaluations.

## D.3 Ultra

Finally, to test AnyCQ's ability of inductive link prediction over unseen relation, we consider Ultra (Galkin et al., 2024a), a prominent knowledge graph foundation models, as the third studied predictor for zero-shot inference. For the AnyCQ experiments, we directly apply the 3g checkpoints from the original Ultra repository, which are pre-trained on FB15k-237 (Toutanova et al., 2015), WN18RR (Dettmers et al., 2018), and CoDEx Medium (Safavi and Koutra, 2020) for 10 epochs with 80,000 steps per epoch, with AdamW (Loshchilov and Hutter, 2019) using learning rate of 0.0005. Similarly to the methodology applied to NBFNet, we binarize the output of the Ultra model $\upsilon$ when equipping it to AnyCQ. In this case, following validation, we choose $t = 0.4$ as the best threshold for small query QAC experiments, and a higher $t = 0.9$ for formulas in large QAC and QAR splits.

As an additional baseline, we compare another state-of-the-art CQA method over unseen relation Ultra-Query (Galkin et al., 2024b), which also utilizes Ultra as its link predictor. We validate that 0.2 is the best answer classification threshold for UltraQuery checkpoints provided ni the original repository, trained only on the CQA benchmark based on FB15k-237. We highlight that the results of UltraQuery on NELL are hence *zero-shot inference*, since NELL is not in the pretraining dataset of the evaluated checkpoints.

## D.4 Incorporating the observable knowledge graph

To account for the knowledge available in the observable graph $G$, we augment all considered link predictors $\rho$, setting $\rho(r, a, b) = 1$ if $r(a, b) \in E(G)$. To distinguish between known and predicted connections, we clip the predictor's estimations to the range $[0, 0.9999]$. Combining all these steps together, given a predictor $\rho$, in our experiments we use:

$$\pi(r, a, b) = \begin{cases} 1 & \text{if } r(a, b) \in E(G) \\ \min\big(\rho(r, a, b), 0.9999\big) & \text{otherwise} \end{cases}$$

This methodology is applied for all AnyCQ experiments, to each of $\rho_{\mathbb{C}}, \rho_\nu$ and $\rho_\upsilon$, obtaining the final $\pi_{\mathbb{C}}, \pi_\nu$ and $\pi_\upsilon$, directly used for ComplEx-based, NBFNet-based and Ultra-based evaluations, respectively.

Table 11: Average F1-scores of AnyCQ on the query answer classification task.

| Dataset | Predictor | 2p | 3p | pi | ip | inp | pin | 3-hub | 4-hub | 5-hub |
|---|---|---|---|---|---|---|---|---|---|---|
| **FB15k-237-QAC** | ComplEx | 66.9 | 63.1 | 70.7 | 67.6 | **78.4** | 75.2 | 39.5 | 32.3 | 36.1 |
| | NBFNet | **75.8** | **71.3** | **82.1** | 78.8 | 76.7 | **75.7** | **52.4** | **49.9** | **51.9** |
| | Ultra | 70.4 | 56.2 | 77.3 | 70.6 | 72.4 | 73.0 | 32.6 | 26.9 | 29.1 |
| **NELL-QAC** | ComplEx | 63.8 | 64.0 | 68.2 | 61.7 | 74.8 | 75.0 | 39.1 | 40.0 | 34.9 |
| | NBFNet | 76.2 | **72.3** | 79.0 | 75.4 | **76.7** | **75.3** | **57.2** | **52.6** | **58.2** |
| | Ultra | 76.0 | 23.0 | **81.2** | **76.3** | 70.8 | 74.0 | 33.2 | 30.8 | 25.5 |

Table 12: F1-scores of AnyCQ equipped with different predictors on the QAR datasets.

| Dataset | Predictor | 3-hub | | | | 4-hub | | | | 5-hub | | | |
|---|---|---|---|---|---|---|---|---|---|---|---|---|---|
| | | $k=1$ | $k=2$ | $k=3$ | total | $k=1$ | $k=2$ | $k=3$ | total | $k=1$ | $k=2$ | $k=3$ | total |
| **FB15k-237-QAR** | ComplEx | 67.3 | 56.3 | 43.4 | 56.3 | 57.7 | **54.4** | 45.6 | 52.7 | 62.8 | 54.3 | **44.1** | 54.1 |
| | NBFNet | **67.8** | **62.3** | **50.2** | **60.5** | **60.4** | 54.0 | **48.2** | **54.5** | **63.0** | **56.9** | 43.1 | **54.8** |
| | Ultra | 65.3 | 57.1 | 44.1 | 56.0 | 57.1 | 52.4 | 42.2 | 50.8 | 59.4 | 54.3 | 41.3 | 52.0 |
| **NELL-QAR** | ComplEx | 62.8 | 50.0 | 34.6 | 51.4 | 61.7 | 52.1 | 40.7 | 53.0 | 55.1 | 50.0 | 36.5 | 48.4 |
| | NBFNet | **66.7** | **55.1** | **39.1** | **55.8** | **65.1** | **57.1** | **46.5** | **57.6** | **58.7** | **51.1** | **39.6** | **51.1** |
| | Ultra | 57.4 | 44.6 | 31.5 | 46.5 | 56.4 | 43.8 | 35.2 | 46.7 | 49.1 | 40.4 | 31.5 | 41.5 |

## D.5 Combination with AnyCQ

As mentioned in Section 5.4, the AnyCQ framework can be equipped with any link predictor capable of predicting relations over the studied knowledge graph. For this reason, to ensure that our choice matches the most accurate setup, we validate the performance of the predictors described in previous subsections, and test AnyCQ combined with ComplEx-based predictor with FIT scaling (Appendix D.1), NBFNet (Appendix D.2) and Ultra (Appendix D.3). Following validation, we choose NBFNet to be equipped for AnyCQ evaluations in all main experiments (Table 1 and Table 2).

As an additional ablation study, we generate the test results of the remaining combinations. The results on the QAC task are presented in Table 11, while the scores on QAR benchmarks are shown in Table 12. For the small-query QAC splits, we notice that NBFNet and Ultra strongly outperform ComplEx on positive formulas ("2p", "3p", "ip", "pi"), while struggling more with queries with negations ("inp", "pin"). The observed drop in Ultra results on the "3p" split is due to the model predicting too many links to be true, limiting the guidance from PE labels (almost each entity can be a head/tail of each relation). For large query classification, NBFNet produces much better F1-scores, exceeding 50% on almost all splits, while the remaining predictors consistently score under 40%. The evaluations on the QAR benchmarks further justify the choice of NBFNet as the equipped link predictor for AnyCQ – it consistently outperforms ComplEx and Ultra, achieving the best results on most FB15k-237-QAR, and all NELL-QAR splits.

Interestingly, we again point out that the used Ultra model has not been trained on the NELL dataset. Regardless, it manages to match the performance of the ComplEx-based predictor on NELL-QAC and NELL-QAR benchmarks. Combining this observation with our ablation of transferability of AnyCQ models between datasets (Section 6.4, Table 4), we can assume that similar results would be achieved when running the evaluation with AnyCQ model trained on FB15k-237. Such framework would then answer queries over NELL in a true zero-shot, fully inductive setting. In future work, we look forward to exploring combinations of AnyCQ search engines trained over broad, multi-dataset data, with fully inductive link predictors (like Ultra), to achieve foundation models capable of answering arbitrary queries over arbitrary, even unseen, knowledge graphs.

Table 13: Recall on the easy samples from the QAR datasets with different SQL timeouts.

| Model | Timeout [s] | 3-hub | | | | 4-hub | | | | 5-hub | | | |
|---|---|---|---|---|---|---|---|---|---|---|---|---|---|
| | | $k=1$ | $k=2$ | $k=3$ | total | $k=1$ | $k=2$ | $k=3$ | total | $k=1$ | $k=2$ | $k=3$ | total |
| **FB15k-237-QAR** | | | | | | | | | | | | | |
| AnyCQ | 60 | 83.3 | **84.2** | **71.6** | **80.5** | 79.1 | 71.1 | **72.1** | 74.3 | 74.0 | **67.0** | 52.3 | 65.0 |
| SQL | 30 | 82.8 | 51.5 | 20.3 | 55.6 | 71.9 | 49.7 | 34.4 | 53.6 | 74.0 | 51.0 | 43.8 | 56.8 |
| | 60 | 88.7 | 61.2 | 26.4 | 62.8 | 87.2 | 71.7 | 52.6 | 71.9 | 85.3 | 63.6 | 61.4 | 70.5 |
| | 120 | **93.2** | 69.4 | 35.8 | 69.9 | **90.8** | **76.5** | 55.2 | **75.8** | **88.7** | **67.0** | **67.6** | **74.7** |
| **NELL-QAR** | | | | | | | | | | | | | |
| AnyCQ | 60 | 93.0 | **89.4** | **86.5** | **90.7** | 89.6 | **88.8** | **79.6** | **87.1** | 89.4 | **84.5** | **81.3** | **86.1** |
| SQL | 30 | 88.5 | 57.5 | 33.8 | 69.0 | 82.4 | 57.8 | 55.9 | 69.2 | 80.1 | 63.1 | 46.7 | 67.5 |
| | 60 | 94.5 | 69.0 | 55.4 | 79.6 | 90.1 | 69.0 | 63.4 | 77.9 | 88.8 | 78.6 | 64.0 | 80.2 |
| | 120 | **95.5** | 73.5 | 58.1 | 81.9 | **93.8** | 71.5 | 68.8 | 81.6 | **91.3** | 82.5 | 65.4 | 82.9 |

## E Extended evaluation over QAR processing times

We analyze the processing times and retrieval performance of AnyCQ and the SQL engine on the QAR benchmarks, focusing on how both systems scale with query complexity.

To understand the impact of the timeout threshold on SQL performance, we evaluate it on all QAR splits using three different time limits: 30, 60, and 120 seconds. Results are reported in Table 13. We observe that the SQL engine's recall on *easy* queries improves marginally as the timeout increases, particularly for queries with multiple free variables. However, its performance consistently deteriorates with increasing arity, even under extended time limits. In contrast, AnyCQ maintains strong performance across all splits and consistently outperforms SQL on retrieving easy answers to high-arity queries. Additionally, it is worth noting that AnyCQ is also capable of retrieving *hard* answers, which SQL fails to find under any timeout setting.

Table 14 presents the average processing times per query under the 60-second timeout. We find that AnyCQ exhibits significantly more stable and predictable runtimes compared to SQL. By design, AnyCQ's processing time is independent of the number of free variables or other structural properties of the query, and remains efficient even on the larger NELL-QAR benchmark. In contrast, the SQL engine shows a steep increase in average runtime as query arity increases, with many queries exceeding the timeout. Notably, for queries with arity of at least 2, AnyCQ outperforms the SQL engine on average across all FB15k-237-QAR splits, while remaining competitive for single-variable queries. This reflects SQL's sensitivity to query structure and its inefficiency in evaluating high-arity or cyclic formulas.

Across both tables, AnyCQ consistently demonstrates superior scalability and robustness. It handles structurally complex queries with multiple free variables more efficiently than SQL and is uniquely able to retrieve both easy and hard answers. Meanwhile, the SQL engine is unreliable in this setting - its performance is heavily dependent on query structure and its success rate is strongly limited by the time budget. These results validate our motivation for introducing QAR and support AnyCQ as a reliable neural alternative to classical engines for complex query answering.

Table 14: The comparison of processing times (in seconds) on the QAR task. We report the minimum, maximum and average processing time per split, together with the standard deviation (sd) and the number of unanswered samples due to the time restriction $n_{exc}$.

| Dataset | Split | Arity | Model | min | max | avg | sd | $n_{exc}$ |
|---------|-------|-------|-------|-----|-----|-----|-----|-----|
| FB15k-237-QAR | 3-hub | $k=1$ | AnyCQ | 12.96 | 30.74 | 19.70 | 2.91 | 0 |
| | | | SQL | 0.66 | 60.00 | 11.14 | 17.20 | 31 |
| | | $k=2$ | AnyCQ | 13.01 | 30.56 | 19.57 | 2.99 | 0 |
| | | | SQL | 0.32 | 60.00 | 30.87 | 24.84 | 143 |
| | | $k=3$ | AnyCQ | 13.05 | 30.53 | 19.11 | 2.74 | 0 |
| | | | SQL | 0.33 | 60.00 | 43.77 | 22.99 | 235 |
| | 4-hub | $k=1$ | AnyCQ | 13.37 | 32.72 | 20.44 | 2.98 | 0 |
| | | | SQL | 0.30 | 60.00 | 13.90 | 19.34 | 40 |
| | | $k=2$ | AnyCQ | 13.36 | 32.78 | 20.78 | 3.10 | 0 |
| | | | SQL | 0.37 | 60.00 | 26.63 | 24.49 | 115 |
| | | $k=3$ | AnyCQ | 14.24 | 32.99 | 20.42 | 2.89 | 0 |
| | | | SQL | 0.36 | 60.00 | 32.77 | 24.88 | 154 |
| | 5-hub | $k=1$ | AnyCQ | 13.43 | 36.24 | 20.73 | 3.08 | 0 |
| | | | SQL | 0.27 | 60.00 | 11.98 | 18.40 | 37 |
| | | $k=2$ | AnyCQ | 13.95 | 36.25 | 21.74 | 3.05 | 0 |
| | | | SQL | 0.57 | 60.00 | 23.47 | 24.40 | 106 |
| | | $k=3$ | AnyCQ | 15.96 | 29.04 | 21.33 | 2.58 | 0 |
| | | | SQL | 0.26 | 60.00 | 28.83 | 25.39 | 131 |
| NELL-QAR | 3-hub | $k=1$ | AnyCQ | 21.87 | 59.57 | 35.35 | 6.27 | 0 |
| | | | SQL | 0.16 | 60.00 | 6.59 | 13.77 | 15 |
| | | $k=2$ | AnyCQ | 22.20 | 59.65 | 35.33 | 7.09 | 0 |
| | | | SQL | 0.13 | 60.00 | 20.94 | 23.91 | 69 |
| | | $k=3$ | AnyCQ | 18.81 | 59.69 | 35.29 | 7.61 | 0 |
| | | | SQL | 0.14 | 60.00 | 21.77 | 25.13 | 75 |
| | 4-hub | $k=1$ | AnyCQ | 19.15 | 58.00 | 36.46 | 6.93 | 0 |
| | | | SQL | 0.14 | 60.00 | 7.79 | 15.59 | 24 |
| | | $k=2$ | AnyCQ | 19.14 | 57.58 | 35.70 | 7.54 | 0 |
| | | | SQL | 0.13 | 60.00 | 17.47 | 22.54 | 51 |
| | | $k=3$ | AnyCQ | 23.13 | 56.66 | 36.04 | 6.63 | 0 |
| | | | SQL | 0.12 | 60.00 | 19.49 | 23.99 | 62 |
| | 5-hub | $k=1$ | AnyCQ | 23.40 | 60.00 | 37.27 | 6.58 | 3 |
| | | | SQL | 0.17 | 60.00 | 6.71 | 15.01 | 22 |
| | | $k=2$ | AnyCQ | 25.91 | 60.00 | 36.97 | 6.63 | 2 |
| | | | SQL | 0.16 | 60.00 | 14.62 | 20.66 | 38 |
| | | $k=3$ | AnyCQ | 24.25 | 60.00 | 36.84 | 5.91 | 1 |
| | | | SQL | 0.17 | 60.00 | 17.44 | 22.18 | 45 |

