# OpenReview forum: "One Model, Any Conjunctive Query: Graph Neural Networks for Answering Queries over Incomplete Knowledge Graphs"
_TMLR — Rejected by TMLR_

### Review · Reviewer_qMEP · 2025-05-23

**Summary Of Contributions:**

The submission reformulates an extensive body of existing works on (neural) query answering from purely ranking formulation to the actual query answering formulation (with either checking the truthfulness or answer retrieval). The paper formulates the query answering as a combinatorial optimisation problem over an arbitrary link predictor. The experiments are quite extensive and really validate many aspects of the method.

**Audience:**

Yes

**Broader Impact Concerns:**

No concerns

**Claims And Evidence:**

Yes

**Requested Changes:**

Properly position your work with respect to the work of Arakelyan et al and Van Krieken. This is essential to secure my recommendation.

Explaining the motivation of design choices and their consequences (fee vs existential variables; constants among entities; assumption on the range of [0,1]). This is the essential aspect.

Properly contextualising the work within the declarative problem-solving perspective. Not essential, but would make it stronger.

Improving the writing of the method by simplifying it -- not essential because the paper is already quite readable, but it would make it stronger

**Strengths And Weaknesses:**

STRENGTHS
---------------

The paper tackles one of the big open challenges in the neural QA field, which rendered the existing approaches essentially useless -- the existing link predictors merely assign scores to facts and the scores themselves are often not comparable nor interpretable.


The proposal is also very general in the way that it is parametrised by a link predictor. This is a major strength as the proposed framework can incorporate any future advances in link predictions.

I find the technical formulation of the problem elegant. The framework is technically sound, I have not found issues beyond the few choices mentioned below. I also really appreciate the completeness and soundness theorems. But I do have two questions: should I interpret the completeness theorem essentially the same as the completeness in the case of Markov Chain Monte Carlo methods (essentially, with sufficient number of samples, the result is correct)?

The experimental evaluation is done really well. The experiments include various baselines and look into many aspects of the methods beyond "it performs well". I really appreciate this.

The paper is also very well written (for the most part*). The problem analysis is done really well, the formulations are clear and sensical, and the interpretations are not overstated.





WEAKNESSES

One big weakness that I see is in addressing the related work. The paper of Arakelyan et al (2020) is very briefly mentioned, but I see quite a few connections between this work and the submission -- both tackle a similar problem and formulate it as a constraint optimisation problem. A deeper discussion on the differences and relative advantages/disadvantages is really needed here. Moreover, this proposal really goes in the way of declarative problem solving, which is prominent in the symbolic approaches to question answering and follows the same idea of "one model, any query". Positioning this work within the concept of declarative problem solving does it justice, but also positions the current work within the idea that has a significant history in AI, while being missing from many learning approaches.

On a similar line, as the proposed framework relies on fuzzy logic to interpret logical formulas, it would be fair to also be explicitly aware of the limitation fuzzy logic imposes on complex queries. This is excellently covered in the following publication

Analyzing differentiable fuzzy logic operators
E Van Krieken, E Acar, F Van Harmelen
Artificial Intelligence 302, 103602


While I think I got a good understanding of the proposed method, the writing made that a bit challenging. This is a shame, because the method itself is the only part of the paper that is written a bit subpar. Sometimes it is the motivation for the choice that is missing, sometimes the missing part if the consequence of a particular choice. Below, I list the parts that confused me.

The manuscript makes a strict distinction between existential and free variables. While I see the syntactic difference between them, I don't see the consequences of this distinction, nor how the method treats them differently. If I have a query with free variables that I want answered, they are treated as existentially quantified variables in effect, meaning that any answer to them coming from a KG is good enough. The only interpretation that I could somewhat see is that the free variables could be interpreted as variables whose values we don't care about. The free variables also just disappear from the remaining formulation. So, I really think their role should be explicitly mentioned because right now the emphasis on the distinction makes them look important, but the consequences are left too implicit.

When it comes to the technical choices, several of them are left unjustified. For example, I don't see why you would include constants among the entity nodes? Their values cannot change, and there is no decision to be made there. Similarly, are negative literals treated in any way differently from positive literals?
When describing which constants end up in the graph formulation, the manuscript states that the "value vertices correspond to feasible entity-value assignments" The word feasible does a lot of heavy lifting here. What does that exactly mean here? The safe choice would be to include all possible entities, which has a major impact on the scalability, or assume the existence of types, which then makes fewer options. Please make clear how do you determine the feasible constants.

One thing that I am wondering is the assumption that the link predictors produce values in the range of [0,1]. This is not generally the case, no? They generally produce arbitrary values. Do you do anything to make sure they produce such values? If not, and you assume that the link predictor does that, which is completely fine, then the paper makes a too strong of a claim that it can work with any link predictor.
Along the same lines, I would be a bit weary of interpreting the score their produce as confidence value for the threshold of 0.5. This aspect should be discussed in the manuscript.

It took me a while to understand the edge labels. If I understood correctly, the potential links are come from the link predictor and they are essentially triplets for which the link predictor score is > 0.5, and the light labels are the current assignment to the variables. Is that correct? This information is not easy to digest from the current description, it took me several readings to get to this but in the end it is not that complicated. As this is really an essential part, I would strongly suggest to make it as easy to understand as possible.

While minor, divergence from the standard meaning of some terms makes it confusing to read. For example, the "entity" is used to refer to both constants and variables, while the standard meaning makes it equivalent to constants.

the paper positions itself with a pretrained link predictor already provided. I am wondering, for the sake of really understanding the method, could you also learn the link predictor if a pre-trained one is not available? Would that result in a link predictor that behaves different than the one trained only on triplets?

---

> ### Author Response · Authors · 2025-06-07
> **Response 1/n: Related Work**
>
> > Properly position your work with respect to the work of Arakelyan et al and Van Krieken. This is essential to secure my recommendation.
>
> Arakelyan et al. (2020) introduced the CQD framework, which was indeed one of the first to formalize complex query answering using fuzzy logic over KG embeddings. Their work pioneered the idea of treating CQA as an optimization problem, specifically maximizing the fuzzy logic truth value of a query over entity assignments. This formulation has since influenced a broad range of methods, including QTO and FIT, which adopt similar fuzzy logic semantics but differ in how they navigate the search space. For example, QTO avoids the beam search approximation of CQD by exploiting the sparsity of the underlying relation matrices, allowing exact inference for tree-like queries. FIT extends this line to handle cyclic queries, though at the cost of exponential time complexity. In this sense, CQD can be viewed as a foundational work in the fuzzy logic-based view of CQA, but not uniquely close to our proposed approach. Many later methods also build on and improve this perspective, which became the standard formulation in the CQA literature.
>
> CQD-CO, the continuous optimization variant proposed by Arakelyan et al., departs more significantly from our framework. In CQD-CO, variables are assigned continuous embeddings rather than concrete entities, and the query is optimized over these latent representations using gradient-based methods. This is conceptually closer to fully neural CQA models such as LMPNN or CLMPT, where query answers are produced by embedding propagation and comparison in the latent space. In contrast, AnyCQ performs a discrete search over grounded assignments in the knowledge graph, guided by a link predictor, and evaluates the satisfiability of symbolic Boolean formulas. This enables principled reasoning with explicit variable bindings over observed and predicted knowledge, which CQD-CO does not support. However, we acknowledge its unique approach and will include the description of CQD-CO in the Related Work section.
>
> Overall, while CQD laid foundational ideas for fuzzy logic-based reasoning in CQA, and CQD-CO presents a novel neural optimization view, our work is both methodologically and operationally distinct. The limitations of CQD and related methods: inability to handle cyclic queries, restriction to formulas with a single free variable, and unclear embedding-to-probability transition, are already discussed in the paper. For these reasons, we do not perceive CQD as an approach particularly worth highlighting, but we are eager to discuss this matter further.
>
> > It would be fair to also be explicitly aware of the limitation fuzzy logic imposes on complex queries.
>
> We are aware of several limitations associated with fuzzy logic, particularly in the context of scaling to large and structurally complex queries. While fuzzy logic-based aggregation is widely adopted in the CQA literature and performs well on small, standard query types, we observed that certain t-norms, such as the product t-norm, introduce significant numerical degradation as query size grows. For example, in a conjunction of 10 literals each scored at 0.95, the product t-norm yields an aggregate score of $0.95^{10} \approx 0.6$, and the value drops rapidly as the number of literals increases. Given that our benchmarks include queries with over 20 literals, this behavior poses a serious issue for classification.
>
> To mitigate this, we adopt the Gödel t-norm (min operator), which maintains more stable scores even for large formulas. In the above example, it would return 0.95. Moreover, using the Gödel t-norm in combination with a 0.5 classification threshold is equivalent to applying propositional logic after binarizing the link predictor outputs with the threshold of 0.5. We will clarify this choice in Section 3 of the revised manuscript.
>
> Regarding the work of van Krieken et al. (2022), we note that their analysis primarily concerns the behavior of fuzzy logic operators in differentiable optimization settings. This is less relevant to our setup: although AnyCQ uses fuzzy logic to evaluate Boolean formula scores, these scores are not involved in end-to-end gradient-based learning. Specifically, the fuzzy score serves as a reward signal within the REINFORCE objective, and no gradients are backpropagated through the aggregation function itself. Furthermore, the link predictor used by AnyCQ is pre-trained independently on atomic link prediction instances, without requiring fuzzy logic for its optimization.
>
> In summary, while their insights are valuable in differentiable learning contexts, they do not directly apply to our framework trained in the reinforcement learning setting. Nonetheless, we appreciate the suggestion and agree that a broader discussion of fuzzy logic's limitations, especially regarding scalability, can be informative. We will consider including such a discussion in the appendix.

---

> > ### Comment · Reviewer_qMEP · 2025-06-18
> >
> > First of all, I apologise for the delay in the discussion.
> >
> > Re CQD: I follow your point and see the differences. My argument to stress this work more is that they introduce the foundational idea and should be credited for it. But the summary you provide in the last paragraph of your response I find very useful for the community to properly understand the limitations you overcome. I don't remember reading it as clearly in your manuscript. I would advise to add this directly.
> >
> > Re fuzzy semantics: this choice of the t-norm is essential in your design. And it seems now that it came with a proper motivation, but that has slipped my attention while reading your manuscript. Knowing this motivation is what makes your paper more relevant for the community and I would add it explicitly (apologies if I just missed it in the paper). Then I agree that the broader discussion can just go to the appendix.

---

> ### Author Response · Authors · 2025-06-07
> **Response 2/n: Design choices (part 1)**
>
> >  Explaining the motivation of design choices and their consequences (fee vs existential variables; constants among entities; assumption on the range of [0,1]). This is the essential aspect. \
> > The manuscript makes a strict distinction between existential and free variables. While I see the syntactic difference between them, I don't see the consequences of this distinction, nor how the method treats them differently.
>
> We appreciate the reviewer’s observation and agree that the distinction between free and existential variables could be more explicitly clarified in the manuscript. While syntactic, it is motivated primarily by the formulation of our studied tasks, particularly Query Answer Retrieval (QAR).
>
> In QAR, the goal is to return a concrete assignment to the free variables $\vec{x}$ such that the query $Q(\vec{x})$ is satisfied. The existential variables $\vec{y}$, on the other hand, serve as internal witnesses whose specific values are not required. Thus, the distinction reflects which variables we aim to recover in the output (the free variables), versus which ones serve only to validate the satisfiability of the formula and need not be specified. Shortly speaking, we do not care about the values of existential variables. This separation is standard in logic and query answering literature when defining what constitutes a valid "answer" to a query.
> From a modeling perspective, we agree with the reviewer that AnyCQ does not treat these two classes of variables differently during the search process. As explained in **Section 5**, the framework always searches for a full assignment over all variables in the query. For this reason, our implementation focuses on Boolean queries (i.e., no free variables). The distinction becomes relevant only at the final stage of the QAR task, where the answer is extracted by projecting the full satisfying assignment onto the free variables.
>
> We understand that this may have been unclear and will revise **Section 4** to explicitly state that the QAR objective is specifically to recover an assignment to the free variables of the input query.
>
> > Are negative literals treated in any way differently from positive literals?
>
> Thank you for the question. No, negative literals are not treated differently from positive ones, as our framework handles both uniformly, and the construction in **Section 5** applies equally to both cases. The only practical difference lies in how the edge labels (PE and LE) behave during the search. Specifically, since most assignments will *not* match a given positive literal, their corresponding PE and LE labels tend to be 0 for most of the search space. In contrast, the majority of assignments *will* satisfy a negative literal, resulting in PE and LE labels typically being 1 for those cases. This asymmetry means that positive literals provide stronger guidance for identifying promising assignments, while negative literals primarily act as constraints to filter out incorrect ones. Despite this difference in how they are supposed to influence the search dynamics, they are processed identically in the model architecture and message-passing steps.
>
> > For example, the "entity" is used to refer to both constants and variables, while the standard meaning makes it equivalent to constants. \
> > When describing which constants end up in the graph formulation, the manuscript states that the "value vertices correspond to feasible entity-value assignments". What does that exactly mean here? [...] Please make clear how do you determine the feasible constants.
>
> Referring to both constants and variables as "entities'' is indeed non-standard and was an oversight, especially as earlier in the paper, we introduce the term "term'' to mean both constants and variables. We will revise the manuscript to use "term'' where appropriate and reserve "entity'' exclusively for constants.
>
> In the current formulation, as stated immediately following the cited sentence, for each term $e$ mentioned in the Boolean formula $\Phi$, we construct value vertices $v_{e \rightarrow a}$ for each $a \in \mathcal{D}(e)$, where:
> $\mathcal{D}(e) = \{e\}$ if $e$ is a constant
> $\mathcal{D}(e) = V(G)$ if $e$ is a variable
> Thus, in our current implementation, we assume that variables can take any known constant as their value. We agree with the reviewer that although it proved sufficient for the benchmarks considered, this approach may not scale well on large graphs. We briefly touch on this issue in **Appendix A.6**, where we discuss potential strategies for domain restriction.

---

> > ### Comment · Reviewer_qMEP · 2025-06-18
> >
> > Re free vs existential: This is exactly how I would interpret this, but find it confusing as so much emphasis is put on it where in the end the distinction doesn't matter in the entire process. So, maybe rephrase it as a projection operation in the end that only preserves the variables the user cares about?
> >
> >
> > Re negative literals: understood, that was my interpretation. I was wondering if actually acknowledging that the negation is an operator on a literal, so making it as an additional node acting as an inverter of label, would make better use of the information during message-passing?

---

> > > ### Author Response · Authors · 2025-06-21
> > >
> > > > I was wondering if actually acknowledging that the negation is an operator on a literal, so making it as an additional node acting as an inverter of label, would make better use of the information during message-passing?
> > >
> > > Thank you for the suggestion. Indeed, in our current setup, we handle negated literals $\neg r(a,b)$ more as separate atomic relations $r_{neg} (a, b)$ rather than modeling negation as an operator on the base atom. This design follows from the structure of our computational graph, where literal nodes serve two purposes: (1) supplying edge labels (PE and LE) that guide search, and (2) passing messages between the terms mentioned in the literal.
> > >
> > > In this framework, introducing an explicit inverter node for negation would not be particularly useful. The "non-inverted" edge labels would not contribute to the optimization or message passing, as only the inverted logic matters for satisfying the literal. Therefore, representing negation implicitly achieves the same effect with a simpler structure.
> > >
> > > That said, if we were to generalize the edge labels as mentioned earlier:
> > > > One could generalize the PE and LE labels to be real-valued and parameterized by the predictor, allowing tighter integration between the two components.
> > >
> > > a more expressive and graph structure could become meaningful.
> > >
> > > To conclude, while such modification is not currently advantageous within our architecture, we agree that augmenting the computational graph to support such constructs is a promising direction for future research, especially in extending the framework toward broader classes of logical queries, such as disjunctions or nested structures.

---

> ### Author Response · Authors · 2025-06-07
> **Response 3/n: Design choices (part 2)**
>
> > I don't see why you would include constants among the entity nodes?
>
> This is effectively a design choice. Indeed, a binary relation $R(a,x)$ may be released as a unary relation $R_a(x)$, allowing us to remove all vertices representing constants from the computational graph. However, from the message propagation perspective, we notice that the value vertex corresponding to a constant contains a hidden embedding, storing information about the entity, which may serve as an ‘anchor’ during search. Moreover, including them does not impact the computational cost, which is anyway dominated by the value vertices for variables. Seeing potential benefits for little to no cost, we decided to include constants in our computational graph.
>
> > One thing that I am wondering is the assumption that the link predictors produce values in the range of [0,1]. This is not generally the case, no? They generally produce arbitrary values. Do you do anything to make sure they produce such values?
>
> We thank the reviewer for highlighting this important point. Indeed, many traditional link predictors, such as ComplEx, produce unbounded similarity scores rather than calibrated probabilities. However, because our objective in this work is to enable calibrated, decision-oriented reasoning (e.g., classifying whether a candidate answer is correct), we require link predictor outputs to lie in the [0,1] range and to represent interpretable confidence scores.
>
> Some link predictors, such as NBFNet and Ultra, are already trained using binary cross-entropy loss and naturally output values within [0,1] that approximate probabilities. For methods like ComplEx, which produce unbounded scores, we follow the approaches introduced in QTO and FIT and apply a softmax-based normalization over the marginal scores. This post-processing step, described in **Appendix D.1**, maps the raw similarity values to probabilities, ensuring compatibility with our framework. That said, as we report in **Appendix D.5**, even after applying such normalization, ComplEx-based models perform worse on QAC compared to models that natively operate in the probability domain. This performance gap further supports our motivation to build QAC and QAR around probability-calibrated predictors, better suited for making thresholded decisions.
>
> > If I understood correctly, the potential links are come from the link predictor and they are essentially triplets for which the link predictor score is > 0.5, and the light labels are the current assignment to the variables.
>
> During the search, we want to provide our AnyCQ model with two types of information:
> - Which assignments to existential variables are worth considering?
> - What changes to the current assignment will make the literals satisfied?
>
> We use PE labels to answer the first question, and LE labels for the second one. Let $\psi$ be a literal mentioning a variable $z$, and $a\in \mathcal{D}(z)$ be an entity. Consider the edge in the computational graph between the vertex representing $\psi$ and $z\rightarrow a$.
>
> The PE label for that edge is a binary answer to the question: “Is there any assignment that maps $z$ to $a$, and makes $\psi$ satisfied?”. In other words, can the substitution $z\rightarrow a$ lead to an assignment that satisfies $\psi$?
>
> The LE label is connected to the current assignment $\alpha$. It answers the question: “If we change the current assignment $\alpha$, so that $z$ is mapped to $a$ (but mappings for all other variables remain the same), will $\psi$ be satisfied?”. In our notation, this is equivalent to asking whether $\psi$ is satisfied under $\alpha_{z\rightarrow a}$. Hence, LE labels represent the marginal changes to the current assignment that make literals satisfied.
>
> We agree that the PE and LE labels are central to our framework, and a clear understanding of them is essential for grasping the overall method. We will revise their introduction in **Section 5.1** to improve clarity and will expand **Appendix A.6** with both formal definitions and intuitive explanations of PE and LE, to make the underlying concepts more accessible to the reader.

---

> ### Author Response · Authors · 2025-06-07
> **Response 4/n: Additional questions**
>
> > Should I interpret the completeness theorem essentially the same as the completeness in the case of Markov Chain Monte Carlo methods (essentially, with sufficient number of samples, the result is correct)?
>
> Indeed, the goal of our completeness theorem is to say that after a sufficient number of search steps, the model will visit an assignment that the underlying link predictor perceives as correct. However, AnyCQ is itself not a finite Markov Chain, as the embeddings of the value vertices in the computational graph are updated after each step. As a result, the derived set of distributions $\mu$ from which the next assignment is sampled can be distinct at each step, even when visiting the same assignment multiple times.
>
> > I am wondering, for the sake of really understanding the method, could you also learn the link predictor if a pre-trained one is not available? Would that result in a link predictor that behaves different than the one trained only on triplets?
>
> This is an insightful question and points to an interesting research direction. In our current setup, the link predictor is pre-trained independently on atomic link prediction tasks and remains fixed during training. The predictions it provides serve as scalar rewards in the REINFORCE objective; there is no gradient flow from the search policy (GNN) back to the predictor. This separation ensures modularity and allows the GNN to generalize across different predictors.
>
> In principle, the framework could support joint training. Viewed through a reinforcement learning lens, the link predictor (combined with fuzzy logic aggregation) acts as a value function, and the GNN acts as the policy. One could generalize the PE and LE labels to be real-valued and parameterized by the predictor, allowing tighter integration between the two components. This would enable backpropagation from the policy to the predictor and potentially allow the predictor to adapt to complex query structures rather than just atomic facts.
>
> That said, several challenges arise. First, training the predictor on entire complex queries may face issues with fuzzy logic aggregation, like those identified by van Krieken et al. Second, any change to the predictor would require recomputation of PE labels, which can be too expensive. Finally, tighter coupling may reduce the generality and reusability of the search policy, which is currently predictor-agnostic. To conclude, while joint training is technically feasible and may lead to interesting behavior, we expect it would come at the cost of stability, efficiency, and modularity.

---

### Review · Reviewer_tf7p · 2025-05-24

**Summary Of Contributions:**

In this paper, the authors identify answering conjunctive queries as a constraint satisfaction problem, tailoring the existing ANYCSP framework (2023) to suit the task of deciding the satisfiability of Boolean formulas over incomplete KGs. Currently, CQA is formulated as a ranking problem. The authors argue that the ranking-based formulation has significantly limited the progress in query answering over formulas with multiple free variables. To address the limitations of ranking-based formulation, the authors propose query answer classification (QAC) and query answer retrieval (QAR) to provide more targeted responses while ensuring scalability for more complex logical queries. Overall, the research perspective is innovative and the theoretical part is solid. From the perspective of evaluation, the authors develop new validation and test datasets on top of FB15k-237 and NELL. In particular, they increase the number of variables mentioned in the original datasets from 3 to between 12 and 20, resulting in queries with complexity exceeding the processing power of known approaches. Experimental results validate the effectiveness of the proposed method compared with an SQL engine implemented by DuckDB (2019) on the constructed new datasets.

**Audience:**

Yes

**Claims And Evidence:**

No

**Requested Changes:**

Major questions:
- Is there any evaluation on QAC processing complexity?
- In the bottom part of page 9, the authors “limit the processing time to 60 seconds, ensuring termination in a reasonable time”. Can you describe the reason for using 60 seconds as the threshold? Would it be too small?
- In Appendix D.4, why do you clip the predictor’s estimations to the range [0, 0.9999]?

Minor issues:
- Fix the presentation issue in the first paragraph of Appendix C.1, i.e., “. testing, respectively”.
- In Appendix C.4, the metric of measuring both the performance of query answer classification and query answer retrieval is denoted as $F1(\theta)$. Besides, the model $\theta$ does not appear in the right-hand side of the first equation of $F1(\theta)$.
- The format of right quotation marks in the second paragraph of Appendix D.5 should be corrected.
- Appendix E should be cited in the main text.
- Improve the layout of this manuscript to make it more readable. For instance, the citation of Table 2 appears on page 11, but the content of Table 2 is presented on page 9. Besides, many important details (e.g., the description of dataset constructions) are moved to the Appendix due to space limitations. These reduce the readability of the manuscript.

**Strengths And Weaknesses:**

Strengths:
- The paper is written clearly with good term definitions.
- The research perspective is innovative and the theoretical part is solid.

Weaknesses:
In contrast to the ranking-based strategy, the authors propose QAC and QAR tasks for CQA by constructing new evaluation datasets. Since the complexity of each new dataset exceeds the processing capacity of existing methods, only the results of QTO (2023), FIT (2024), GNN-QE (2022), and ULTRAQEURY (2024) on 2p, 3p, pi, ip, inp, and pin are presented in Table 1. All other results are limited to the proposed ANYCQ and an SQL engine implemented by DuckDB (2019). This results in several limitations of the evaluation part: First, the authors evaluate the performance of existing methods (i.e., QTO, FIT, GNN-QE, and ULTRAQEURY) on QAC, but those existing methods are all developed based on the ranking-based strategy. To fairly reflect the performance comparison of existing methods and ANYCQ, it is suggested to present the Mean Reciprocal Rank (MRR) and Hit at K (H@K) results of all models, using an evaluation protocol based on the ranking-based strategy. Second, apart from 2p, 3p, pi, ip, inp, and pin, there are also other queries such as 2i, 3i, 2u, up, and pni in the conventional CQA evaluation. The authors should provide the detailed reasons of using 2p, 3p, pi, ip, inp, and pin only. Third, since the experimental results of all existing methods are obtained by running them using an evaluation protocol based on QAC, it is suggested to release the code of evaluating baselines in your GitHub repository.

---

> ### Author Response · Authors · 2025-06-07
> **Response 1/2: Weaknesses**
>
> Thank you for your review and the time spent to assess our paper. Please find the explanations to the raised questions below:
>
> > To fairly reflect the performance comparison of existing methods and ANYCQ, it is suggested to present the Mean Reciprocal Rank (MRR) and Hit at K (H@K) results of all models, using an evaluation protocol based on the ranking-based strategy.
>
> We appreciate the reviewer’s suggestion. However, our primary motivation in introducing the QAC task is to highlight the limitations of the existing ranking-based paradigm in supporting classification-oriented objectives. While QTO, FIT, GNN-QE, and ULTRAQUERY are originally evaluated using MRR and H@K in prior work, the QAC task is fundamentally a binary classification problem: given a query $Q(x)$ and a candidate answer $a$, the goal is to determine whether $Q(a/x)$ is satisfied in the complete KG. As such, ranking metrics like MRR and H@K are not applicable in this setting. Moreover, applying ranking metrics in this context would not measure methods’ ability to make calibrated decisions - precisely the objective we aim to investigate. Therefore, we intentionally evaluate all models using classification metrics (F1-score) to fairly assess their performance in this new setup. This is in line with our objective to encourage research that goes beyond ranking and toward more decision-oriented reasoning.
>
>
> > Second, apart from 2p, 3p, pi, ip, inp, and pin, there are also other queries such as 2i, 3i, 2u, up, and pni in the conventional CQA evaluation. The authors should provide the detailed reasons of using 2p, 3p, pi, ip, inp, and pin only.
>
> As noted in **Appendix C.2**, query types without projections, such as 2i, 3i, 2in, 3in, and 2u, effectively reduce to pure link prediction (LP) under the QAC objective. For example, given a 2i query like $Q(x)=r(a,x)\land s(b,x)$ and an answer $c$, the goal is to decide if $Q(c/x)$ is satisfied, i.e. whether $r(a,c)$ and $s(b,c)$ hold. Including such queries would, hence, not meaningfully evaluate reasoning capabilities beyond link prediction. Additionally, query types with disjunctions, such as up, can be decomposed into independent subqueries under the QAC setting without introducing additional complexity (unlike in the ranking-based formulation), as described at the end of **Section 4**.
>
> Finally, our main focus lies in evaluating performance on structurally complex queries in the large x-hub splits. The smaller queries are included primarily to enable a fair comparison with prior methods - otherwise, SQL would be the only baseline we could compare against. We selected representative types to span projections (2p, 3p), conjunctions (ip, pi), and negations (inp, pin), which we believe provide sufficient coverage to support our claims. Following the reviewer’s feedback, we will extend the related part of **Appendix C.2** with a more detailed motivation behind our choice.
>
> > Third, since the experimental results of all existing methods are obtained by running them using an evaluation protocol based on QAC, it is suggested to release the code of evaluating baselines in your GitHub repository.
>
> For QTO, FIT, and UltraQuery, we follow the evaluation procedure introduced in the QTO repository, where the predictions of the underlying link predictors are pre-computed and stored in tensorized form. These are then evaluated on our QAC benchmarks using the provided `test_reasoners_qac.py` script. The tensorized predictions for QTO and FIT (based on ComplEx) are already publicly available through the repository link, and those for UltraQuery will be included in the non-anonymized version of our repository. We will also release the pre-computed tensorized predictions from GNN-QE’s NBFNet model, used to equip AnyCQ, as part of the final code release.
>
> For GNN-QE evaluation, we use its original codebase, applying minimal modifications to the evaluation script to accommodate our QAC benchmark format and compute classification metrics. Given the small scope of these changes, we believe it is unnecessary to duplicate the full GNN-QE codebase in our repository.

---

> > ### Comment · Reviewer_tf7p · 2025-06-08
> >
> > Thank you for the responses. In my view, the task type—classification, retrieval, or conventional ranking—determines the choice of metrics for a benchmark. However, I am confused by the apparent lack of an inherent connection between metric selection and query structure. For example, when a model predicts probabilities for all candidate answers to a query, QAR employs F1 scores (requiring prediction binarization), whereas ranking utilizes MRR or Hit@K against negative samples. Crucially, ranking evaluation remains applicable even for complex queries (e.g., 3-hub).
> > This relates to a broader technical distinction: While probability-based models (including AnyCQ) require threshold selection to binarize predictions, energy-based models (e.g., the link predictor ComplEx) must learn a real-valued threshold. Importantly, the need for binarization does not pose an insurmountable challenge for energy-based approaches. I contend that the comparative efficacy and fundamental relationship between these paradigms constitute an open research problem.
> > Please correct me if I misinterpreted any technical aspects.

---

> > > ### Author Response · Authors · 2025-06-08
> > >
> > > We thank the reviewer for their thoughtful follow-up and would like to clarify several key points.
> > >
> > > > when a model predicts probabilities for all candidate answers to a query, QAR employs F1 scores...
> > >
> > > Please note that in the QAR setting, models are *not* required to predict probabilities for all candidate answers. Instead, they must return a single assignment to the free variables (or None, if no answer exists). The evaluation then compares this single prediction against the ground truth. Thus, our F1 score effectively reflects Hit@1, augmented by the necessity to make a binary decision on whether any correct answer exists. This is quite different from evaluating over a ranked list or thresholded score vector.
> > >
> > > > Crucially, ranking evaluation remains applicable even for complex queries (e.g., 3-hub).
> > >
> > > We respectfully disagree. While ranking metrics can, in principle, be applied to any query structure, they become computationally infeasible in practice when the number of free variables increases. Our 3-hub queries often involve two or three free variables, which under a classical ranking paradigm would require scoring $O(|V(G)|^2)$ or $O(|V(G)|^3)$ candidate tuples, respectively. This scalability issue has already been identified in prior work, such as [1], where even two-variable queries required approximation techniques or marginal scoring. As discussed in **Section 4**, such limitations make ranking-based evaluation inappropriate for high-arity queries, which is a central motivation behind our classification-based reformulation.
> > >
> > > > While probability-based models (including AnyCQ) require threshold selection to binarize predictions, energy-based models (e.g., the link predictor ComplEx) must learn a real-valued threshold.
> > >
> > > This is partially true: energy-based models can indeed be paired with a real-valued threshold. However, in our early experiments, we found that using a fixed threshold across all relations for ComplEx led to poor performance. This is due to varying score distributions across relation types. For instance, scores for $r(a,x)$ might lie in $[0,6]$, while for $s(b,x)$ they may fall in $[−4,2]$. A score of 2 in the former could indicate a weak match, while in the latter, it may be highly confident. Because energy-based models like ComplEx are not calibrated and trained only to optimize relative rankings, a single decision threshold does not generalize well across relations. On another hand, defining a separate threshold for each relation is not trivial itself, and raises more questions on stability.
> > >
> > > To address this, QTO and FIT introduce score-to-probability transformations for energy-based models (as described in **Appendix D.1**). While this calibration improves performance, we find that such methods still fall short of approaches like NBFNet and Ultra (see **Appendix D.5**), which natively produce probabilities and are better suited to binary classification tasks. Therefore, while we agree that binarization is not fundamentally impossible for energy-based approaches, in practice, it remains a non-trivial challenge, especially for tasks requiring calibrated and consistent outputs. In this sense, we agree with the reviewer that adapting energy-based methods to decision-focused objectives like QAC or QAR is an open and important research direction, but we maintain that their current limitations justify our decision to use classification metrics throughout the paper.
> > >
> > > ---
> > >
> > > [1] Yin, Hang, et al. "EFOk-CQA: Towards Knowledge Graph Complex Query Answering beyond Set Operation."

---

> ### Author Response · Authors · 2025-06-07
> **Response 2/2: Requested Changes**
>
> > Is there any evaluation on QAC processing complexity?
>
> We thank the reviewer for this question. The complexity analysis presented in **Section 6.4** applies uniformly to both QAR and QAC, as AnyCQ’s runtime depends solely on the query size, not the number of free variables. In fact, a QAC instance $\langle Q(x), c \rangle$ can be viewed as a special case of QAR, with zero free variables. Empirically, our runtime evaluations on QAR already cover queries of varying sizes and structures, and thus fully capture the behavior expected for QAC as well. For this reason, we omitted separate complexity plots for QAC from the manuscript.
>
> > In the bottom part of page 9, the authors “limit the processing time to 60 seconds, ensuring termination in a reasonable time”. Can you describe the reason for using 60 seconds as the threshold? Would it be too small?
>
> The 60-second threshold was chosen to balance two considerations. First, it is intended to prevent classical query answering approaches, like FIT, from exhaustively enumerating assignments over multiple free variables, which is computationally infeasible at scale and diverges from the intended reasoning task. Second, we wanted the timeout to be sufficiently large to ensure that when the SQL engine fails to return an answer within this window, it reflects genuine difficulty in solving the query rather than an overly strict time constraint. We manually verified that queries exceeding this threshold typically involve substantial structural complexity. With that in mind, the exact value of 60 seconds is a design choice - we could as well have chosen it to be 30, 120, or 240 seconds, and adapt accordingly.
>
> Ultimately, for any chosen threshold, it would be possible to construct even more challenging queries by increasing their size or the number of free variables, which would disproportionately impact the performance of classical engines like SQL, while only marginally affecting AnyCQ. However, our goal was not to tailor the benchmark to favor our method, and the current setup offers a fair and principled framework for evaluating scalability. By design, it emphasizes the core challenge of the QAR task: efficiently retrieving correct answers to complex queries that traditional engines struggle to handle.
>
> > In Appendix D.4, why do you clip the predictor’s estimations to the range [0, 0.9999]?
>
> We apply this clipping to distinguish between observed and inferred relations. Specifically, facts present in the observable knowledge graph are assigned a score of exactly 1, while predicted (i.e., unobserved) links are capped at 0.9999. This distinction ensures that, when evaluating Boolean formula scores, any assignment yielding a score of 1 must consist entirely of observed facts and is therefore guaranteed to be correct.
>
> This becomes particularly important in the QAR task, where the model must return a single answer. When multiple assignments yield similar scores, this scheme enables AnyCQ to break ties in favor of assignments that rely only on observed data. As a result, the model prefers verifiable answers from the graph over those that depend on uncertain predictions.
>
> > Fix the presentation issue in the first paragraph of Appendix C.1, i.e., “. testing, respectively”.
> > The format of right quotation marks in the second paragraph of Appendix D.5 should be corrected.
> > Appendix E should be cited in the main text.
>
> Thank you for noticing these inconsistencies. The formatting issues will be fixed, and **Appendix E** will be mentioned in **Section 6.4** as a further validation for the study of AnyCQ’s time complexity.
>
> > In Appendix C.4, the metric of measuring both the performance of query answer classification and query answer retrieval is denoted as $F1(\theta)$. Besides, the model $\theta$ does not appear in the right-hand side of the first equation of $F1(\theta)$.
>
> Thank you for pointing this out. We will denote the F1-score evaluation functions used for QAC and QAR as $F1_{QAC}$ and $F1_{QAR}$, respectively, to avoid unclear repetitions. As for the first equation, we indeed first introduce the set of answers to $Q(x)$  returned by the model $\theta$ as $A(\theta, Q)$, and then inconsistently call it $A_Q$ in the formula. We will correct this in the updated manuscript.

---

> > ### Comment · Reviewer_tf7p · 2025-06-08
> >
> > Thank you for the responses to my requested changes.

---

### Review · Reviewer_uUPs · 2025-05-26

**Summary Of Contributions:**

The paper addresses two main issues. First, unlike traditional machine learning approaches that rely on classification metrics, current CQA methods use ranking-based metrics such as MRR and Hits@k. The authors argue that these metrics are not well-suited for evaluation of CQA and instead introduce two alternative settings— **query answering classification** and **query answering retrieval**. Second, existing methods are limited in the types of queries they can handle, typically supporting only tree-structured queries with a single target variable. To address this limitations, the authors propose a new method called ANYCQ, able to answer any conjunctive query, along with new benchmarks that include queries with multiple target variables and cycles.

**Audience:**

Yes

**Broader Impact Concerns:**

No ethical concern

**Claims And Evidence:**

No

**Requested Changes:**

Suggested actions for addressing each of the weaknesses listed above, with the most critical being 1, 4, and 5.

1. Run ANYCQ on some of the existing benchmarks mentioned above.
2. Provide more details on the 3-4-5 hub queries in the main text (see my questions above), including a visual example of the queries. I suggest adding more details on the design choice for the new benchmark in the main text.
3. Include a discussion for queries without constant anchor nodes. Does ANYCQ allow such queries? If so, corresponding queries should be added to the evaluation.
4. Statements about easy answers are confusing and need additional explanations. Why including easy answers in the evaluation?
5. Can the authors detail more about the reason why the existing methods cannot be used for the QAR task? In page 9, why do you require calibration of the score without knowing the easy answer? From the observed graph, we can always retrieve such answers, so maybe at least FIT could be used in QAR setting? Can the authors provide more details about the choice of limiting the processing time of SQL to 60 seconds, and provide additional results with larger and smaller timeout?
 6. See my question above

**Strengths And Weaknesses:**

### Strenghts:

1. The idea of having ***alternative settings for evaluating CQA is interesting***, as it offers a different perspective for evaluation.
2. ***Explicit problem definition in section 4.2*** is rigorous and is crucial to understand the new settings introduced by the authors.
3.  By only changing the pre-trained link predictor used by ANYCQ, the ***model can transfer to any knowledge graph***. Moreover, it inherits the inductive capabilities of NbfNet and Ultra

### Weaknesses:

1. ***Limited experiments on existing benchmarks.*** The authors did not compare ANYCQ for all query types from [5], i.e. 2i,3i,2u,up, 2in,3in, pni queries are missing. Why did the authors not consider such query types? Also, since AnyCQ can be executed on any conjunctive query, the authors should consider running their method on more recent benchmarks, as some of those already addressed some of the limitations mentioned in the paper e.g.,  [1] contains EFO queries with up to two target variables, [2] contains queries allowing cycles, [3] contains DAG queries, and [4] propose benchmarks built to comprise the full spectrum of hardness.

2. ***Limited explanation of the newly proposed query structures and benchmarks.*** As far as I understand, the new queries proposed in the paper are the 3-4-5-hub. What is intended with the term ‘’hub’’? Why QAR and QAC benchmarks are different? Only QAR hub-queries have multiple target variables? For the existing query structure (e.g., 2p,3p, etc), did you re-use the query-answers already sampled in [5], or did you include easy answers in the evaluation?
3. ***No discussion of queries without anchor nodes.*** The authors argue that ANYCQ works for any conjunctive query, but I could not find a discussion for queries without constant anchor nodes. (e.g. (?x,r,?y)).
4. ***Limited explanation on the need of considering easy answers in the evaluation.*** In page 9 the authors mention ‘’…CQA benchmarks admit easy answers …. We hence develop new benchmarks…’’. Following [5], easy answers are answers directly retrievable from the observed graph, and as such, are ***filtered out*** from existing CQA benchmarks. Then, in page 10, the authors mention ‘’ In contrast with CQA evaluation, we do not distinguish between easy and hard answers…”. Those two statements are in contradiction and create confusion. Finally, the authors motivate the addition of the easy answer in their benchmarks stating that easy answers are not trivial, ‘’as demonstrated in the experiments’’, but I could not find a direct experiment motivating the need of including easy answers in the evaluation of a ML model. Such answers can be retrieved with 100% accuracy without a ML model and should not be included in the test set.
5. ***SQL not convincing as a baseline.*** In table 2,3,4, reporting SQL as the only competitor to AnyCQ is not convincing, as it will perform 100% precision/recall for easy answers (unless the timeout plays a role here) and 0% on hard answers; then depending on their proportion, AnyCQ will be outperforming SQL if there are more hard answers, and SQL will outperform AnyCQ otherwise.
6. ***Unclear how the ‘query answering retrieval’ task works when there are multiple answers to a query***

### Minor notes/questions:

- In complex query answering related work, the authors write “QTO achieves superior performance by training directly over queries, without pre-trained embedding modes”, which is confusing, as QTO is based on a pre-trained link predictor model.
- In 6.2 the authors write ‘’as expected, GNN-QE and ULTRAQUERY outperform QTO’’. As in the existing ranking setting QTO outperforms both methods for FB15k237 and NELL995, is there a specific reason for which the authors expected this result?
- In table 1, why in your opinion ANYCQ outperforms GNNQE in NELL, but not in FB15k237? As both GNN-QE and ANYCQ are equipped with NBFNet, what is making a difference?
- In D.4 the authors show that they incorporate the observed graph by setting the score of training links=1, as done in QTO [6], FIT [2], and CQD-hybrid [4]. Do you do so in both settings? Can you run an ablation experiment without such feature? As far as I know GNNQE does not have this feature, and I’d like to see a direct comparison between GNNQE and ANYCQ without this feature enabled.

[1] Yin, Hang, et al. "$\text {EFO} _ {k} $-CQA: Towards Knowledge Graph Complex Query Answering beyond Set Operation." arXiv preprint arXiv:2307.13701 (2023).

[2] Yin, Hang, Zihao Wang, and Yangqiu Song. "Rethinking Complex Queries on Knowledge Graphs with Neural Link Predictors." The Twelfth International Conference on Learning Representations.

[3] He, Yunjie, et al. "Dage: Dag query answering via relational combinator with logical constraints." Proceedings of the ACM on Web Conference 2025. 2025.

[4] Gregucci, Cosimo, et al. "Is Complex Query Answering Really Complex?." arXiv preprint arXiv:2410.12537 (2024).

[5] Ren, Hongyu, and Jure Leskovec. "Beta embeddings for multi-hop logical reasoning in knowledge graphs." Advances in Neural Information Processing Systems 33 (2020): 19716-19726.

[6] Bai, Yushi, et al. "Answering complex logical queries on knowledge graphs via query computation tree optimization." International Conference on Machine Learning. PMLR, 2023.

---

> ### Author Response · Authors · 2025-06-07
> **Response 1/n: Query types**
>
> > The authors did not compare ANYCQ for all query types from [5], i.e. 2i,3i,2u,up, 2in,3in, pni queries are missing. Why did the authors not consider such query types?
>
> Our broader goal in this work is to move beyond the limitations of traditional CQA benchmarks and introduce a more challenging and realistic setup for reasoning over knowledge graphs. We focus on structurally complex queries and adopt a classification-based objective, which we believe is better suited for evaluating a model's ability to make reliable decisions in the presence of missing information. In particular, we aim to study query types that go beyond what can be solved with simple link prediction or lightweight structural heuristics, and that more accurately reflect the types of reasoning needed in practical settings.
>
> That said, to maintain continuity with prior work, we include a subset of small-query types in our QAC benchmarks. These were selected to cover a range of key logical constructs (projection, conjunction, and negation) and to allow for direct comparison with existing CQA methods under the new classification setting. Without them, SQL would be the only viable baseline. We chose not to include other types for several reasons. First, query types without projections (e.g., 2i, 3i, 2u, 2in, 3in) reduce to pure link prediction under QAC. For instance, a 2i query like $Q(x) = r(a,x) \land s(b,x)$, when paired with an answer $c$, becomes a test of whether both $r(a,c)$ and $s(b,c)$ hold, i.e., two LP queries. These do not meaningfully evaluate reasoning capabilities beyond atomic fact prediction. Second, as noted at the end of **Section 4**, disjunctions can be evaluated by decomposing the query into independent subqueries, which introduces little additional complexity and makes them less informative in this context.
>
> In summary, our evaluation focuses on query types that are most relevant to the goals of this paper: scalable and interpretable reasoning over structurally complex queries. We will revise **Appendix C.2** to make this motivation clearer and to justify our selection of query types more explicitly.
>
> > As far as I understand, the new queries proposed in the paper are the 3-4-5-hub. What is intended with the term ‘’hub’’?
>
> The term “hub” in our benchmark refers to a set of nodes used in the query generation process, as detailed in **Appendix C.1**. Concretely, we first randomly select a central entity $z$, and then sample $k$ additional entities from the 2-hop neighborhood of $k$; we refer to these $k+1$ nodes as “hubs.” We then construct a subgraph induced by the 1-hop neighborhoods of each hub. This subgraph is used to synthesize a complex logical query. The term “k-hub” reflects the number of such hubs involved in the query generation. Increasing $k$ typically leads to more densely connected and structurally intricate queries, deviating further from the tree-like patterns found in earlier benchmarks.
>
> > Why QAR and QAC benchmarks are different?
>
> QAC benchmarks additionally include small-query splits (e.g., 2p, 3p), which are intentionally included to establish continuity with the CQA literature and to highlight the limitations of existing embedding-based methods in reliably classifying answers, even for relatively simple query structures. In contrast, QAR benchmarks omit these small-query types and focus exclusively on structurally complex queries. This is because, in QAR, retrieving any correct answer is sufficient, and small queries tend to have easy answers in the observed knowledge graph. Recovering such answers is often trivial and does not provide meaningful insight into a model’s reasoning capabilities. As a result, we designed QAR to challenge models on more demanding queries, where retrieving even observable answers is nontrivial.
>
> > Only QAR hub-queries have multiple target variables?
>
> Yes, only QAR queries in our benchmarks involve multiple free variables. This is by design, and is based on a key property of the QAC setting. In a QAC instance $\langle Q(x), a \rangle$, the task is to determine whether the candidate answer $a$ satisfies the query. If $x$ includes multiple free variables, the full assignment $a$ grounds all of them. As a result, the query effectively becomes Boolean after grounding: we evaluate whether $Q(a/x)$ is satisfied. In particular, a QAC instance of the form $\langle Q(x_1, x_2), (a_1, a_2) \rangle$ is equivalent to an instance $\langle Q(x_1, a_2/x_2), a_1\rangle$. This reduction makes the number of free variables in the original query irrelevant for the QAC task, and for this reason, we restrict our QAC benchmarks to queries with a single free variable for simplicity and consistency.
>
> In contrast, QAR explicitly requires returning an assignment for the free variables, which may include multiple targets. Therefore, the complexity associated with handling multiple free variables arises only in the QAR setting.

---

> ### Author Response · Authors · 2025-06-07
> **Response 2/n: Benchmarks**
>
> > **CH1**: Run ANYCQ on some of the existing benchmarks mentioned above.
>
> We appreciate the reviewer’s suggestion and agree that recent benchmarks have begun to explore important limitations in prior CQA datasets. However, our position is that while each of these benchmarks addresses specific structural or modeling challenges, none provides a unified setting that enforces all of them simultaneously at a scale sufficient to expose the practical limitations of current methods. Our goal in this work is precisely to introduce such a setting and to demonstrate that ANYCQ can scale to meet this broader challenge.
>
> Existing benchmarks are limited to queries with at most 4 variables, limiting structural complexity. While [1] makes the important point that many benchmark queries can be reduced to simpler forms, it does not go further to propose structurally richer or larger queries. In contrast, our large-query splits involve between 10 and 20 variables, closer in spirit to the types of queries that real-world systems must eventually handle.
>
> Secondly, although EFOk-CQA [2] includes queries with up to two free variables, its evaluation strategy relies on either marginal approximations or enumeration over one of the variables. This makes such queries tractable even with limited architectural support. To challenge this, our benchmarks introduce queries with up to three free variables, raising the naive enumeration complexity to $O(|V(G)|^2)$ , thus preventing shortcut solutions via partial enumeration.
>
> Third, while FIT [3] supports cyclic queries, it handles them by enumerating over variable assignments to break cycles. This may suffice for shallow or sparsely connected queries, but as the number of cycles grows, the combinatorial cost becomes infeasible. In our benchmark, we include multi-cycle queries specifically to demonstrate how enumeration scales poorly in such cases and to highlight the need for truly scalable reasoning strategies.
>
> Finally, multi-edges, as in DagE [4], are trivially supported by existing fuzzy logic models via element-wise matrix operations and do not present significant additional difficulty.
>
> In contrast to these prior works, our benchmarks combine all of these challenges at once. The focus of this paper is not to compare performance on previously studied queries, but to motivate and evaluate a system capable of handling a much more demanding class of reasoning tasks. For this reason, we do not believe additional experiments on other benchmarks are necessary to support our core claims. We leave a precise study on CQA approaches’ classification abilities for future work, beyond the scope of this paper.
>
> [1] Gregucci, Cosimo, et al. "Is Complex Query Answering Really Complex?." \
> [2] Yin, Hang, et al. "EFOk-CQA: Towards Knowledge Graph Complex Query Answering beyond Set Operation."\
> [3] Yin, Hang, Zihao Wang, and Yangqiu Song. "Rethinking Complex Queries on Knowledge Graphs with Neural Link Predictors."\
> [4] He, Yunjie, et al. "Dage: Dag query answering via relational combinator with logical constraints."
>
> > For the existing query structure (e.g., 2p,3p, etc), did you re-use the query-answers already sampled in [5], or did you include easy answers in the evaluation?
>
> For the small query splits in QAC benchmarks, we used both easy and hard answers. Table 9 in the appendix provides a breakdown of this distribution, showing that a substantial portion of the instances are indeed hard, requiring models to infer missing links.
>
> > **CH2**: Provide more details on the 3-4-5 hub queries in the main text, including a visual example of the queries.
>
> Following the reviewer’s suggestion, we will extend the description of the sampled queries in Section 6.1 and include a figure (part of the current Figure 6.) displaying the structures of our queries in the main body.
>
> > **CH3**: Include a discussion for queries without constant anchor nodes.
>
> We appreciate the reviewer’s observation. In the traditional CQA literature, “anchor nodes” typically refer to constants at the leaves of the query graph that help ground the query in the observed KG. Many large queries in our benchmarks, including those shown in **Figure 6**, contain leaves that are existential variables, not constants. AnyCQ is designed to operate over arbitrary conjunctive queries, regardless of whether they contain constant anchor nodes. The construction of the computational graph and the variable assignment search mechanism do not rely on the presence of constants. Queries composed entirely of variables are handled without any special treatment; the model performs a search over all grounded variable assignments, guided by the link predictor, just as it would for queries with constants
>
> We agree that this capability is implicit in the framework, and we hope that including the aforementioned figure displaying queries from our large splits will clarify the range of applicable query structures.

---

> ### Author Response · Authors · 2025-06-07
> **Response 3/n: Easy answers & SQL**
>
> > In page 9 the authors mention ‘’…CQA benchmarks admit easy answers …. We hence develop new benchmarks…’’.\
> > Then, in page 10, the authors mention ‘’ In contrast with CQA evaluation, we do not distinguish between easy and hard answers…”. Those two statements are in contradiction and create confusion.
>
> When we state that “most of the simple instances inherited from CQA benchmarks admit easy answers,” we are referring to the fact that, for many queries $Q(x)$ in those benchmarks, there *exists* at least one easy answer $a$. That is, these queries *admit* easy answers, even though such answers are excluded during evaluation in the original CQA setting, which focuses exclusively on hard (inferred) answers. While there exist queries with only hard answers, a vast majority of them has answers in the observable graph as well. We acknowledge that the wording on page 9 may have caused confusion, and we will revise that sentence to more accurately reflect our intent.
>
> > **CH4**: Why including easy answers in the evaluation?\
> > Finally, the authors motivate the addition of the easy answer in their benchmarks stating that easy answers are not trivial, ‘’as demonstrated in the experiments’’\
> > Such answers can be retrieved with 100% accuracy without a ML model and should not be included in the test set.
>
> We agree that, in theory, answers supported by the observable knowledge graph ("easy answers") can be retrieved with perfect accuracy using classical systems. However, the key issue is not *whether* these answers can be found, but how efficiently this can be done in practice, particularly for complex queries. Satisfiability of Boolean conjunctive queries is NP-complete, and as query size and structure grow, classical engines like SQL become increasingly inefficient. This is demonstrated in **Table 3**, where SQL frequently fails to return correct observable answers within the specified time. This motivates our claim that retrieving easy answers is far from trivial in structurally complex settings.
>
> From a practical perspective, especially in downstream applications like KG-based retrieval-augmented generation (RAG), the distinction between easy and hard answers is irrelevant to the end user. What matters is whether an answer can be retrieved quickly and whether it is correct. Since conjunctive query answering is NP-complete, any efficient algorithm, whether symbolic or learned, cannot be expected to consistently retrieve correct answers, even when they are fully observable in the KG. Therefore, we do not believe test sets should exclude observable answers; rather, benchmarks should reflect the real-world need for fast, scalable reasoning regardless of answer type.
>
> > **CH5**: Can the authors provide more details about the choice of limiting the processing time of SQL to 60 seconds, and provide additional results with larger and smaller timeout?
>
> The 60-second timeout was chosen to strike a balance between two goals. First, it prevents classical approaches like FIT or SQL from relying on exhaustive enumeration, which becomes infeasible in large, complex queries. Second, it is sufficiently generous to ensure that timeouts reflect genuine computational difficulty, not an artificially strict limit. We verified that queries on which SQL exceeds this threshold involve substantial structural complexity.
>
> That said, the exact value is ultimately a design choice - alternative thresholds (e.g., 30, 120, or 240 seconds) could be used. Ultimately, for any chosen threshold, it would be possible to construct even more challenging queries by increasing their size or the number of free variables, which would impact the performance of classical engines like SQL, while only marginally affecting AnyCQ.
> However, our aim was not to tune the benchmark to favor AnyCQ, but to define a principled evaluation setup that exposes scalability limitations in classical engines. However, our goal was not to tailor the benchmark to favor our method, and the current setup offers a fair and principled framework for evaluating scalability. For this reason, we do not believe additional evaluations with alternative timeouts are necessary. The current setup already highlights the core challenge posed by QAR: retrieving correct answers efficiently for structurally complex queries where classical engines fail to scale.

---

> ### Author Response · Authors · 2025-06-07
> **Response 4/n: Baselines & minor questions**
>
> > **CH5**: Can the authors detail more about the reason why the existing methods cannot be used for the QAR task?
>
> The QAR task introduces several challenges that make most existing CQA methods inapplicable or impractical:
> - Support for cyclic queries: Many methods, including BetaE, Query2Box, CQD-Beam, ConE, QTO, and GNN-QE, are designed for tree-like queries. As QAR benchmarks include cyclic structures, these approaches cannot be directly applied.
> - Handling multiple free variables: Most neural CQA models (e.g., Q2T, LMPNN, CLMPT) are designed to embed queries with a single free variable. They compute an embedding for the query and rank all entities based on similarity to this embedding. Extending these approaches to handle multiple free variables would require infeasible enumeration over joint assignments, or non-trivial architectural changes. This point is, in fact, not limited to neural approaches, but true for essentially all existing CQA methods.
> - Lack of calibrated outputs: The QAR task requires identifying a correct assignment or confidently returning None if no valid answer exists. This calls for well-calibrated, probabilistic predictions. Many existing models are trained for ranking tasks and do not produce outputs interpretable as calibrated probabilities, making them ill-suited for threshold-based decision-making.
>
> FIT is more general than many earlier approaches in that it can, in principle, process cyclic queries. However, it does so by enumerating over variable assignments to break cycles. In the same way, it can reduce formulas with multiple free variables to a series of single-variable samples. While this is feasible for small queries or limited search spaces (as in the EFO-k or original FIT benchmarks), such enumeration becomes computationally intractable in our setting, which includes queries with large diameters, several cycles and free variables. To ensure a meaningful evaluation, we design our QAR benchmarks to explicitly prevent such enumeration-based “shortcuts.” This setup reveals the practical limitations of existing symbolic methods like FIT and highlights the need for more scalable strategies.
>
> > **CH6**: Unclear how the ‘query answering retrieval’ task works when there are multiple answers to a query
>
> Thank you for pointing this out. We agree that this aspect could be more explicitly described and will revise **Appendix C.4** accordingly.
>
> In the QAR task, the objective is: given a query $Q(x)$, return a single assignment $a$ to the free variables $x$ such that $Q(a/x)$ is satisfied in the complete KG, or return None if no such assignment exists. The model is only required to produce one correct answer per query, even when multiple valid answers may exist. Evaluation is based on the correctness of this single output. If the returned assignment satisfies the query, it is counted as correct. Similarly, if the model returns None and no satisfying assignment exists in the complete KG, this is also considered correct.
>
> > In 6.2 the authors write ‘’as expected, GNN-QE and ULTRAQUERY outperform QTO’’. As in the existing ranking setting QTO outperforms both methods for FB15k237 and NELL995, is there a specific reason for which the authors expected this result?
>
> We appreciate the reviewer’s observation and are happy to clarify. Our expectation was based on the difference in how the underlying link predictors are trained. QTO uses a ComplEx-based link predictor, which outputs uncalibrated similarity scores not inherently confined to the [0,1] range. In contrast, both NBFNet and Ultra (used by GNN-QE and ULTRAQUERY, respectively) are trained using binary cross-entropy loss, and thus produce calibrated outputs that approximate probabilities. For these reason, we predicted them to perform better than ComplEx-based approaches, and this belief was indeed validated by the experiments.
>
> > In table 1, why in your opinion ANYCQ outperforms GNNQE in NELL, but not in FB15k237? As both GNN-QE and ANYCQ are equipped with NBFNet, what is making a difference?
>
> We believe the observed difference arises from how NBFNet is used in each framework.
> In GNN-QE, NBFNet is employed to simulate a projection over all candidate entities simultaneously, enabling efficient vectorized inference. This means that the predictor needs to derive the probability of each entity being a tail of the relation, given a list of potential heads (each assigned its own probability).
> In contrast, AnyCQ queries the link predictor only on grounded atomic facts of the form $r(a,b)$, making local decisions rather than relying on global inference. This setting may allow NBFNet to produce more reliable predictions, without any noise, especially in large, sparsely connected graphs like NELL.
>
> > In D.4 the authors show that they incorporate the observed graph by setting the score of training links=1, as done in QTO [6], FIT [2], and CQD-hybrid [4]. Do you do so in both settings?
>
> Yes, this is done for both QAC and QAR.

---

> > ### Comment · Reviewer_uUPs · 2025-06-09
> >
> > > For instance, a 2i query like $Q(x) = r(a,x) \land s(b,x)$, ... These do not meaningfully evaluate reasoning capabilities beyond atomic fact prediction.
> >
> > I disagree. In fact, as shown in [1], the high performance of 2i/3i queries was the result of an high percentage of queries being reduced to an easier types. If you check the performance of existing models on the benchmarks proposed in [1], you'll see that the performance on 2i are much lower than 1p, showing that even 2i queries are important for evaluating reasoning capabilities. The perception of 2i/3i queries being easier is the result of the inflation in performance found in the benchmark from the BetaE paper [2] due to such reductions. ***My overall suggestion is to use the benchmarks and the query types proposed in [1] for both tasks you propose.***
> >
> > > This is because, in QAR, retrieving any correct answer is sufficient, and small queries tend to have easy answers in the observed knowledge graph.
> >
> > I still do not understand why you have different query types in the two tasks. Moreover, ***you should not consider easy answer in your evaluation, as they would just inflate performance.***
> >
> > >Yes, only QAR queries in our benchmarks involve multiple free variables.
> >
> > Why did you make this desing choice? To me, having different query types in the two tasks create unnecessary confusion.
> >
> > > In contrast to these prior works, our benchmarks combine all of these challenges at once.
> >
> > I disagree, as considering easy answers in your benchmark goes in the opposite direction of [1]
> >
> > > However, our aim was not to tune the benchmark to favor AnyCQ
> >
> > The timeout is likely playing a role in your results, even if you did not tune it. You should highlight such role providing experiments with different timeouts.
> >
> > > FIT is more general than many earlier approaches in that it can, in principle, process cyclic queries.
> >
> > If I understood correctly, your QAR task is not only about cyclic queries, rather assessing the correctness of the top-ranked result, which is independent of the query type. Hence, as said above, my suggestion is to include the query types in [1] for both tasks, in such a way you can have a comparison of some query types with other methods too.
> >
> > > The model is only required to produce one correct answer per query, even when multiple valid answers may exist. Evaluation is based on the correctness of this single output. If the returned assignment satisfies the query, it is counted as correct.
> >
> > If I understand correctly, you're only considering top-1 ranking. I think this task could be highly influenced by the number of answers of a query; if a query has a single answer it might be harder to retrieve it. Do you have any additional study on this aspect? Moreover, since you're evaluating on easy answers too, ***the results on this task might be highly inflated by them***, as the model will likely output an easy answer rather than an hard answer, if there is one.
> >
> > >In D.4 the authors show that they incorporate the observed graph by setting the score of training links=1, as done in QTO [6], FIT [2], and CQD-hybrid [4]. Do you do so in both settings?
> > >>Yes, this is done for both QAC and QAR.
> >
> > I also have asked you
> > >As far as I know GNNQE does not have this feature, and I’d like to see a direct comparison between GNNQE and ANYCQ without this feature enabled.
> >
> > This is crucial to see the actual difference in performance between GNNQE and ANYCQ. Do you have any insight on this?
> >
> > ___
> >
> > [1] Gregucci, Cosimo, et al. "Is Complex Query Answering Really Complex?." arXiv preprint arXiv:2410.12537 (2024).
> >
> > [2] Ren, Hongyu, and Jure Leskovec. "Beta embeddings for multi-hop logical reasoning in knowledge graphs." Advances in Neural Information Processing Systems 33 (2020): 19716-19726.

---

> > > ### Author Response · Authors · 2025-06-10
> > >
> > > We appreciate the reviewer’s comments and would like to clarify several concerns which, as we understand, stem in large part from a disagreement with our decision to include easy answers in evaluation. We address that design choice separately in the second response. Here, we focus on justifying the structure of our benchmarks, the selected query types, and model comparisons, assuming the inclusion of easy answers as part of the task definition.
> > >
> > > > My overall suggestion is to use the benchmarks and the query types proposed in [1] for both tasks you propose.
> > > > I still do not understand why you have different query types in the two tasks.
> > >
> > > Thank you for the suggestion to unify the benchmarks across tasks. However, given that easy answers are included in our evaluation, the distinction between query types in QAC and QAR is intentional and grounded in the design of the tasks.
> > >
> > > In QAC, the input is a pair $(Q(x), a)$ and, the goal is to decide whether $Q(a/x)$ is satisfied. Consider $Q(x)=r(b,x)\land s(c,x)$. The objective is to decide if $Q(a/x)=r(b,a) \land s(c,a)$ holds, which is equivalent to “Does $r(b,a)$ hold and does $s(c,a)$ hold?”. Hence, this instance is equivalent to two LP instances: $r(b,a)$ and $s(c,a)$. Note that in the ranking formulation, one still needs to generate a combined score for Q(x), and hence queries like “2i” or “3i” are harder than a singular “1p”. In the classification setting, no score aggregation is needed: the answer is simply the binary conjunction of atomic link predictions. This makes such queries equivalent to multiple independent LP tasks and uninformative for evaluating reasoning capabilities. For this reason, query types without projections (e.g., 2i, 3i, 2u) are excluded from QAC.
> > >
> > > Moreover, including multiple free variables in QAC brings no benefit. A QAC instance $(Q(x_1,x_2), (a_1, a_2))$ with two free variables is equivalent to $(Q(x_1, a_2/x_2), a_1)$, as they both ask if $Q(a_1/x_1, a_2/x_2)$ is true. Thus, such cases collapse into the single-variable setting, as all free variables are grounded anyway.
> > >
> > > In QAR, small queries (like 2p or 3p) are typically easy - retrieving any correct answer is often trivial, especially with observed facts. Most queries from existing benchmarks indeed admit easy answers. While it would be possible to include small queries with only hard answers, our aim is not to study classification accuracy in isolation, but to introduce a setting that pushes the limits of existing systems. As discussed in **Section 4**, we believe classification is a more scalable objective than ranking in high-arity settings. Investigating the classification performance of existing CQA approaches (e.g. on the full transformed BetaE datasets) is a valuable direction, but one that deserves a dedicated and focused study. A thorough analysis of this kind falls outside the scope of the present work.
> > >
> > > For the reasons stated above:
> > > our QAC benchmarks exclude queries without projections, as they reduce to LP
> > > our QAC benchmarks exclude multiple free variables, as they do not change the decision task
> > > our QAR benchmarks exclude small queries, as they are often trivial under the objective.
> > >
> > > > I disagree, as considering easy answers in your benchmark goes in the opposite direction of [1]
> > >
> > > Our benchmarks combine multiple challenges: cycles, multi-edges, multiple free variables, and non-anchored leaves, within structurally complex queries. Many of these queries do not admit easy answers. Including easy answers in evaluation does not change the fact that all these structural challenges are present simultaneously.
> > >
> > > > The timeout is likely playing a role in your results [...] You should highlight such role providing experiments with different timeouts.
> > >
> > > We agree that the timeout setting affects the relative performance of SQL. We will include experiments with varied timeouts in the revised appendix to better illustrate how SQL performance degrades with increasing complexity.
> > >
> > > > since you're evaluating on easy answers too, the results on this task might be highly inflated by them
> > >
> > > Indeed, queries with easy answers tend to yield higher retrieval accuracy. This is directly visible in **Table 3**, where we report results disaggregated by easy vs. hard cases.
> > >
> > > > As far as I know GNNQE does not have this feature, and I’d like to see a direct comparison between GNNQE and ANYCQ without this feature enabled.
> > >
> > > The reviewer is correct that the original GNN-QE paper does not explicitly mention reasoning over the observed graph. However, the official implementation does include this feature. In particular, the model includes logic for checking if a predicted answer is already supported in the observable graph:
> > > https://github.com/DeepGraphLearning/GNN-QE/blob/master/gnnqe/model.py#L169
> > > As such, the GNN-QE results we report are for the version with observable-graph reasoning enabled, and we believe this comparison to AnyCQ is fair.

---

> > > ### Author Response · Authors · 2025-06-10
> > >
> > > One of the reviewer’s central concerns is our decision to include easy answers in the evaluation. We understand this choice diverges from the CQA conventions and appreciate the opportunity to clarify our reasoning. Below, we outline the motivation behind this decision, closely tied to the direction of our work: designing benchmarks built around structurally complex queries that cannot be efficiently solved by classical engines.
> > >
> > > ---
> > >
> > > In the long term, we believe the ultimate goal of query answering systems is to return *all* correct answers to a query $Q(x)$, whether observed (easy) or inferred (hard). This is far more ambitious than existing objectives: CQA does not involve classification, QAC evaluates a single candidate answer, and QAR asks only for one correct answer.
> > >
> > > Reaching this goal requires solving *two core challenges*:
> > > 1. reasoning beyond the observable graph to infer missing facts, and
> > > 2. efficiently processing structurally complex queries that push classical solvers to their limits.
> > >
> > > While prior work focuses mostly on the first challenge, the second has received less attention. Some recent benchmarks isolate specific structures (e.g., FIT for cycles, EFOk for multiple variables), but do not combine them. Our work shifts focus toward this second axis - reasoning over structurally hard queries.
> > >
> > > This motivation reflects how most neuro-symbolic CQA methods are structured. Typically, they reduce to a two-step process:
> > > 1. Neural completion: use a link predictor to augment the KG (e.g., tensorized matrices in QTO/FIT).
> > > 2. Symbolic reasoning: after step 1., the problem becomes purely combinatorial; apply classical algorithms over the completed graph (e.g., via propagation over the query graph).
> > >
> > > As a result, when classical solvers suffice, **most differences between CQA methods reduce to the strength of the underlying link predictor**.
> > >
> > > To move beyond this *complete-then-solve* paradigm, we propose a setup that is hard for classical solvers, breaking the assumption that symbolic reasoning over a completed graph can be handled by standard methods. Our large-query benchmarks include queries with 10–20 variables, multiple free variables, cycles, multi-edges, and non-anchored leaves, well beyond the 2-4 variable range, typically considered “complex” in the literature. This setup demands models that scale and reason, not ones relying on symbolic routines, easily mimicked by classical tools.
> > >
> > > Since conjunctive query answering is NP-complete, any system returning all correct answers must either:
> > > - become **intractable** as the structural complexity of the query grows, or
> > > - return an **incomplete** set of answers.
> > >
> > > As we aim to develop ML-based methods that remain efficient in such scenarios, we must accept that these models will, by design, **not always return the full set of answers** (or solve every QAC or QAR instance correctly). The objective is not exact completeness, but reliable and efficient reasoning over queries that are otherwise too complex for classical methods to process effectively. In particular, *we cannot expect even easy answers to be recovered with 100% accuracy*. Though easy answers are, in theory, simpler, their retrieval is still hard in the presence of complex query structures.  This also makes our task definitions more natural. Framing QAR as “Retrieve a correct answer, but not one that is observable, because those can be found by classical solvers” is problematic; not only because our benchmarks are explicitly designed to challenge classical solvers, but also because determining whether an answer is not ‘easy’ is itself non-trivial. In QAC, we could technically restrict evaluation to hard answers, but for consistency with QAR, we include both. Split protocols would only weaken coherence and introduce ambiguity.
> > >
> > > In summary, the inclusion of easy answers follows from the setup we target - one that exceeds the capabilities of both classical engines and existing CQA methods, and where finding any correct answer, observed or not, is challenging. Our primary goal is not to benchmark the classification abilities of CQA models; this could already be done using existing benchmarks by reframing them under a full classification objective (i.e., predicting all answers), with a wide range of methods available for adaptation. However, we believe such an investigation deserves a dedicated analysis, outside the scope of this paper.
> > >
> > > Instead, we introduce two intermediate objectives, QAC and QAR, and propose AnyCQ as a scalable, neuro-symbolic framework for complex queries. We do not suggest using AnyCQ for simple queries like “3p”, where approaches like FIT are more efficient. Our contribution is a general, scalable solution for queries where existing methods, even the classical engines, break down.
> > >
> > > Following the reviewer’s suggestion, we agree that this motivation could be made clearer. We can add a summary of this explanation to the updated manuscript to better justify our design decisions.

---

### Author Response · Authors · 2025-06-08

We thank all reviewers for their detailed, constructive, and insightful feedback. Your comments greatly helped us clarify and improve the presentation and scope of the paper.

In response, we have revised the manuscript accordingly. The updated version includes the following key changes:
- An expanded Related Work section (**Section 2.**), including a discussion of CQD-CO and a fix of the incorrect description of QTO.
- A revised introduction to LE and PE labels in **Section 5.1**, along with an extended explanation in **Appendix A.6**, to provide both formal and intuitive understanding of their role in the search process.
- Clarifications in **Section 6.1** and **Appendix C.2** regarding the design of our benchmark datasets, including our rationale for the selected query types and the distinction between QAR and QAC settings.
- A new visualization of query structures (**Figure 4**) to aid in understanding the complexity of the introduced large queries.
- General improvements to formatting and table layout for improved readability and consistency.

The revised manuscript is available alongside this response. We hope that these improvements address your concerns and better communicate the motivations, design decisions, and contributions of our work.

---

### Author Response · Authors · 2025-06-22

After an additional round of discussions with the reviewers, we are grateful for the thoughtful feedback and the opportunity to clarify our design choices. We believe that all major concerns have now been addressed, and we thank the reviewers for helping us improve the presentation of the manuscript.

We have made a final update to the manuscript, incorporating the following changes:
- We added a brief discussion in **Appendix A**, acknowledging known limitations of fuzzy logic, particularly in the context of large and complex queries, including issues related to score vanishing and unstable gradients.
- In **Appendix E**, we extended the evaluation of the SQL baseline by measuring its recall on easy instances from our QAR benchmarks under three timeout thresholds: 30, 60, and 120 seconds. The results show that AnyCQ consistently outperforms SQL on queries with two or more free variables - *even when the answers are present in the observable graph*. While SQL’s recall improves slightly with a longer timeout, it remains ineffective at processing large, cyclic queries within reasonable time limits.

We thank the reviewers once again for their insights and careful consideration. The revised manuscript is available in the updated submission, and the applied changes are marked in blue.

---

### Decision · Action_Editor_q8Et · 2025-08-02

**Recommendation:** Reject

**Additional Comments:**

Specifically, these are the aspects required to be addressed in the “major” revision:
- Amend all mentions to intractability, clarifying whether the issue is a polynomial-vs-exponential dichotomy in task complexity, and discussing the complexity of perfectly solving the QAC and QAR tasks, while distinguishing it with the complexity of the approximate classification with AnyCQ
- Amend the claims about calibrated probabilities, i.e., you cannot say that all models for rCQA are not calibrated or that QAC is providing inherently calibrated models. This can be done by providing (empirical) evidence that the probabilities of AnyCQ are indeed better calibrated than competitors, e.g., by designing and running novel experiments reporting ECE [d]. They should mention neural link predictions that are better calibrated [c] and do not require the calibration steps used for ComplEx or AnyCQ (which are the same of FIT/QTO)
- Run experiments on a variant of the benchmark from [e] where easy queries-answers pairs are omitted, to adapt to QAC to appreciate the difference w.r.t. rCQA where easy queries greatly inflate results but authors claim this should not happen on QAC. The benchmarks for QAC include more complex queries than those in [e] (nice!), but the authors can pick the query-answers pairs from [e] for some query types (e.g., 2p 3p pi ip inp pin or a subset) and do the same stratified analysis done in [e] to see if easy instances inflate performance as in rCQA. As the QAC benchmarks are now, this is not clear and while authors wanted to include these queries for retro-compatibility, they are doing it in a potentially contaminated way. This is important to show the community that the CQA benchmarks are solid and not skewed, something that unfortunately did not happen for rCQA until much recently [e].
- Add some retrocompatibility for QAR benchmarks, involving queries with a single free variable and introduce (some) neural link prediction as a baseline (e.g., FiT only the cyclic queries it can solve) to see where they stand and complement the SQL baseline. Adding these simpler queries will also help better understand the limitations of AnyCQ and where it starts shining w.r.t. SQL engines.  In fact, it seems from Table 13 that SQL can be better on simpler queries (e.g., 3-hub k=1).

[e] Gregucci, Cosimo, et al. "Is Complex Query Answering Really Complex?." ICML 2025

**Audience:**

Yes

**Audience Explanation:**

All reviewers appreciated the novel perspective around QAC/QAR and the clever methodology behind AnyCQ. This paper contains good material that with one more iteration can be turned into a strong submission that can be of interest not only for the TMLR community but also for the broader ML audience. The current manuscript however has some shortcomings.

During the discussion, the reviewers highlighted some concerns, such as the lack of clarity between the distinction of QAC and QAR w.r.t. rCQA, the motivation behind the experimental choices (e.g. why including/excluding easy queries in the benchmark and the different design choices for the QAC/QAR benchmarks) and the claims around old rCQA benchmarks (e.g., acknowledging previous benchmarks that already go beyond some of structural limitations and clearly stating why even partial experiment of QAC are not possible on existing benchmarks).

The authors answered many of the criticisms raised by reviewers in the discussion (such as the nominal distinction between QAC/QAR and rCQA, well done!), but some answers were unsatisfactory and still need to be addressed.  In the end, the reviewers split concerning recommendation (“Accept” on the one hand and “Reject” on the other) and there is no easy way to reconcile them.
I understand both perspectives, and I believe that the concerns raised can be easily addressed in a new revision. As TMLR has no option for a “major” revision though, I recommend rejection at this round but warmly invite authors to resubmit.

**Claims And Evidence:**

No

**Claims Explanation:**

The paper introduces the novel tasks of Query Answer Classification (QAC) and Query Answer Retrieval (QAR) as opposed to the traditional ranking-based complex query answering (rCQA) to address some limitations of many models introduced for rCQA. And a novel neuro-symbolic framework AnyCQ is proposed to solve QAC and QAR.
These limitations as stated by the authors are i) intractability of rCQA (*), ii) limited structural complexity of existing rCQA benchmarks and iii) lack of probabilistic calibration of rCQA methods.

Introducing QAC and QAR as new tasks has its own merits, and deserves attention. However, at the same time, as highlighted by some reviewers, these claims need to be revised further. In fact, QAC/QAR (and AnyQC) partially addresses the issues claimed by authors.
In fact, i) the proposed formulations of QAC/QAR look intractable and in fact they should be as one could translate from Boolean satisfiability as mentioned by the authors. Instead, the authors might want to say that training a classifier to approximately solve QAC/QAR could be more scalable than learning to rank triples in rCQA.
Many rCQA benchmarks have already tried to address issue ii), but in a fragmented way. Their role is dismissed too easily and it would be helpful to see the performance of AnyCQ on some common ground, e.g., on some old benchmarks adapted to QAC/QAR, see comments below.
Finally, iii) there is no further reference on the probabilistic calibration capabilities of AnyCQ despite the fact that it outputs a probability. Outputting a probability instead of an energy does (like many classical neural link predictors) not guarantee calibration, even for classifiers. Authors need to include calibration scores and ECE as done for neural link predictors in [b, c] or their analogous to classification [d] if they want to retain the claim about calibration.

(*) this is incorrect (and somehow common mistake in the ML literature) and needs to be changed in the revision everywhere “intractability” is mentioned: the quadratic complexity mentioned for QAR is polytime and hence not intractable. It is infeasible in practice, that is true. The dichotomy theorem of [a] states that many queries can be solved in polytime.

[a] Dalvi, Nilesh, and Dan Suciu. "Efficient query evaluation on probabilistic databases." The VLDB Journal 16.4 (2007): 523-544.

[b] Zhu, Ruiqi, et al. "A closer look at probability calibration of knowledge graph embedding." Proceedings of the 11th International Joint Conference on Knowledge Graphs. 2022.

[c] Loconte, Lorenzo, et al. "How to turn your knowledge graph embeddings into generative models." Advances in Neural Information Processing Systems 36 (2023): 77713-77744.

[d] Guo, Chuan, et al. "On calibration of modern neural networks." International conference on machine learning. PMLR, 2017.

**Resubmission Of Major Revision:**

The authors may consider submitting a major revision at a later time.

---

> ### Author Response · Authors · 2025-08-06
>
> Unfortunately, the meta-review provided by the Action Editor exhibits several serious misunderstandings and as a result misplaces the contribution of our work. Most importantly, the Action Editor answers the question **“Are the claims made in the submission supported by accurate, convincing and clear evidence?”** as “No”, and uses this as a basis for rejection.  To substantiate this answer, the action editor states our motivation as:
>
> > “These limitations as stated by the authors are i) intractability of rCQA (*), ii) limited structural complexity of existing rCQA benchmarks and iii) lack of probabilistic calibration of rCQA methods.”
>
>
> We explain below concretely that these points are either factually incorrect or simple misunderstandings:
>
> ---
>
> Regarding (i), the meta-review states:
>
> >“In fact, i) the proposed formulations of QAC/QAR look intractable and in fact they should be as one could translate from Boolean satisfiability as mentioned by the authors.”
>
> First of all, the fact that QAC/QAR is computationally intractable does not mean that our approach is intractable. In fact, our model is only a neural approximation which is efficient, but provides no guarantees in terms of accuracy. The goal of our work is **NOT** to reformulate the original query answering problem with a tractable version, but rather introduce a problem setup, which is useful in practice and solve it efficiently though a new method. This is exactly what our work claims to achieve and we are happy to receive constructive criticism regarding this.
>
> ---
>
> > “(*) this is incorrect (and somehow common mistake in the ML literature) and needs to be changed in the revision everywhere “intractability” is mentioned: the quadratic complexity mentioned for QAR is polytime and hence not intractable. It is infeasible in practice, that is true. The dichotomy theorem of [a] states that many queries can be solved in polytime.
> [a] Dalvi, Nilesh, and Dan Suciu. "Efficient query evaluation on probabilistic databases." The VLDB Journal 16.4 (2007): 523-544.”
>
> We are well aware of the complexity of (probabilistic) query evaluation, and are surprised to see this “correction” which is claimed to be a common “mistake in the ML literature”.  Probabilistic query evaluation for (unions of) conjunctive queries is intractable in combined complexity (even classical conjunctive query answering is intractable in combined complexity). This is precisely our setting and rCQA and QAC/QAR are all intractable, because they contain at least the classical query answering problem which is known to be intractable since the seminal result of Chandra & Merlin (Chandra, Ashok K. and Philip M. Merlin (1977). “Optimal Implementation of Conjunctive Queries in Relational Data Bases”. STOC-77).
>
> Nilesh and Suciu’s paper — cited by the AE — on the other hand refers to data complexity of probabilistic query evaluation. This is not our setting and we make no assumptions on the query being “fixed” when discussing the intractability of the problems. Moreover, even if we were in the context of data complexity, it is clear from the famous result of Nilesh and Suciu that almost all queries are intractable, where tractable queries are mostly limited to hierarchical queries. To give an idea, we would like to point to one intractable query (in data complexity): $C(x) \land r(x,y) \land D(y)$. This query is tree-shaped, acyclic, and yet #P-hard.
>
> AE states that QAR is quadratic and hence in poly time, which is also incorrect: QAR is a hard problem as stated before, and what is poly-time is the algorithm and not the problem itself.
>
> ---
>
> > “Instead, the authors might want to say that training a classifier to approximately solve QAC/QAR could be more scalable than learning to rank triples in rCQA.”
>
> This is exactly the claim of the paper. The problems are all hard and intractable, but one problem formulation lends itself to better approximations, and that’s what our method exploits. We concretely write why ranking algorithms are infeasible using exact same wording the AE suggests: “Already for queries $Q(x_1, x_2)$ with two free variables, this entails scoring  $|V(G)|^2$ pairs of entities $(a_1, a_2) \in V(G)^2$, which is computationally infeasible for modern knowledge graphs [1,2] containing thousands of nodes.”

---

> > ### Comment · Action_Editor_q8Et · 2025-08-10
> > **Please read and revise section 4.1 (I)**
> >
> > I thank the authors for engaging in a discussion. I carefully re-read my comments in the decision, the relevant paragraphs of the paper as referenced in the authors’ further comments and I invite the authors to do the same. I explain below why my comments for rejection are valid.
> >
> > > Unfortunately, the meta-review provided by the Action Editor exhibits several serious misunderstandings and as a result misplaces the contribution of our work. Most importantly, the Action Editor answers the question “Are the claims made in the submission supported by accurate, convincing and clear evidence?” as “No”, and uses this as a basis for rejection
> >
> > My comments are based on what you wrote on the paper. In particular, in the introduction but more crucially in section 4.1 which is titled “Limitations of existing problem formulations” and lists 4 paragraphs:
> >
> > Intractability of high-arity query evaluation
> > Limited structural complexity in existing benchmarks
> > Lack of probabilistic calibration in ranking-based methods.
> >
> > Later, in Section 4.2 authors write
> >
> > > To address these limitations, we propose two new query answering tasks designed to provide more targeted responses while ensuring scalability for more complex logical queries
> >
> > This writing is problematic as it implies that the three points above are a limitation of rCQA and QAC and QAR overcome them. As I wrote in my decision, there are misleading and need to be addressed for a resubmission.
> >
> >
> > > First of all, the fact that QAC/QAR is computationally intractable does not mean that our approach is intractable. The goal of our work is NOT to reformulate the original query answering problem with a tractable version, but rather introduce a problem setup, which is useful in practice and solve it efficiently though a new method. This is exactly what our work claims to achieve and we are happy to receive constructive criticism regarding this.
> >
> > There is a misunderstanding here. From your side.
> > The way you write in the introduction about rCQA **“Firstly, this evaluation becomes intractable for cases where multiple free variables are allowed”** and as in the first paragraph of Section 4.1 makes the reader understand that rCQA is intractable whereas QAC/R are not. This is not correct, as already discussed.
> >
> > Requested changes:
> > - Write that rCQA and QAC/R are all potentially intractable if one wants to solve them exactly. Therefore, QAC/R should not be introduced as solving the intractability of rCQA.
> > - Write that rCQA admits polytime exacts solutions for several queries, citing Suciu’s dichotomy.
> > - Discuss the complexity of AnyCQ as well as of the neural approximations of rCQA, mentioning the latter are still polytime hence not intractable
> > - Therefore amend all references of “intractable” as in your writing it is confusing with “scales badly on large KGs”. E.g.,  **“Firstly, this evaluation becomes intractable for cases where multiple free variables are allowed”** can be turned into **“Firstly, this evaluation becomes ***impractical*** for cases where multiple free variables are allowed”** as it can still be done in quadratic time exactly
> >
> > Now, regarding combined, vs data complexity, of course the combined is always going to be higher, but in ML we can always assume to know the query. If the authors want to keep the distinction about combined vs data complexity, they have to write it explicitly. Note that even under the light of combined complexity the claim that the introduced QAC/R address the limitation of rCQA is still factually wrong.
> >
> > > Nilesh and Suciu that almost all queries are intractable, where tractable queries are mostly limited to hierarchical queries.
> >
> > Consider the  tractability according to Suciu’s of the queries used in classical rCQA benchmarks as in [e]: all are tractable but 3/4p. In practice, at least in the current ML benchmarks, most of the queries could be solved in quadratic time.
> >
> > > AE states that QAR is quadratic and hence in poly time, which is also incorrect: QAR is a hard problem as stated before, and what is poly-time is the algorithm and not the problem itself.
> >
> > Apologies, that is clearly a typo. It is rCQA and not QAR, as understandable from the context, and referring to what you write in the first paragraph of section 4.1, which is called “Intractability” but mentions only a quadratic complexity issue when referring to rCQA.
> >
> > > This is exactly the claim of the paper. The problems are all hard and intractable, but one problem formulation lends itself to better approximations, and that’s what our method exploits.
> >
> > Please write explicitly in the paper (as requested above) that “The problems are all hard and intractable”. As the introduction is written, and section 4.1, this is not evident at all.

---

> ### Author Response · Authors · 2025-08-06
> **Official Comment by Authors (2/2)**
>
> > “Amend all mentions to intractability, clarifying whether the issue is a polynomial-vs-exponential dichotomy in task complexity, and discussing the complexity of perfectly solving the QAC and QAR tasks, while distinguishing it with the complexity of the approximate classification with AnyCQ”
>
> Given our explanations, we do not see why the claim of intractability should be amended. The complexity of query answering is very well-known, and QAC and QAR contain query answering (in the simple case where the graph is complete). However, discussing the precise complexity of QAC and QAR is not possible without a formal description of the complete graph, as this is a prediction task relative to a complete, unknown graph.
>
> We sincerely hope that this note will help clarify these misunderstandings, and we are happy to change any ambiguous phrasing. Since none of these points came up during the rebuttal period, we never had a chance to respond, and we find it rather disappointing to see this point being used as one of the reasons for rejection.
>
> ---
>
> Regarding (ii), the meta-review states:
>
> > “Many rCQA benchmarks have already tried to address issue ii), but in a fragmented way. Their role is dismissed too easily and it would be helpful to see the performance of AnyCQ on some common ground, e.g., on some old benchmarks adapted to QAC/QAR, see comments below. “
>
> This is an unfair statement, as we have not dismissed these works without crediting them appropriately. When they could not be used in our task setting, we provided concrete reasons.
>
> ---
>
> Regarding (iii), the meta-review states:
>
> > “Finally, iii) there is no further reference on the probabilistic calibration capabilities of AnyCQ despite the fact that it outputs a probability. Outputting a probability instead of an energy does (like many classical neural link predictors) not guarantee calibration, even for classifiers. Authors need to include calibration scores and ECE as done for neural link predictors in [b, c] or their analogous to classification [d] if they want to retain the claim about calibration.”
>
> We do not claim that our method provides calibrated probabilities anywhere in the paper. Our claim is that rCQA methods generally do not provide calibrated probabilities, and as a result, when we aim to use them in a classification/query answering setup then we face problems in adapting them to this task setting. We experienced this problem first-hand when we tried to use rCQA methods in our setup, because determining a threshold was really a hard task on its own. Our model, on the other hand, is trained directly for the classification task, and therefore naturally adapts to this task setting, regardless of how the generated score distribution looks. This is a subtle but important distinction.
>
> ---
>
> > “Amend the claims about calibrated probabilities, i.e., you cannot say that all models for rCQA are not calibrated or that QAC is providing inherently calibrated models.”
>
> Nowhere in the paper do we claim that all rCQA models are uncalibrated or that QAC inherently produces calibrated models. Rather, we state that many existing rCQA approaches do not provide calibrated probabilities, as their objective is to rank potential answers without requiring a classification threshold. We clarify in the text that methods such as QTO, FIT, NBFNet, and Ultra can be used in a classification setting, and we indeed evaluate them under the QAC task. However, framing the task as classification (QAC, QAR) introduces a need for thresholding decisions, which naturally benefits from calibrated predictions. This is evidenced in our experiments: for instance, models like NBFNet yield more reliable probability estimates than traditional embedding-based models like ComplEx in this setting.
>
> ---
>
> > “This can be done by providing (empirical) evidence that the probabilities of AnyCQ are indeed better calibrated than competitors, e.g., by designing and running novel experiments reporting ECE [d]. They should mention neural link predictions that are better calibrated [c] and do not require the calibration steps used for ComplEx or AnyCQ (which are the same of FIT/QTO)”
>
> These are already provided in Appendix D.5 when testing different link predictors for AnyCQ. We empirically verify that NBFNets, which are trained using a classification-based objective and work in the probability domain exhibit much stronger performance than energy-style ComplEx:
>
> *"For the small-query QAC splits, we notice that NBFNet and Ultra strongly outperform ComplEx on positive formulas..."* \
> *"For large query classification, NBFNet produces much better F1-scores, exceeding 50% on almost all splits, while the remaining predictors consistently score under 40%"*
>
> ---
> [1] Bordes, Antoine, et al. "Translating embeddings for modeling multi-relational data." NIPS (2013).
>
> [2] Carlson, Andrew, et al. "Toward an architecture for never-ending language learning." AAAI (2010).

---

> > ### Comment · Action_Editor_q8Et · 2025-08-10
> >
> > > However, discussing the precise complexity of QAC and QAR is not possible without a formal description of the complete graph, as this is a prediction task relative to a complete, unknown graph.
> >
> > It is not necessary to give the precise complexity but the worst-case intractability is known and you should openly write it down and amend the claim that QAC/R solve the intractability problem of rCQA.
> >
> > > Since none of these points came up during the rebuttal period, we never had a chance to respond, and we find it rather disappointing to see this point being used as one of the reasons for rejection.
> >
> > This is a fair point! Unfortunately, I had to read the paper in detail by myself to try to mediate among reviewers and this issue about claim #1 stood out. Note that reviewers however, already raised the other criticisms about the other claims.
> >
> > > This is an unfair statement, as we have not dismissed these works without crediting them appropriately. When they could not be used in our task setting, we provided concrete reasons.
> >
> > Please re-read the comments from reviewer uUPs carefully and try to incorporate them in your revision, starting from the citations they are asking you to include. I tried my best to distill two actionable points (which honestly are less demanding than what the reviewer asked) among the original requests of the reviewer. The reviewer raised meaningful concerns on how to show some degree of retro-compatibility between your benchmarks and old ones. I see you are not arguing about the two requests I distilled, so I am confident you will incorporate them in the next revision.
> >
> >
> > > We do not claim that our method provides calibrated probabilities anywhere in the paper.
> >
> > Well, you write in section 4.1 that one of the three issues is that “Lack of probabilistic calibration in ranking-based methods” and that most rCQA methods “do[es] not guarantee the output scores correspond to calibrated satisfiability probabilities”. Then section 4.2 starts by saying that QAC/R solve the above limitations. This claim needs to be revised. You have also written that results in Appendix D.5 provide evidence for better calibration.
> >
> > >  Our claim is that rCQA methods generally do not provide calibrated probabilities, and as a result, when we aim to use them in a classification/query answering setup then we face problems in adapting them to this task setting.
> >
> > This is a circular loop: you first want a new task, then you say that old models cannot be easily adapted for the new task. From how Sections 4.1/4.2 are structured however, you introduce the new tasks to solve the limitations of the old models/tasks. Please rewrite section 4.1 and 4.2 accordingly.
> >
> > > We experienced this problem first-hand when we tried to use rCQA methods in our setup, because determining a threshold was really a hard task on its own.
> >
> > Indeed, I do not doubt it! That is also one of the motivations why [c] trains neural link predictors by maximum likelihood and not noise contrastive estimation. However, even doing that is not enough, see below.
> >
> > > This is evidenced in our experiments: for instance, models like NBFNet yield more reliable probability estimates than traditional embedding-based models like ComplEx in this setting. [...] These are already provided in Appendix D.5 when testing different link predictors for AnyCQ. We empirically verify that NBFNets, which are trained using a classification-based objective and work in the probability domain exhibit much stronger performance than energy-style ComplEx:
> >
> > There is a misunderstanding here, as you are pointing to downstream performance in terms of F1 scores. Note that better (average) values for accuracy (or F1 scores) do not necessarily imply calibration. An accurate neural network can be uncalibrated. Furthermore, just training with cross entropy also does not guarantee calibration (otherwise all neural nets trained by cross entropy classification should be automatically calibrated). Please revisit metrics such as ECE and the references I provided in [b-d], and amend the claims as I suggest in my decision post.
> >
> > All in all, as I stated in the my original decision, I do believe that QAC and QAR are two interesting tasks per se that deserve publication. At the same time, the way the manuscript is currently written, and the lack of a "gentle" but rigorous empirical transition from old benchmarks to new ones, make me opt for "major revision". I warmly invite authors to reread their section 4, amend it, add the missing experiments and resubmit.
> >
> > I am also available for further clarifications.

---

> > > ### Author Response · Authors · 2025-08-13
> > > **Asking clarifications for two points**
> > >
> > > We acknowledge that the sentence at the beginning of Sec 4.2 could be interpreted as the AE suggests (and this should be changed), but this is a ***single sentence*** in the whole paper, and could have been revised very easily to clarify any misunderstandings. Apart from this point, we are puzzled by the additional responses given by the AE, as they suggest (more) misinterpretations including the literature.
> > >
> > > 1. We would be thrilled if the AE could explain precisely what is meant by the following:
> > > >Now, regarding combined, vs data complexity, of course the combined is always going to be higher, but in ML we can always assume to know the query.
> > >
> > >     Specifically, why do we know the query in ML? We are not aware of any such convention in the rCQA literature. Are there any credible references to back up this point?
> > >
> > > 2. Some of the revisions requested by the AE cause further confusion. For example, they ask us to cite the data complexity dichotomy result of Dalvi and Nilesh related to the problem of rCQA. How is probabilistic query evaluation (PQE) related to rCQA or the problems studied in our paper? The formal definitions of the studied problems are as follows:
> > >
> > >     PQE:
> > >
> > >     - *Input*: a probabilistic database (each fact has a probability) D and a fixed query q
> > >     - *Output*: the probability of the query on this database: P_D(q)
> > >
> > >     rCQA:
> > >
> > >     - *Input*: a knowledge graph (no probabilities for triples) G and a query q
> > >     - *Output*: a ranking of the answers based on some scoring function which ranks “true answers” higher than “false answers”, which are determined relative to a complete version of the graph G.
> > >
> > >     Since these revisions is requested from a complexity-theoretic viewpoint, and we fail to see a reduction between these problems, it becomes essential to answer this question to accommodate the requested change.

---

> > > > ### Comment · Action_Editor_q8Et · 2025-08-18
> > > >
> > > > Thanks for following up, I am answering below.
> > > >
> > > > > We acknowledge that the sentence at the beginning of Sec 4.2 could be interpreted as the AE suggests (and this should be changed), but this is a single sentence in the whole paper, and could have been revised very easily to clarify any misunderstandings
> > > >
> > > > I am afraid it is not just a sentence, otherwise we won't be doing a major revision. As per my previous comments (please have a careful read!), the whole section 4.1, and parts of the introduction as noted before, need to be rewritten, because they are claiming that there are three limitations of rCQA that QAC/R solve. This is not true (as discussed above). I asked two other senior colleagues to read and give me their opinion and they also agree that the current paper reads in that way.
> > > >
> > > > > Specifically, why do we know the query in ML? We are not aware of any such convention in the rCQA literature. Are there any credible references to back up this point?
> > > >
> > > > Consider the current benchmarks and how we solve them: we compute the probability of each query given each possible completion. See also answer below.
> > > >
> > > > > Some of the revisions requested by the AE cause further confusion. For example, they ask us to cite the data complexity dichotomy result of Dalvi and Nilesh related to the problem of rCQA. How is probabilistic query evaluation (PQE) related to rCQA or the problems studied in our paper?
> > > >
> > > > You are almost there!
> > > > Consider the following: every link predictor plus a knowledge graph yields a probabilistic database, as the link predictor outputs the probability of each triple in the KG, which in turn are tuples of the PDB. Note that it does not matter if probabilities are calibrated w.r.t. some ground truth for the reduction (it matters for downstream performance). This connection has been done in papers like [f].
> > > > Now, to answer rCQA, we rank the possible completions based on the probability of each complete query according to the PDB constructed before. Note that the task of CQA does not require ranking per se, but requires computing the entity completion that maximises the probability of a query. Other algorithms apart from those proposed in [f], such as QTO/FIT should perform something extremely similar to Dalvi and Suciu's algorithm to get the max prob completion.
> > > >
> > > > I hope this clarifies the last two doubts. I am assuming you accepted the criticism about the lack of continuity w.r.t. previous benchmarks and the misinterpretation of calibration.
> > > >
> > > >
> > > >
> > > > [f] Friedman, Tal, and Guy Broeck. "Symbolic querying of vector spaces: Probabilistic databases meets relational embeddings." Conference on Uncertainty in Artificial Intelligence. PMLR, 2020.